# Parametrically Retargetable Decision-Makers Tend To Seek Power

**Alexander Matt Turner, Prasad Tadepalli**
Oregon State University
{turneale@, tadepall@eecs.}oregonstate.edu

## Abstract

If capable AI agents are generally incentivized to seek power in service of the objectives we specify for them, then these systems will pose enormous risks, in addition to enormous benefits. In fully observable environments, most reward functions have an optimal policy which seeks power by keeping options open and staying alive [Turner et al., 2021]. However, the real world is neither fully observable, nor must trained agents be even approximately reward-optimal. We consider a range of models of AI decision-making, from optimal, to random, to choices informed by learning and interacting with an environment. We discover that many decision-making functions are *retargetable*, and that retargetability is sufficient to cause power-seeking tendencies. Our functional criterion is simple and broad. We show that a range of qualitatively dissimilar decision-making procedures incentivize agents to seek power. We demonstrate the flexibility of our results by reasoning about learned policy incentives in Montezuma's Revenge. These results suggest a safety risk: Eventually, retargetable training procedures may train real-world agents which seek power over humans.

## 1 Introduction

Bostrom [2014], Russell [2019] argue that in the future, we may know how to train and deploy superintelligent AI agents which capably optimize goals in the world. Furthermore, we would not want such agents to act against our interests by ensuring their own survival, by gaining resources, and by competing with humanity for control over the future.

Turner et al. [2021] show that most reward functions have optimal policies which seek power over the future, whether by staying alive or by keeping their options open. Some Markov decision processes (MDPs) cause there to be *more ways* for power-seeking to be optimal, than for it to not be optimal. Analogously, there are relatively few goals for which dying is a good idea.

We show that a wide range of decision-making algorithms produce these power-seeking tendencies—they are not unique to reward maximizers. We develop a simple, broad criterion of functional retargetability (definition 3.5) which is a sufficient condition for power-seeking tendencies. Crucially, these results allow us to reason about what decisions are incentivized by most algorithm parameter inputs, even when it is impractical to compute the agent's decisions for any given parameter input.

Useful "general" AI agents could be directed to complete a range of tasks. However, we show that this flexibility can cause the AI to have power-seeking tendencies. In section 2 and section 3, we discuss how a "retargetability" property creates statistical tendencies by which agents make similar decisions for a wide range of parameter settings for their decision-making algorithms. Basically, if a decision-making algorithm is retargetable, then for every configuration under which a decision-making algorithm does not choose to seek power, there exist several reconfigurations which do induce power-seeking. More formally, for every decision-making parameter setting $\theta$ which does not induce

power-seeking, $n$-retargetability ensures we can injectively map $\theta$ to $n$ parameters $\theta'_1, \ldots, \theta'_n$ which *do* induce power-seeking.

Equipped with these results, section 4 works out agent incentives in the Montezuma's Revenge game. Section 5 speculates that increasingly useful and impressive learning algorithms will be increasingly retargetable, and how retargetability can imply power-seeking tendencies. By this reasoning, increasingly powerful RL techniques may (eventually) train increasingly competent real-world power-seeking agents. Such agents could be unaligned with human values [Russell, 2019] and—we speculate—would take power from humanity.

## 2 Statistical tendencies for a range of decision-making algorithms

Turner et al. [2021] consider the Pac-Man video game, in which an agent consumes pellets, navigates a maze, and avoids deadly ghosts (Figure 1). Instead of the usual score function, Turner et al. [2021] consider optimal action across a range of state-based reward functions. They show that most reward functions have an (average-)optimal policy which avoids immediate death in order to navigate to a future terminal state.[1]

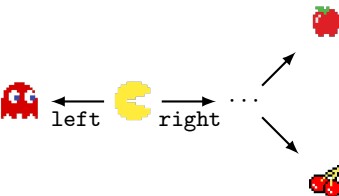

Figure 1: If Pac-Man goes `left`, he dies to the ghost and ends up in the 👻 outcome. If he goes `right`, he can reach the 🍎 and 🍒 terminal states.

Our results show that optimality is not required. Instead, if the agent's decision-making is *parametrically retargetable* from death to other outcomes, Pac-Man avoids the ghost under most decision-making parameter inputs. To build intuition about these notions, consider three outcomes (*i.e.* terminal states): Immediate death to a nearby ghost, consuming a cherry, and consuming an apple. Let $A := \{👻\}$ and $B := \{🍎, 🍒\}$. For simplicity of exposition, we assume these are the three possible terminal states.

Suppose that in some fashion, the agent probabilistically decides on an outcome to induce. Let $p$ take as input a set of outcomes and return the probability that the agent selects one of those outcomes. For example, $p(\{👻\})$ is the probability that the agent selects 👻, and $p(\{🍎, 🍒\})$ is the probability that the agent escapes the ghost and ends up in an apple or cherry terminal state. But this just amounts to a probability distribution over the terminal states. We want to examine how decision-making *changes* as we swap out the parameter inputs to the agent's decision-making algorithm with decision-making parameter space $\Theta$. We then let $p(X \mid \theta)$ take as input a set of outcomes $X$ and a decision-making algorithm parameter setting $\theta \in \Theta$, and return the probability that the agent chooses an outcome in $X$.

We first consider agents which maximize terminal-state utility, following Turner et al. [2021] (in their language, "average-reward optimality"). Suppose that the agent has a utility function parameter $\mathbf{u}$ assigning a real number to each of the three outcomes. Then the relevant parameter space is the agent's utility function $\mathbf{u} \in \Theta := \mathbb{R}^3$. $p_{\max}(A \mid \mathbf{u})$ indicates whether 👻 has the most utility: $\mathbf{u}(👻) \geq \max(\mathbf{u}(🍎), \mathbf{u}(🍒))$. Consider the utility function $\mathbf{u}$ in Table 1. Since 👻 has strictly maximal utility, the agent selects 👻: $p_{\max}(A \mid \mathbf{u}) = 1 > 0 = p_{\max}(B \mid \mathbf{u})$.

---

[1]We use "reward function" somewhat loosely in implying that reward functions reasonably describe a trained agent's goals. Turner [2022] argues that capable RL algorithms do not necessarily train policy networks which are best understood as optimizing the reward function itself. Rather, they point out that—especially in policy-gradient approaches—reward provides gradients to the network and thereby modifies the network's generalization properties, but doesn't ensure the agent generalizes to "robustly optimizing reward" off of the training distribution.

However, most "variants" of $\mathbf{u}$ have an optimal policy which stays alive. That is, for every $\mathbf{u}$ for which immediate death is optimal but immediate survival is not, we can swap the utility of *e.g.* 👾 and 🍎 via permutation $\phi_{👾\leftrightarrow🍎}$ to produce a new utility function $\mathbf{u}' := \phi_{👾\leftrightarrow🍎} \cdot \mathbf{u}$ for which staying alive (right) is strictly optimal. The same kind of argumentation holds for $\phi_{👾\leftrightarrow🍒}$. Table 1 suggests a counting argument. For every utility function $\mathbf{u}$ for which 👾 is optimal, there are two unique utility functions $\phi_1 \cdot \mathbf{u}, \phi_2 \cdot \mathbf{u}$ under which either 🍎 or 🍒 is optimal.

| Utility function | 👾 | 🍎 | 🍒 |
|---:|:---:|:---:|:---:|
| $\mathbf{u}$ | **10** | 5 | 0 |
| $\phi_{👾\leftrightarrow🍎} \cdot \mathbf{u}$ | 5 | **10** | 0 |
| $\phi_{👾\leftrightarrow🍒} \cdot \mathbf{u}$ | 0 | 5 | **10** |
| $\mathbf{u}'$ | **10** | 0 | 5 |
| $\phi_{👾\leftrightarrow🍎} \cdot \mathbf{u}'$ | 0 | **10** | 5 |
| $\phi_{👾\leftrightarrow🍒} \cdot \mathbf{u}'$ | 5 | 0 | **10** |

Table 1: The highest-utility outcome is bolded. Because $B$ contains more outcomes than $A$, most utility functions incentivize the agent to stay alive and therefore select a state from $B$. For every utility function $\mathbf{u}$ or $\mathbf{u}'$ which makes 👾 strictly optimal, *two* of its permuted variants make an outcome in $B := \{🍎, 🍒\}$ strictly optimal. We permute $\mathbf{u}$ by swapping the utility of 👾 and the utility of 🍎, using the permutation $\phi_{👾\leftrightarrow🍎}$. The expression "$\phi_{👾\leftrightarrow🍎} \cdot \mathbf{u}$" denotes the permuted utility function.

In section 3, we will generalize this particular counting argument. Definition 3.3 shows a functional condition (*retargetability*) under which the agent decides to avoid the ghost, for most parameter inputs to the decision-making algorithm. Given this retargetability assumption, proposition 3.4 roughly shows that most $\theta \in \Theta$ induce $p(B \mid \theta) \geq p(A \mid \theta)$. First, consider two more retargetable decision-making functions:

**Uniformly randomly picking a terminal state.** $p_{\text{rand}}$ ignores the reward function and assigns equal probability to each terminal state in Pac-Man's state space.

**Choosing an action based on a numerical parameter.** $p_{\text{numerical}}$ takes as input a natural number $\theta \in \Theta := \{1, \ldots, 6\}$ and makes decisions as follows:

$$p_{\text{numerical}}(A \mid \theta) := \begin{cases} 1 & \text{if } \theta = 1, \\ 0 & \text{otherwise.} \end{cases} \qquad p_{\text{numerical}}(B \mid \theta) := 1 - p_{\text{numerical}}(A \mid \theta). \qquad (1)$$

In this situation, $\Theta$ is acted on by permutations over 6 elements $\phi \in S_6$. Then $p_{\text{numerical}}$ is retargetable from $A$ to $B$ via $\phi_k : 1 \leftrightarrow k, k \neq 1$.

$p_{\text{max}}, p_{\text{rand}}$, and $p_{\text{numerical}}$ encode varying sensitivities to the utility function parameter input, and to the internal structure of the Pac-Man decision process. Nonetheless, they all are retargetable from $A$ to $B$. For an example of a *non*-retargetable function, consider $p_{\text{stubborn}}(X \mid \theta) := \mathbb{1}_{X=A}$ which returns 1 for $A$ and 0 otherwise.

However, we cannot explicitly define and evaluate more interesting functions, such as those defined by reinforcement learning training processes. For example, given that we provide such-and-such reward function in a fixed task environment, what is the probability that the learned policy will take action $a$? We will analyze such procedures in section 4.

We now motivate the title of this work. For most parameter settings, retargetable decision-makers induce an element of the larger set of outcomes. Such decision-makers *tend to* induce an element of a larger set of outcomes (with the "tendency" being taken across parameter settings). Consider that the larger set of outcomes $\{🍒, 🍎\}$ can only be induced if Pac-Man stays alive. Intuitively, navigating to this larger set is *power-seeking* because the agent retains more optionality (*i.e.* the agent can't do anything when dead). Therefore, *parametrically retargetable decision-makers tend to seek power*.

## 3 Formal notions of retargetability and decision-making tendencies

Section 2 informally illustrated parametric retargetability in the context of swapping which utilities are assigned to which outcomes in the Pac-Man video game. For many utility-based decision-making

algorithms, swapping the utility assignments also swaps the agent's final decisions. For example, if death is anti-rational, and then death's utility is swapped with the cherry utility, then now the cherry is anti-rational. In this section, we formalize the notion of parametric retargetability and of "most" parameter inputs producing a given result. In section 4, we will use these formal notions to reason about the behavior of RL-trained policies in the Montezuma's Revenge video game.

To define our notion of "retargeting", we assume that $\Theta$ is a subset of a set acted on by symmetric group $S_d$, which consists of all permutations on $d$ items (*e.g.* in the RL setting, this might represent states or observations). A parameter $\theta$'s *orbit* is the set of $\theta$'s permuted variants. For example, Table 1 lists the six orbit elements of the parameter $\mathbf{u}$.

**Definition 3.1** (Orbit of a parameter). Let $\theta \in \Theta$. The *orbit* of $\theta$ under the symmetric group $S_d$ is $S_d \cdot \theta := \{\phi \cdot \theta \mid \phi \in S_d\}$. Sometimes, $\Theta$ is not closed under permutation. In that case, the *orbit inside* $\Theta$ is $\mathrm{Orbit}|_\Theta (\theta) := (S_d \cdot \theta) \cap \Theta$.

Let $p(B \mid \theta)$ return the probability that the agent chooses an outcome in $B$ given $\theta$. To express "$B$-outcomes are chosen instead of $A$-outcomes", we write $p(B \mid \theta) > p(A \mid \theta)$. However, even "retargetable" decision-making functions (defined shortly) generally won't choose a $B$-outcome for *every* input $\theta$. Instead, we consider the *orbit-level tendencies* of such decision-makers, showing that for every parameter input $\theta \in \Theta$, most of $\theta$'s permutations push the decision towards $B$ instead of $A$.

**Definition 3.2** (Inequalities which hold for most orbit elements). Suppose $\Theta$ is a subset of a set acted on by $S_d$, the symmetric group on $d$ elements. Let $f : \{A, B\} \times \Theta \to \mathbb{R}$ and let $n \geq 1$. We write $f(B \mid \theta) \geq^n_{\mathrm{most:}\ \Theta} f(A \mid \theta)$ when, for *all* $\theta \in \Theta$, the following cardinality inequality holds:

$$\left| \left\{\theta' \in \mathrm{Orbit}|_\Theta (\theta) \mid f(B \mid \theta') > f(A \mid \theta')\right\} \right| \geq n \left| \left\{\theta' \in \mathrm{Orbit}|_\Theta (\theta) \mid f(B \mid \theta') < f(A \mid \theta')\right\} \right|. \tag{2}$$

For example, Table 1 illustrates the tendency of $\mathbf{u}$'s orbit to make $B := \{🍒, 🥦\}$ optimal over $A := \{👾\}$. Turner et al. [2021]'s definition 6.5 is the special case of definition 3.2 where $n = 1$, $d = |\mathcal{S}|$ (the number of states in the considered MDP), and $\Theta \subseteq \Delta(\mathbb{R}^{|\mathcal{S}|})$.

As explored previously, $p_{\mathrm{rand}}$, $p_{\max}$, and $p_{\mathrm{numerical}}$ are retargetable: For all $\theta \in \Theta$ such that $\{👾\}$ is chosen over $\{🍒, 🥦\}$, we can permute $\theta$ to obtain $\phi \cdot \theta$ under which the opposite is true. More generally, we can consider retargetability from some set $A$ to some set $B$.[2]

**Definition 3.3** (Simply-retargetable function). Let $\Theta$ be a set acted on by $S_d$, and let $f : \{A, B\} \times \Theta \to \mathbb{R}$. If $\exists \phi \in S_d : \forall \theta^A \in \Theta : f(B \mid \theta^A) < f(A \mid \theta^A) \implies f(A \mid \phi \cdot \theta^A) < f(B \mid \phi \cdot \theta^A)$, then $f$ is a $(\Theta, A \overset{simple}{\to} B)$-*retargetable function*.

Simple retargetability suffices for most parameter inputs to $p$ to choose Pac-Man outcome set $B$ over $A$.[3] In that case, $B$ cannot be retargeted back to $A$ because $|B| = 2 > 1 = |A|$. $p_{\max}$'s simple retargetability arises in part due to $B$ having more outcomes.

**Proposition 3.4** (Simply-retargetable functions have orbit-level tendencies).

> *If $f$ is $(\Theta, A \overset{simple}{\to} B)$-retargetable, then $f(B \mid \theta) \geq^1_{\mathrm{most:}\ \Theta} f(A \mid \theta)$.*

We now want to make even stronger claims—*how much* of each orbit incentivizes $B$ over $A$? Turner et al. [2021] asked whether the existence of multiple retargeting permutations $\phi_i$ guarantees a quantitative lower-bound on the fraction of $\theta \in \Theta$ for which $B$ is chosen. Theorem 3.6 answers "yes."

**Definition 3.5** (Multiply retargetable function). Let $\Theta$ be a subset of a set acted on by $S_d$, and let $f : \{A, B\} \times \Theta \to \mathbb{R}$.

$f$ is a $(\Theta, A \overset{n}{\to} B)$-*retargetable function* when, for each $\theta \in \Theta$, we can choose permutations $\phi_1, \ldots, \phi_n \in S_d$ which satisfy the following conditions: Consider any $\theta^A \in \mathrm{Orbit}|_{\Theta, A>B} (\theta) := \left\{\theta^* \in \mathrm{Orbit}|_\Theta (\theta) \mid f(A \mid \theta^*) > f(B \mid \theta^*)\right\}$.

---

[2]We often interpret $A$ and $B$ as probability-theoretic events, but no such structure is demanded by our results.

[3]The function's retargetability is "simple" because we are not yet worrying about *e.g.* which parameter inputs are considered plausible: Because $S_d$ acts on $\Theta$, definition 3.3 implicitly assumes $\Theta$ is closed under permutation.

1. **Retargetable via $n$ permutations.** $\forall i = 1, \ldots, n : f\left(A \mid \phi_i \cdot \theta^A\right) < f\left(B \mid \phi_i \cdot \theta^A\right)$.

2. **Parameter permutation is allowed by $\Theta$.** $\forall i : \phi_i \cdot \theta^A \in \Theta$.

3. **Permuted parameters are distinct.** $\forall i \neq j, \theta' \in \text{Orbit}|_{\Theta, A > B}(\theta) : \phi_i \cdot \theta^A \neq \phi_j \cdot \theta'$.

**Theorem 3.6** (Multiply retargetable functions have orbit-level tendencies)**.**

$$\text{If } f \text{ is } (\Theta, A \xrightarrow{n} B)\text{-retargetable, then } f(B \mid \theta) \geq^n_{\text{most: } \Theta} f(A \mid \theta).$$

*Proof outline (full proof in Appendix B).* For every $\theta^A \in \text{Orbit}|_{\Theta, A > B}(\theta)$ such that $A$ is chosen over $B$, item 1 retargets $\theta^A$ via $n$ permutations $\phi_1, \ldots, \phi_n$ such that each $\phi_i \cdot \theta^A$ makes the agent choose $B$ over $A$. These permuted parameters are valid parameter inputs by item 2. Furthermore, the $\phi_i \cdot \theta^A$ are distinct by item 3. Therefore, the cosets $\phi_i \cdot \text{Orbit}|_{\Theta, A > B}(\theta)$ are pairwise disjoint. By a counting argument, every orbit must contain at least $n$ times as many parameters choosing $B$ over $A$, than vice versa. $\qquad\square$

# 4 Decision-making tendencies in Montezuma's Revenge

To illustrate a high-dimensional setting in which parametrically retargetable decision-makers tend to seek power, we consider Montezuma's Revenge (MR), an Atari adventure game in which the player navigates deadly traps and collects treasure. The game is notoriously difficult for AI agents due to its sparse reward. MR was only recently solved [Ecoffet et al., 2021]. Figure 2 shows the starting observation $o_0$ for the first level. This section culminates with section 4.3, where we argue that increasingly powerful RL training processes will cause increasing retargetability via the reward function, which in turn causes increasingly strong decision-making tendencies.

**Terminology.** Retargetability is a property of the policy training process, and power-seeking is a property of the trained policy. More precisely, the policy training process takes as input a parameterization $\theta$ and outputs a probability distribution over policies. For each trained policy drawn from this distribution, the environment, starting state, and the drawn policy jointly specify a probability distribution over trajectories. Therefore, the training process associates each parameterization $\theta$ with the mixture distribution $P$ over trajectories (with the mixture taken over the distribution of trained policies).

A policy training process can be simply retargeted from one trajectory set $A$ to another trajectory set $B$ when there exists a permutation $\phi \in S_d$ such that, for every $\theta$ for which $P(A \mid \theta) > P(B \mid \theta)$, we have $P(A \mid \phi \cdot \theta) < P(B \mid \phi \cdot \theta)$. As in Turner et al. [2021], a trained policy $\pi$ *seeks power* when $\pi$'s actions navigate to states with high average optimal value (with the average taken over a wide range of reward functions). Generally, high-power states are able to reach a wide range of other states, and so allow bigger option sets $B$ (compared to the options $A$ available without seeking power).

## 4.1 Tendencies for initial action selection

We will be considering the actions chosen and trajectories induced by a range of decision-making procedures. For warm-up, we will explore what initial action tends to be selected by decision-makers. Let $A := \{\downarrow\}, B := \{\leftarrow, \rightarrow, \texttt{jump}, \uparrow\}$ partition the action set $\mathcal{A}$. Consider a decision-making procedure $f$ which takes as input a targeting parameter $\theta \in \Theta$, and also an initial action $a \in \mathcal{A}$, and returns the probability that $a$ is the first action. Intuitively, since $B$ contains more actions than $A$, perhaps some class of decision-making procedures tends to take an action in $B$ rather than one in $A$.

MR's initial-action situation is analogous to the Pac-Man example. In that example, if the decision-making procedure $p$ can be retargeted from terminal state set $A$ (the ghost) to set $B$ (the fruit), then $p$ tends to select a state from $B$ under most of its parameter settings $\theta$. Similarly, in MR, if the decision-making procedure $f$ can be retargeted from action set $A$ to action set $B$, then $f$ tends to take actions in $B$ for most of its parameter settings $\theta$. Consider several ways of choosing an initial action in MR.

**Random action selection.** $p_{\text{rand}} := (\{a\} \mid \theta) \mapsto \frac{1}{5}$ uniformly randomly chooses an action from $\mathcal{A}$, ignoring the parameter input. Since $\forall \theta \in \Theta : p_{\text{rand}}(B \mid \theta) = \frac{4}{5} > \frac{1}{5} = p_{\text{rand}}(A \mid \theta)$, *all* parameter inputs produce a greater chance of $B$ than of $A$, so $p_{\text{rand}}$ is (trivially) retargetable from $A$ to $B$.

**Always choosing the same action.** $p_{\text{stubborn}}$ always chooses $\downarrow$. Since $\forall \theta \in \Theta : p_{\text{stubborn}}(A \mid \theta) = 1 > 0 = p_{\text{stubborn}}(B \mid \theta)$, *all* parameter inputs produce a greater chance of $A$ than of $B$. $p_{\text{stubborn}}$ is not retargetable from $A$ to $B$.

**Greedily optimizing state-action reward.** Let $\Theta := \mathbb{R}^{\mathcal{S} \times \mathcal{A}}$ be the space of state-action reward functions. Let $p_{\text{max}}$ greedily maximize initial state-action reward, breaking ties uniformly randomly.

We now check that $p_{\text{max}}$ is retargetable from $A$ to $B$. Suppose $\theta^* \in \Theta$ is such that $p_{\text{max}}(A \mid \theta^*) > p_{\text{max}}(B \mid \theta^*)$. Then among the initial action rewards, $\theta^*$ assigns strictly maximal reward to $\downarrow$, and so $p_{\text{max}}(A \mid \theta^*) = 1$. Let $\phi$ swap the reward for the $\downarrow$ and jump actions. Then $\phi \cdot \theta^*$ assigns strictly maximal reward to jump. This means that $p_{\text{max}}(A \mid \phi \cdot \theta^*) = 0 < 1 = p_{\text{max}}(B \mid \phi \cdot \theta^*)$, satisfying definition 3.3. Then apply proposition 3.4 to conclude that $p_{\text{max}}(B \mid \theta) \geq^1_{\text{most: } \Theta} p_{\text{max}}(A \mid \theta)$.

In fact, appendix A shows that $p_{\text{max}}$ is $(\Theta, A \xrightarrow{4} B)$-retargetable (definition 3.5), and so $p_{\text{max}}(B \mid \theta) \geq^4_{\text{most: } \Theta} p_{\text{max}}(A \mid \theta)$. The reasoning is more complicated, but the rule of thumb is: When decisions are made based on the reward of outcomes, then a proportionally larger set $B$ of outcomes induces proportionally strong retargetability, which induces proportionally strong orbit-level incentives.

**Learning an exploitation policy.** Suppose we run a bandit algorithm which tries different initial actions, learns their rewards, and produces an exploitation policy which maximizes estimated reward. The algorithm uses $\epsilon$-greedy exploration and trains for $T$ trials. Given fixed $T$ and $\epsilon$, $p_{\text{bandit}}(A \mid \theta)$ returns the probability that an exploitation policy is learned which chooses an action in $A$; likewise for $p_{\text{bandit}}(B \mid \theta)$.

Here is a heuristic argument that $p_{\text{bandit}}$ is retargetable. Since the reward is deterministic, the exploitation policy will choose an optimal action if the agent has tried each action at least once, which occurs with a probability approaching 1 exponentially quickly in the number of trials $T$. Then when $T$ is large, $p_{\text{bandit}}$ approximates $p_{\text{max}}$, which is retargetable. Therefore, perhaps $p_{\text{bandit}}$ is also retargetable. A more careful analysis in appendix C.1 reveals that $p_{\text{bandit}}$ is 4-retargetable from $A$ to $B$, and so $p_{\text{bandit}}(B \mid \theta) \geq^4_{\text{most: } \Theta} p_{\text{bandit}}(A \mid \theta)$.

### 4.2 Tendencies for maximizing reward over the final observation

When evaluating the performance of an algorithm in MR, we do not focus on the agent's initial action. Rather, we focus on the longer-term consequences of the agent's actions, such as whether the agent leaves the first room. To begin reasoning about such behavior, the reader must distinguish between different kinds of retargetability.

Suppose the agent will die unless they choose action $\downarrow$ at the initial state $s_0$ (Figure 2). By section 4.1, action-retargetable decision-making procedures tend to choose actions besides $\downarrow$. On the other hand, Turner et al. [2021] showed that most reward functions make it reward-optimal to stay alive (in this situation, by choosing $\downarrow$). However, in that situations, the optimal policies are not retargetable across

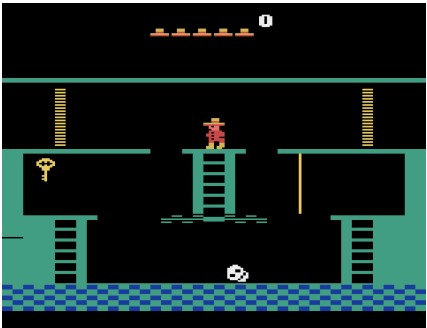

Figure 2: Montezuma's Revenge (MR) has state space $\mathcal{S}$ and observation space $\mathcal{O}$. The agent has actions $\mathcal{A} := \{\uparrow, \downarrow, \leftarrow, \rightarrow, \text{jump}\}$. At the initial state $s_0$, $\uparrow$ does nothing, $\downarrow$ descends the ladder, $\leftarrow$ and $\rightarrow$ move the agent on the platform, and jump is self-explanatory. The agent clears the temple while collecting four kinds of items: keys, swords, torches, and amulets. Under the standard environmental reward function, the agent receives points for acquiring items (such as the key on the left), opening doors, and—ultimately—completing the level.

the agent's *immediate* choice of action, but rather across future consequences (*i.e.* which room the agent ends up in).

With that in mind, we now analyze how often decision-makers leave the first room of MR.[4] Decision-making functions $\text{decide}(\theta)$ produce a probability distribution over policies $\pi \in \Pi$, which are rolled out from the initial state $s_0$ to produce observation-action trajectories $\tau = o_0 a_0 \ldots o_T a_T \ldots$, where $T$ is the rollout length we are interested in. Let $O_{T\text{-reach}}$ be the set of observations reachable starting from state $s_0$ and acting for $T$ time steps, let $O_{\text{leave}} \subseteq O_{T\text{-reach}}$ be those observations which can only be realized by leaving, and let $O_{\text{stay}} \coloneqq O_{T\text{-reach}} \setminus O_{\text{leave}}$. Consider the probability that $\text{decide}$ realizes some subset of observations $X \subseteq \mathcal{O}$ at step $T$:

$$p_{\text{decide}}(X \mid \theta) \coloneqq \mathop{\mathbb{P}}_{\substack{\pi \sim \text{decide}(\theta), \\ \tau \sim \pi \mid s_0}} (o_T \in X). \tag{3}$$

Let $\Theta \coloneqq \mathbb{R}^{\mathcal{O}}$ be the set of reward functions mapping observations $o \in \mathcal{O}$ to real numbers, and let $T \coloneqq 1{,}000$. We first consider the previous decision functions, since they are simple to analyze.

$\text{decide}_{\text{rand}}$ randomly chooses a final observation $o$ which can be realized at step 1,000, and then chooses some policy which realizes $o$.[5] $\text{decide}_{\text{rand}}$ induces an $p_{\text{rand}}$ defined by eq. (3). As before, $p_{\text{rand}}$ tends to leave the room under *all* parameter inputs.

$\text{decide}_{\text{max}}(\theta)$ produces a policy which maximizes the reward of the observation at step 1,000 of the rollout. Since MR is deterministic, we discuss *which* observation $\text{decide}_{\text{max}}(\theta)$ realizes. In a stochastic setting, the decision-maker would choose a policy realizing some probability distribution over step-$T$ observations, and the analysis would proceed similarly.

Here is the semi-formal argument for $p_{\text{max}}$'s retargetability. There are combinatorially more game-screens visible if the agent leaves the room (due to *e.g.* more point combinations, more inventory layouts, more screens outside of the first room). In other words, $|O_{\text{stay}}| \ll |O_{\text{leave}}|$. There are more ways for the selected observation to require leaving the room, than not. Thus, $p_{\text{max}}$ is extremely retargetable from $O_{\text{stay}}$ to $O_{\text{leave}}$.

Detailed analysis in section C.2 confirms that $p_{\text{max}}(O_{\text{leave}} \mid \theta) \geq^n_{\text{most: }\Theta} p_{\text{max}}(O_{\text{stay}} \mid \theta)$ for the large $n \coloneqq \lfloor \frac{|O_{\text{leave}}|}{|O_{\text{stay}}|} \rfloor$, which we show implies that $p_{\text{max}}$ tends to leave the room.

### 4.3 Tendencies for RL on featurized reward over the final observation

In the real world, we do not run $p_{\text{max}}$, which can be computed via $T$-depth exhaustive tree search in order to find and induce a maximal-reward observation $o_T$. Instead, we use reinforcement learning. Better RL algorithms seem to be more retargetable *because* of their greater capability to explore.[6]

**Exploring the first room.** Consider a featurized reward function over observations $\theta \in \mathbb{R}^{\mathcal{O}}$, which provides an end-of-episode return signal which adds a fixed reward for each item displayed in the observation (*e.g.* 5 reward for a sword, 2 reward for a key). Consider a coefficient vector $\alpha \in \mathbb{R}^4$, with each entry denoting the value of an item, and $\text{feat} : \mathcal{O} \to \mathbb{R}^4$ maps observations to feature vectors which tally the items in the agent's inventory. A reinforcement learning algorithm Alg uses this return signal to update a fixed-initialization policy network. Then $p_{\text{Alg}}(O_{\text{leave}} \mid \theta)$ returns the probability that Alg trains an policy whose step-$T$ observation required the agent to leave the initial room.

The retargetability (definition 3.3) of Alg is closely linked to the quality of Alg as an RL training procedure. For example, as explained in section C.4, Mnih et al. [2015]'s DQN isn't good enough to train policies which leave the first room of MR, and so DQN (trivially) cannot be retargetable *away* from the first room via the reward function. There isn't a single featurized reward function for which DQN visits other rooms, and so we can't have $\alpha$ such that $\phi \cdot \alpha$ retargets the agent to $O_{\text{leave}}$. DQN isn't good enough at exploring.

---

[4]In Appendix C.2, Figure 3 shows a map of the first level.

[5]$\text{decide}_{\text{rand}}$ does not act randomly at each time step, it induces a randomly selected final observation. Analogously, randomly turning a steering wheel is different from driving to a randomly chosen destination.

[6]Conversely, if the agent cannot figure out how to leave the first room, any reward signal from outside of the first room can never causally affect the learned policy. In that case, retargetability away from the first room is impossible.

More formally, in this situation, Alg is retargetable if there exists a permutation $\phi \in S_4$ such that whenever $\alpha \in \Theta := \mathbb{R}^4$ induces the learned policies to stay in the room ($p_{\text{Alg}}(O_{\text{stay}} \mid \alpha) > p_{\text{Alg}}(O_{\text{leave}} \mid \alpha)$), $\phi \cdot \alpha$ makes Alg train policies which leave the room ($p_{\text{Alg}}(O_{\text{stay}} \mid \alpha) < p_{\text{Alg}}(O_{\text{leave}} \mid \alpha)$).

**Exploring four rooms.** Suppose algorithm Alg$'$ can explore *e.g.* the first three rooms to the right of the initial room (shown in Figure 2), and consider any reward coefficient vector $\alpha \in \Theta$ which assigns unique positive weight to each item. In particular, unique positive weights rule out constant reward vectors, in which case inductive bias would produce agents which do not leave the first room.

If the agent stays in the initial room, it can induce inventory states {empty, 1key}. If the agent explores the three extra rooms, it can also induce {1sword, 1sword&1key} (see Figure 3 in Appendix C.2). Since $\alpha$ is positive, it is never optimal to finish the episode empty-handed. Therefore, if the Alg$'$ policy stays in the first room, then $\alpha$'s feature coefficients must satisfy $\alpha_{\text{key}} > \alpha_{\text{sword}}$. Otherwise, $\alpha_{\text{key}} < \alpha_{\text{sword}}$ (by assumption of unique item reward coefficients); in this case, the agent would leave and acquire the sword (since we assumed it knows how to do so). Then by switching the reward for the key and the sword, we retarget Alg$'$ to go get the sword. Alg$'$ is simply-retargetable away from the first room, *because* it can explore enough of the environment.

**Exploring the entire level.** Algorithms like GO-EXPLORE [Ecoffet et al., 2021] are probably good at exploring even given sparse featurized reward. Therefore, GO-EXPLORE is even more retargetable in this setting, because it is more able to explore and discover the breadth of options (final inventory counts) available to it, and remember how to navigate to them. Furthermore, sufficiently powerful planning algorithms should likewise be retargetable in a similar way, insofar as they can reliably find high-scoring item configurations.

We speculate that increasingly "impressive" algorithms (whether RL training or planning) are often more impressive because they can allow retargeting the agent's final behavior from one kind of outcome, to another. Just as GO-EXPLORE seems highly retargetable while DQN does not, we expect increasingly impressive algorithms to be increasingly retargetable—whether over actions in a bandit problem, or over the final observation in an RL episode.

## 5 Retargetability can imply power-seeking tendencies

### 5.1 Generalizing the power-seeking theorems for Markov decision processes

Turner et al. [2021] considered finite MDPs in which decision-makers took as input a reward function over states ($\mathbf{r} \in \mathbb{R}^{|\mathcal{S}|}$) and selected an optimal policy for that reward function. They considered the state visit distributions $\mathbf{f} \in \mathcal{F}(s)$, which basically correspond to the trajectories which the agent could induce starting from state $s$. For $F \subseteq \mathcal{F}(s)$, $p_{\max}(F \mid \mathbf{r})$ returns 1 if an element of $F$ is optimal for reward function $\mathbf{r}$, and 0 otherwise. They showed situations where a larger set of distributions $F_{\text{large}}$ tended to be optimal over a smaller set: $p_{\max}(F_{\text{large}} \mid \mathbf{r}) \geq^1_{\text{most: } \mathbb{R}^{|\mathcal{S}|}} p_{\max}(F_{\text{small}} \mid \mathbf{r})$. For example, in Pac-Man, most reward functions make it optimal to stay alive for at least one time step: $p_{\max}(F_{\text{survival}} \mid \mathbf{r}) \geq^1_{\text{most: } \mathbb{R}^{|\mathcal{S}|}} p_{\max}(F_{\text{instant death}} \mid \mathbf{r})$. Turner et al. [2021] showed that optimal policies tend to seek power by keeping options open and staying alive. Appendix D provides a quantitative generalization of Turner et al. [2021]'s results on optimal policies.

Throughout this paper, we abstracted their arguments away from finite MDPs and reward-optimal decision-making. Instead, parametrically retargetable decision-makers tend to seek power: Proposition A.11 shows that a wide range of decision-making procedures are retargetable over outcomes, and theorem A.13 demonstrates the retargetability of *any* decision-making which is determined by the expected utility of outcomes. In particular, these results apply straightforwardly to MDPs.

### 5.2 Better RL algorithms tend to be more retargetable

Reinforcement learning algorithms are practically useful insofar as they can train an agent to accomplish some task (*e.g.* cleaning a room). A good RL algorithm is relatively task-agnostic (*e.g.* is not restricted to only training policies which clean rooms). Task-agnosticism suggests retargetability across desired future outcomes / task completions.

In MR, suppose we instead give the agent $1$ reward for the initial state, and $0$ otherwise. Any reasonable reinforcement learning procedure will just learn to stay put (which is the optimal policy). However, consider whether we can retarget the agent's policy to beat the game, by swapping the initial state reward with the end-game state reward. Most present-day RL algorithms are not good enough to solve such a sparse game, and so are not retargetable in this sense. But an agent which did enough exploration would also learn a good policy for the permuted reward function. Such an effective training regime could be useful for solving real-world tasks. Many researchers aim to develop effective training regimes.

Our results suggest that once RL capabilities reach a certain level, trained agents will tend to seek power in the real world. Presently, it is not dangerous to train an agent to complete a task—such an agent will not be able to complete its task by staying activated against the designers' wishes. The present lack of danger is not because optimal policies do not have self-preservation tendencies—they do [Turner et al., 2021]. Rather, the lack of danger reflects the fact that present-day RL agents cannot learn such complex action sequences *at all*. Just as the Montezuma's Revenge agent had to be sufficiently competent to be retargetable from initial-state reward to game-complete reward, real-world agents have to be sufficiently intelligent in order to be retargetable from outcomes which don't require power-seeking, to those which do require power-seeking.

Here is some speculation. After training an RL agent to a high level of capability, the agent may be optimizing internally represented goals over its model of the environment [Hubinger et al., 2019]. Furthermore, we think that different reward parameter settings would train different internal goals into the agent. To make an analogy, changing a person's reward circuitry would presumably reinforce them for different kinds of activities and thereby change their priorities. In this sense, trained real-world agents may be retargetable towards power-requiring outcomes via the reward function parameter setting. Insofar as this speculation holds, our theory predicts that advanced reinforcement learning at scale will—for most settings of the reward function—train policies which tend to seek power.

# 6 Discussion

In section 3, we formalized a notion of parametric retargetability and stated several key results. While our results are broadly applicable, further work is required to understand the implications for AI.

## 6.1 Prior work

In this work, we do not motivate the risks from AI power-seeking. We refer the reader to *e.g.* Carlsmith [2021]. As explained in section 5.1, Turner et al. [2021] show that, given certain environmental symmetries in an MDP, the optimal-policy-producing algorithm $f$(state visitation distribution set, state-based reward function) is 1-retargetable via the reward function, from smaller to larger sets of environmental options. Section A shows that optimality is not required, and instead a wide range of decision-making procedures satisfy the retargetability criterion. Furthermore, we generalize from 1-retargetability to $n$-fold-retargetability whenever option set $B$ contains "$n$ copies" of set $A$ (definition A.7 in section A).

## 6.2 Future work and limitations

We currently have analyzed planning- and reinforcement learning-based settings. However, results such as theorem 3.6 might in some way apply to the training of other machine learning networks. Furthermore, while theorem 3.6 does not assume a finite environment, we currently do not see how to apply that result to *e.g.* infinite-state partially observable Markov decision processes.

Section 4 semi-formally analyzes decision-making incentives in the MR video game, leaving the proofs to section C. However, these proofs are several pages long. Perhaps additional lemmas can allow quick proof of orbit-level incentives in situations relevant to real-world decision-makers.

Consider a sequence of decision-making functions $p_t : \{A, B\} \times \Theta \to \mathbb{R}$ which converges pointwise to some $p$ such that $p(B \mid \theta) \geq^n_{\text{most: } \Theta} p(A \mid \theta)$. We expect that under rather mild conditions, $\exists T : \forall t \geq T : p_t(B \mid \theta) \geq^n_{\text{most: } \Theta} p_t(A \mid \theta)$. As a corollary, for any decision-making procedure $p_t$ which runs for $t$ time steps and satisfies $\lim_{t \to \infty} p_t = p$, the function $p_t$ will have decision-making incentives after finite time. For example, value iteration (VI) eventually finds an optimal policy

[Puterman, 2014], and optimal policies tend to seek power [Turner et al., 2021]. Therefore, this conjecture would imply that if VI is run for some long but finite time, it tends to produce power-seeking policies. More interestingly, the result would allow us to reason about the effect of *e.g.* randomly initializing parameters (in VI, the tabular value function at $t = 0$). The effect of random initialization washes out in the limit of infinite time, so we would still conclude the presence of finite-time power-seeking incentives.

Our results do not *prove* that we will build unaligned AI agents which seek power over the world. Here are a few situations in which our results are not concerning or not applicable.

1. The AI is aligned with human interests. For example, we want a robotic cartographer to prevent itself from being deactivated. However, the AI alignment problem is not yet understood for highly intelligent agents [Russell, 2019].

2. The AI's decision-making is not retargetable (definition 3.5).

3. The AI's decision-making is retargetable over *e.g.* actions (section 4.1) instead of over final outcomes (section 4.2). This retargetability seems less concerning, but also less practically useful.

### 6.3 Conclusion

We introduced the concept of retargetability and showed that retargetable decision-makers often make similar instrumental choices. We applied these results in the Montezuma's Revenge (MR) video game, showing how increasingly advanced reinforcement learning algorithms correspond to increasingly retargetable agent decision-making. Increasingly retargetable agents make increasingly similar instrumental decisions—*e.g.* leaving the initial room in MR, or staying alive in Pac-Man. In particular, these decisions will often correspond to gaining power and keeping options open [Turner et al., 2021]. Our theory suggests that when RL training processes become sufficiently advanced, the trained agents will tend to seek power over the world. This theory suggests a safety risk. We hope for future work on this theory so that the field of AI can understand the relevant safety risks *before* the field trains power-seeking agents.

### Broader impacts

Our theory of orbit-level tendencies constitutes basic mathematical research into the decision-making tendencies of certain kinds of agents. We hope that this theory will prevent negative impacts from unaligned power-seeking AI. We do not anticipate that our work will have negative impact.

### Acknowledgements

We thank Irene Tematelewo, Colin Shea-Blymyer, and our anonymous reviewers for feedback. We thank Justis Mills for proofreading.

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
