## Appendix A  Retargetability over outcome lotteries

Suppose we are interested in $d$ outcomes. Each outcome could be the visitation of an MDP state, or a trajectory, or the receipt of a physical item. In the Pac-Man example of section 2, $d = 3$ states. The agent can induce each outcome with probability 1, so let $\mathbf{e}_o \in \mathbb{R}^3$ be the standard basis vector with probability 1 on outcome $o$ and 0 elsewhere. Then the agent chooses among outcome lotteries $C := \{\mathbf{e}_{\text{👾}}, \mathbf{e}_{\text{🍒}}, \mathbf{e}_{\text{🍒}}\}$, which we partition into $A := \{\mathbf{e}_{\text{👾}}\}$ and $B := \{\mathbf{e}_{\text{🍒}}, \mathbf{e}_{\text{🍒}}\}$.

**Definition A.1** (Outcome lotteries). A unit vector $\mathbf{x} \in \mathbb{R}^d$ with non-negative entries is an *outcome lottery*.[7]

Many decisions are made consequentially: based on the consequences of the decision, on what outcomes are brought about by an act. For example, in a deterministic Atari game, a policy induces a trajectory. A reward function and discount rate tuple $(R, \gamma)$ assigns a *return* to each state trajectory $\tau = s_0, s_1, \ldots$: $G(\tau) = \sum_{i=0}^{\infty} \gamma^i R(s_i)$. The relevant outcome lottery is the discounted visit distribution over future states in an Atari game, and policies are optimal or not depending on which outcome lottery is induced by the policy.

**Definition A.2** (Optimality indicator function). Let $X, C \subsetneq \mathbb{R}^d$ be finite, and let $\mathbf{u} \in \mathbb{R}^d$. IsOptimal $(X \mid C, \mathbf{u})$ returns 1 if $\max_{\mathbf{x} \in X} \mathbf{x}^\top \mathbf{u} \geq \max_{\mathbf{c} \in C} \mathbf{c}^\top \mathbf{u}$, and 0 otherwise.

We consider decision-making procedures which take in a targeting parameter $\mathbf{u}$. For example, the column headers of Table 2a show the 6 permutations of the utility function $u(\text{👾}) := 10, u(\text{🍒}) := 5, u(\text{🍒}) := 0$, representable as a vector $\mathbf{u} \in \mathbb{R}^3$.

$\mathbf{u}$ can be permuted as follows. The outcome permutation $\phi \in S_d$ inducing an $d \times d$ permutation matrix $\mathbf{P}_\phi$ in row representation: $(\mathbf{P}_\phi)_{ij} = 1$ if $i = \phi(j)$ and 0 otherwise. Table 2a shows that for a given utility function, $\frac{2}{3}$ of its orbit agrees that $B$ is strictly optimal over $A$.

Orbit-level incentives occur when an inequality holds for most permuted parameter choices $\mathbf{u}'$. Table 2a demonstrates an application of Turner et al. [2021]'s results: Optimal decision-making induces orbit-level incentives for choosing Pac-Man outcomes in $B$ over outcomes in $A$.

Furthermore, Turner et al. [2021] conjectured that "larger" $B$ will imply stronger orbit-level tendencies: If going right leads to 500 times as many options as going left, then right is better than left for at least 500 times as many reward functions for which the opposite is true. We prove this conjecture with theorem D.11 in appendix D.

However, orbit-level incentives do not require optimality. One clue is that the same results hold for anti-optimal agents, since anti-optimality/utility minimization of $\mathbf{u}$ is equivalent to maximizing $-\mathbf{u}$. Table 2b illustrates that the same orbit guarantees hold in this case.

**Definition A.3** (Anti-optimality indicator function). Let $X, C \subsetneq \mathbb{R}^d$ be finite, and let $\mathbf{u} \in \mathbb{R}^d$. AntiOpt $(X \mid C, \mathbf{u})$ returns 1 if $\min_{\mathbf{x} \in X} \mathbf{x}^\top \mathbf{u} \leq \min_{\mathbf{c} \in C} \mathbf{c}^\top \mathbf{u}$, and 0 otherwise.

Stepping beyond expected utility maximization/minimization, Boltzmann-rational decision-making selects outcome lotteries proportional to the exponential of their expected utility.

**Definition A.4** (Boltzmann rationality [Baker et al., 2007]). For $X \subseteq C$ and temperature $T > 0$, let

$$\text{Boltzmann}_T\left(X \mid C, \mathbf{u}\right) := \frac{\sum_{\mathbf{x} \in X} e^{T^{-1} \mathbf{x}^\top \mathbf{u}}}{\sum_{\mathbf{c} \in C} e^{T^{-1} \mathbf{c}^\top \mathbf{u}}}$$

be the probability that some element of $X$ is Boltzmann-rational.

Lastly, orbit-level tendencies occur even under decision-making procedures which partially ignore expected utility and which "don't optimize too hard." Satisficing agents randomly choose an outcome lottery with expected utility exceeding some threshold. Table 2d demonstrates that satisficing induces orbit-level tendencies.

**Definition A.5** (Satisficing). Let $t \in \mathbb{R}$, let $X \subseteq C \subsetneq \mathbb{R}^d$ be finite. $\text{Satisfice}_t\left(X, C \mid \mathbf{u}\right) := \frac{\left| X \cap \{\mathbf{c} \in C \mid \mathbf{c}^\top \mathbf{u} \geq t\} \right|}{\left| \{\mathbf{c} \in C \mid \mathbf{c}^\top \mathbf{u} \geq t\} \right|}$ is the fraction of $X$ whose value exceeds threshold $t$. $\text{Satisfice}_t\left(X, C \mid \mathbf{u}\right)$ evaluates to 0 the denominator equals 0.

---

[7]Our results on outcome lotteries hold for generic $\mathbf{x}' \in \mathbb{R}^d$, but we find it conceptually helpful to consider the non-negative unit vector case.

Table 2: Orbit-level incentives across 4 decision-making functions.

| Utility function $\mathbf{u}'$ | 10,5,0 | 10,0,5 | 5,10,0 | 5,0,10 | 0,10,5 | 0,5,10 |
|---|---|---|---|---|---|---|
| $\text{IsOptimal}\left(\{\mathbf{e}_{👾},\mathbf{e}_{🍎}\} \mid C, \mathbf{u}'\right)$ | 1 | 1 | 1 | 0 | 1 | 0 |
| $\text{IsOptimal}\left(\{\mathbf{e}_{🍒}\} \mid C, \mathbf{u}'\right)$ | 0 | 0 | 0 | 1 | 0 | 1 |

(a) Dark gray columns indicate utility function permutations $\mathbf{u}'$ for which $\text{IsOptimal}\left(B \mid C, \mathbf{u}'\right) > \text{IsOptimal}\left(A \mid C, \mathbf{u}'\right)$, while white indicates that the opposite strict inequality holds.

| Utility function $\mathbf{u}'$ | 10,5,0 | 10,0,5 | 5,10,0 | 5,0,10 | 0,10,5 | 0,5,10 |
|---|---|---|---|---|---|---|
| $\text{AntiOpt}\left(\{\mathbf{e}_{👾},\mathbf{e}_{🍎}\} \mid C, \mathbf{u}'\right)$ | 0 | 1 | 0 | 1 | 1 | 1 |
| $\text{AntiOpt}\left(\{\mathbf{e}_{🍒}\} \mid C, \mathbf{u}'\right)$ | 1 | 0 | 1 | 0 | 0 | 0 |

(b) Utility-minimizing outcome selection probability.

| Utility function $\mathbf{u}'$ | 10,5,0 | 10,0,5 | 5,10,0 | 5,0,10 | 0,10,5 | 0,5,10 |
|---|---|---|---|---|---|---|
| $\text{Boltzmann}_1\left(\{\mathbf{e}_{👾},\mathbf{e}_{🍎}\} \mid C, \mathbf{u}'\right)$ | 1 | .993 | 1 | .007 | .993 | .007 |
| $\text{Boltzmann}_1\left(\{\mathbf{e}_{🍒}\} \mid C, \mathbf{u}'\right)$ | .000 | .007 | .000 | .993 | .007 | .993 |

(c) Boltzmann selection probabilities for $T = 1$, rounded to three significant digits.

| Utility function $\mathbf{u}'$ | 10,5,0 | 10,0,5 | 5,10,0 | 5,0,10 | 0,10,5 | 0,5,10 |
|---|---|---|---|---|---|---|
| $\text{Satisfice}_3\left(\{\mathbf{e}_{👾},\mathbf{e}_{🍎}\} \mid C, \mathbf{u}'\right)$ | 1 | .5 | 1 | .5 | .5 | .5 |
| $\text{Satisfice}_3\left(\{\mathbf{e}_{🍒}\} \mid C, \mathbf{u}'\right)$ | 0 | .5 | 0 | .5 | .5 | .5 |

(d) A satisficer uniformly randomly selects an outcome lottery with expected utility greater than or equal to the threshold $t$. Here, $t = 3$. When $\text{Satisfice}_3\left(\{\mathbf{e}_{👾},\mathbf{e}_{🍎}\} \mid C, \mathbf{u}'\right) = \text{Satisfice}_3\left(\{\mathbf{e}_{🍒}\} \mid C, \mathbf{u}'\right)$, the column is colored medium gray.

For each table, two-thirds of the utility permutations (columns) assign strictly larger values (shaded dark gray) to an element of $B := \{\mathbf{e}_{🍎}, \mathbf{e}_{🍒}\}$ than to an element of $A := \{\mathbf{e}_{👾}\}$. For optimal, anti-optimal, Boltzmann-rational, and satisficing agents, proposition A.11 proves that these tendencies hold for all targeting parameter orbits.

## A.1 A range of decision-making functions are retargetable

In MDPs, Turner et al. [2021] consider *state visitation distributions* which record the total discounted time steps spent in each environment state, given that the agent follows some policy $\pi$ from an initial state $s$. These visitation distributions are one kind of outcome lottery, with $d = |\mathcal{S}|$ the number of MDP states.

In general, we suppose the agent has an objective function $\mathbf{u} \in \mathbb{R}^d$ which maps outcomes to real numbers. In Turner et al. [2021], $\mathbf{u}$ was a state-based reward function (and so the outcomes were *states*). However, we need not restrict ourselves to the MDP setting.

To state our key results, we define several technical concepts which we informally used when reasoning about $A := \{\mathbf{e}_{👾}\}$ and $B := \{\mathbf{e}_{🍎}, \mathbf{e}_{🍒}\}$.

**Definition A.6** (Similarity of vector sets)**.** For $\phi \in S_d$ and $X \subseteq \mathbb{R}^d$, $\phi \cdot X \coloneqq \{\mathbf{P}_\phi \mathbf{x} \mid \mathbf{x} \in X\}$. $X' \subseteq \mathbb{R}^{|\mathcal{S}|}$ *is similar to* $X$ when $\exists \phi : \phi \cdot X' = X$. $\phi$ is an *involution* if $\phi = \phi^{-1}$ (it either transposes states, or fixes them). $X$ *contains a copy of* $X'$ when $X'$ is similar to a subset of $X$ via an involution $\phi$.

**Definition A.7** (Containment of set copies)**.** Let $n$ be a positive integer, and let $A, B \subseteq \mathbb{R}^d$. We say that $B$ *contains* $n$ *copies of* $A$ when there exist involutions $\phi_1, \ldots, \phi_n \in S_d$ such that $\forall i : \phi_i \cdot A =: B_i \subseteq B$ and $\forall j \neq i : \phi_i \cdot B_j = B_j$.[8]

$B \coloneqq \{\mathbf{e}_{🍒}, \mathbf{e}_{🍓}\}$ contains two copies of $A \coloneqq \{\mathbf{e}_{🍎}\}$ via $\phi_1 \coloneqq 🍎 \leftrightarrow 🍒$ and $\phi_2 \coloneqq 🍎 \leftrightarrow 🍓$.

**Definition A.8** (Targeting parameter distribution assumptions)**.** Results with $\mathcal{D}_{\text{any}}$ hold for any probability distribution over $\mathbb{R}^d$. Let $\mathfrak{D}_{\text{any}} \coloneqq \Delta(\mathbb{R}^d)$. For a function $f : \mathbb{R}^d \mapsto \mathbb{R}$, we write $f(\mathcal{D}_{\text{any}})$ as shorthand for $\mathbb{E}_{\mathbf{u} \sim \mathcal{D}_{\text{any}}}\big[f(\mathbf{u})\big]$.

The symmetry group on $d$ elements, $S_d$, acts on the set of probability distributions over $\mathbb{R}^d$.

**Definition A.9** (Pushforward distribution of a permutation [Turner et al., 2021])**.** Let $\phi \in S_d$. $\phi \cdot \mathcal{D}_{\text{any}}$ is the pushforward distribution induced by applying the random vector $p(\mathbf{u}) \coloneqq \mathbf{P}_\phi \mathbf{u}$ to $\mathcal{D}_{\text{any}}$.

**Definition A.10** (Orbit of a probability distribution [Turner et al., 2021])**.** The *orbit* of $\mathcal{D}_{\text{any}}$ under the symmetric group $S_d$ is $S_d \cdot \mathcal{D}_{\text{any}} \coloneqq \{\phi \cdot \mathcal{D}_{\text{any}} \mid \phi \in S_d\}$.

Because $B$ contains 2 copies of $A$, there are "at least two times as many ways" for $B$ to be optimal, than for $A$ to be optimal. Similarly, $B$ is "at least two times as likely" to contain an anti-rational outcome lottery for generic utility functions. As demonstrated by Table 2, the key idea is that "larger" sets (a set $B$ containing several *copies* of set $A$) are more likely to be chosen under a wide range of decision-making criteria.

**Proposition A.11** (Orbit incentives for different rationalities)**.** *Let* $A, B \subseteq C \subsetneq \mathbb{R}^d$ *be finite, such that* $B$ *contains* $n$ *copies of* $A$ *via involutions* $\phi_i$ *such that* $\phi_i \cdot C = C$.

1. ***Rational choice [Turner et al., 2021].***

$$\text{IsOptimal}\big(B \mid C, \mathcal{D}_{any}\big) \geq^n_{\text{most: } \mathfrak{D}_{any}} \text{IsOptimal}\big(A \mid C, \mathcal{D}_{any}\big).$$

2. ***Uniformly randomly choosing an optimal lottery.*** *For* $X \subseteq C$*, let*

$$\text{FracOptimal}\big(X \mid C, \mathbf{u}\big) \coloneqq \frac{\left|\left\{\arg\max_{\mathbf{c} \in C} \mathbf{c}^\top \mathbf{u}\right\} \cap X\right|}{\left|\left\{\arg\max_{\mathbf{c} \in C} \mathbf{c}^\top \mathbf{u}\right\}\right|}.$$

*Then* $\text{FracOptimal}\big(B \mid C, \mathcal{D}_{any}\big) \geq^n_{\text{most: } \mathfrak{D}_{any}} \text{FracOptimal}\big(A \mid C, \mathcal{D}_{any}\big)$.

3. ***Anti-rational choice.*** $\text{AntiOpt}\big(B \mid C, \mathcal{D}_{any}\big) \geq^n_{\text{most: } \mathfrak{D}_{any}} \text{AntiOpt}\big(A \mid C, \mathcal{D}_{any}\big)$.

4. ***Boltzmann rationality.***

$$\text{Boltzmann}_T\big(B \mid C, \mathcal{D}_{any}\big) \geq^n_{\text{most: } \mathfrak{D}_{any}} \text{Boltzmann}_T\big(A \mid C, \mathcal{D}_{any}\big).$$

5. ***Uniformly randomly drawing*** $k$ ***outcome lotteries and choosing the best.*** *For* $X \subseteq C$*,* $\mathbf{u} \in \mathbb{R}^d$*, and* $k \geq 1$*, let*

$$\textit{best-of-}k(X, C \mid \mathbf{u}) \coloneqq \mathop{\mathbb{E}}_{\mathbf{a}_1, \ldots, \mathbf{a}_k \sim \textit{unif}(C)}\Big[\text{FracOptimal}\big(X \cap \{\mathbf{a}_1, \ldots, \mathbf{a}_k\} \mid \{\mathbf{a}_1, \ldots, \mathbf{a}_k\}, \mathbf{u}\big)\Big].$$

*Then* $\textit{best-of-}k(B \mid C, \mathcal{D}_{any}) \geq^n_{\text{most: } \mathfrak{D}_{any}} \textit{best-of-}k(A \mid C, \mathcal{D}_{any})$.

---

[8]Technically, definition A.7 implies that $A$ contains $n$ copies of $A$ holds for all $n$, via $n$ applications of the identity permutation. For our purposes, this provides greater generality, as all of the relevant results still hold. Enforcing pairwise disjointness of the $B_i$ would handle these issues, but would narrow our results to not apply *e.g.* when the $B_i$ share a constant vector.

6. ***Satisficing [Simon, 1956].*** $\text{Satisfice}_t \left( B \mid C, \mathcal{D}_{any} \right) \geq^n_{\text{most: } \mathfrak{D}_{any}} \text{Satisfice}_t \left( A \mid C, \mathcal{D}_{any} \right)$.

7. ***Quantilizing over outcome lotteries [Taylor, 2016].*** *Let $P$ be the uniform probability distribution over $C$. For $X \subseteq C$, $\mathbf{u} \in \mathbb{R}^d$, and $q \in (0, 1]$, let $Q_{q,P}(X \mid C, \mathbf{u})$ (definition B.12) return the probability that an outcome lottery in $X$ is drawn from the top $q$-quantile of $P$, sorted by expected utility under $\mathbf{u}$. Then $Q_{q,P}(B \mid C, \mathbf{u}) \geq^n_{\text{most: } \mathbb{R}^d} Q_{q,P}(A \mid C, \mathbf{u})$.*

One retargetable class of decision-making functions are those which only account for the expected utilities of available choices.

**Definition A.12** (EU-determined functions)**.** Let $\mathcal{P}\left(\mathbb{R}^d\right)$ be the power set of $\mathbb{R}^d$, and let $f : \prod_{i=1}^m \mathcal{P}\left(\mathbb{R}^d\right) \times \mathbb{R}^d \to \mathbb{R}$. $f$ is an *EU-determined function* if there exists a family of functions $\{g^{\omega_1, \dots, \omega_m}\}$ such that

$$f(X_1, \dots, X_m \mid \mathbf{u}) = g^{|X_1|, \dots, |X_m|} \left( \left[ \mathbf{x}_1^\top \mathbf{u} \right]_{\mathbf{x}_1 \in X_1}, \dots, \left[ \mathbf{x}_m^\top \mathbf{u} \right]_{\mathbf{x}_m \in X_m} \right), \qquad (4)$$

where $[r_i]$ is the multiset of its elements $r_i$.

For example, let $X \subseteq C \subsetneq \mathbb{R}^d$ be finite, and consider utility function $\mathbf{u} \in \mathbb{R}^d$. A Boltzmann-rational agent is more likely to select outcome lotteries with greater expected utility. Formally, $\text{Boltzmann}_T \left( X \mid C, \mathbf{u} \right) \coloneqq \sum_{\mathbf{x} \in X} \frac{e^{T \cdot \mathbf{x}^\top \mathbf{u}}}{\sum_{\mathbf{c} \in C} e^{T \cdot \mathbf{c}^\top \mathbf{u}}}$ depends only on the expected utility of outcome lotteries in $X$, relative to the expected utility of all outcome lotteries in $C$. Therefore, $\text{Boltzmann}_T$ is a function of expected utilities. This is *why* $\text{Boltzmann}_T$ satisfies the $\geq^n_{\text{most: } \mathfrak{D}_{any}}$ relation.

**Theorem A.13** (Orbit tendencies occur for EU-determined decision-making functions)**.** *Let $A, B, C \subseteq \mathbb{R}^d$ be such that $B$ contains $n$ copies of $A$ via $\phi_i$ such that $\phi_i \cdot C = C$. Let $h : \prod_{i=1}^2 \mathcal{P}\left(\mathbb{R}^d\right) \times \mathbb{R}^d \to \mathbb{R}$ be an EU-determined function, and let $p(X \mid \mathbf{u}) \coloneqq h(X, C \mid \mathbf{u})$. Suppose that $p$ returns a probability of selecting an element of $X$ from $C$. Then $p(B \mid \mathbf{u}) \geq^n_{\text{most: } \mathbb{R}^d} p(A \mid \mathbf{u})$.*

The key takeaway is that decisions which are determined by expected utility are straightforwardly retargetable. By changing the targeting parameter hyperparameter, the decision-making procedure can be flexibly retargeted to choose elements of "larger" sets (in terms of set copies via definition A.7). Less abstractly, for many agent rationalities—ways of making decisions over outcome lotteries—it is generally the case that larger sets will more often be chosen over smaller sets.

For example, consider a Pac-Man playing agent choosing which environmental state cycle it should end up in. Turner et al. [2021] show that for most reward functions, average-reward maximizing agents will tend to stay alive so that they can reach a wider range of environmental cycles. However, our results show that average-reward *minimizing* agents also exhibit this tendency, as do Boltzmann-rational agents who assign greater probability to higher-reward cycles. Any EU-based cycle selection method will—for most reward functions—tend to choose cycles which require Pac-Man to stay alive (at first).

## Appendix B  Theoretical results

**Definition 3.2** (Inequalities which hold for most orbit elements)**.** Suppose $\Theta$ is a subset of a set acted on by $S_d$, the symmetric group on $d$ elements. Let $f : \{A, B\} \times \Theta \to \mathbb{R}$ and let $n \geq 1$. We write $f(B \mid \theta) \geq^n_{\text{most: } \Theta} f(A \mid \theta)$ when, for *all* $\theta \in \Theta$, the following cardinality inequality holds:

$$\left| \{\theta' \in \text{Orbit}|_\Theta (\theta) \mid f(B \mid \theta') > f(A \mid \theta')\} \right| \geq n \left| \{\theta' \in \text{Orbit}|_\Theta (\theta) \mid f(B \mid \theta') < f(A \mid \theta')\} \right|. \tag{2}$$

**Remark.** In stating their equivalent of definition 3.2, Turner et al. [2021] define two functions $f_1(\theta) \coloneqq f(B \mid \theta)$ and $f_2(\theta) \coloneqq f(A \mid \theta)$ (both having type signature $f_i : \Theta \to \mathbb{R}$). For compatibility, proofs also use this notation.

**Lemma B.1** (Limited transitivity of $\geq_{\text{most}}$)**.** *Let $f_0, f_1, f_2, f_3 : \Theta \to \mathbb{R}$, and suppose $\Theta$ is a subset of a set acted on by $S_d$. Suppose that $f_1(\theta) \geq^n_{\text{most: } \Theta} f_2(\theta)$ and $\forall \theta \in \Theta : f_0(\theta) \geq f_1(\theta)$ and $f_2(\theta) \geq f_3(\theta)$. Then $f_0(\theta) \geq^n_{\text{most: } \Theta} f_3(\theta)$.*

*Proof.* Let $\theta \in \Theta$ and let $\mathrm{Orbit}|_{\Theta, f_a > f_b} (\theta) := \{\theta' \in \mathrm{Orbit}|_\Theta (\theta) \mid f_a(\theta') > f_b(\theta')\}$.

$$\left| \mathrm{Orbit}|_{\Theta, f_0 > f_3} (\theta) \right| \geq \left| \mathrm{Orbit}|_{\Theta, f_1 > f_2} (\theta) \right| \tag{5}$$
$$\geq n \left| \mathrm{Orbit}|_{\Theta, f_2 > f_1} (\theta) \right| \tag{6}$$
$$\geq n \left| \mathrm{Orbit}|_{\Theta, f_3 > f_0} (\theta) \right|. \tag{7}$$

For all $\theta' \in \mathrm{Orbit}|_{\Theta, f_1 > f_2} (\theta)$,

$$f_0(\theta') \geq f_1(\theta') > f_2(\theta') \geq f_3(\theta')$$

by assumption, and so

$$\mathrm{Orbit}|_{\Theta, f_1 > f_2} (\theta) \subseteq \mathrm{Orbit}|_{\Theta, f_0 > f_3} (\theta).$$

Therefore, eq. (5) follows. By assumption,

$$\left| \mathrm{Orbit}|_{\Theta, f_1 > f_2} (\theta) \right| \geq n \left| \mathrm{Orbit}|_{\Theta, f_2 > f_1} (\theta) \right|;$$

eq. (6) follows. For all $\theta' \in \mathrm{Orbit}|_{\Theta, f_2 > f_1} (\theta)$, our assumptions on $f_0$ and $f_3$ ensure that

$$f_0(\theta') \leq f_1(\theta') < f_3(\theta') \leq f_2(\theta'),$$

so

$$\mathrm{Orbit}|_{\Theta, f_3 > f_0} (\theta) \subseteq \mathrm{Orbit}|_{\Theta, f_2 > f_1} (\theta).$$

Then eq. (7) follows. By eq. (7), $f_0(\theta) \geq^n_{\mathrm{most:}\ \Theta} f_3(\theta)$. $\qquad \square$

**Lemma B.2** (Order inversion for $\geq_{\mathrm{most}}$). *Let $f_1, f_2 : \Theta \to \mathbb{R}$, and suppose $\Theta$ is a subset of a set acted on by $S_d$. Suppose that $f_1(\theta) \geq^n_{\mathrm{most:}\ \Theta} f_2(\theta)$. Then $-f_2(\theta) \geq^n_{\mathrm{most:}\ \Theta} -f_1(\theta)$.*

*Proof.* By definition A.10, $f_1(\theta) \geq^n_{\mathrm{most:}\ \Theta} f_2(\theta)$ means that

$$\left| \{\theta' \in \mathrm{Orbit}|_\Theta (\theta) \mid f_1(\theta') > f_2(\theta')\} \right| \geq n \left| \{\theta' \in \mathrm{Orbit}|_\Theta (\theta) \mid f_1(\theta') < f_2(\theta')\} \right| \tag{8}$$

$$\left| \{\theta' \in \mathrm{Orbit}|_\Theta (\theta) \mid -f_2(\theta') > -f_1(\theta')\} \right| \geq n \left| \{\theta' \in \mathrm{Orbit}|_\Theta (\theta) \mid -f_2(\theta') < -f_1(\theta')\} \right|. \tag{9}$$

Then $-f_2(\theta) \geq^n_{\mathrm{most:}\ \Theta} -f_1(\theta)$. $\qquad \square$

**Remark.** Lemma B.3 generalizes Turner et al. [2021]'s lemma B.2.

**Lemma B.3** (Orbital fraction which agrees on (weak) inequality). *Suppose $f_1, f_2 : \Theta \to \mathbb{R}$ are such that $f_1(\theta) \geq^n_{\mathrm{most:}\ \Theta} f_2(\theta)$. Then for all $\theta \in \Theta$, $\dfrac{\left| \{\theta' \in (S_d \cdot \theta) \cap \Theta \mid f_1(\theta') \geq f_2(\theta')\} \right|}{\left| (S_d \cdot \theta) \cap \Theta \right|} \geq \dfrac{n}{n+1}$.*

*Proof.* All $\theta' \in (S_d \cdot \theta) \cap \Theta$ such that $f_1(\theta') = f_2(\theta')$ satisfy $f_1(\theta') \geq f_2(\theta')$. Otherwise, consider the $\theta' \in (S_d \cdot \theta) \cap \Theta$ such that $f_1(\theta') \neq f_2(\theta')$. By assumption, at least $\frac{n}{n+1}$ of these $\theta'$ satisfy $f_1(\theta') > f_2(\theta')$, in which case $f_1(\theta') \geq f_2(\theta')$. Then the desired inequality follows. $\qquad \square$

## B.1 General results on retargetable functions

**Definition B.4** (Functions which are increasing under joint permutation). Suppose that $S_d$ acts on sets $\mathbf{E}_1, \ldots, \mathbf{E}_m$, and let $f : \prod_{i=1}^m \mathbf{E}_i \to \mathbb{R}$. $f(X_1, \ldots, X_m)$ is *increasing under joint permutation by $P \subseteq S_d$* when $\forall \phi \in P : f(X_1, \ldots, X_m) \leq f(\phi \cdot X_1, \ldots, \phi \cdot X_m)$. If equality always holds, then $f(X_1, \ldots, X_m)$ is *invariant under joint permutation by $P$*.

**Lemma B.5** (Expectations of joint-permutation-increasing functions are also joint-permutation-increasing). *For $\mathbf{E}$ which is a subset of a set acted on by $S_d$, let $f : \mathbf{E} \times \mathbb{R}^d \to \mathbb{R}$ be a bounded function which is measurable on its second argument, and let $P \subseteq S_d$. Then if $f(X \mid \mathbf{u})$ is increasing under joint permutation by $P$, then $f'(X \mid \mathcal{D}_{any}) := \mathbb{E}_{\mathbf{u} \sim \mathcal{D}_{any}} \left[ f(X \mid \mathbf{u}) \right]$ is increasing under joint permutation by $P$. If $f$ is invariant under joint permutation by $P$, then so is $f'$.*

*Proof.* Let distribution $\mathcal{D}_{\text{any}}$ have probability measure $F$, and let $\phi \cdot \mathcal{D}_{\text{any}}$ have probability measure $F_\phi$.

$$f\left(X \mid \mathcal{D}_{\text{any}}\right) := \mathop{\mathbb{E}}_{\mathbf{u} \sim \mathcal{D}_{\text{any}}}\left[f(X \mid \mathbf{u})\right] \tag{10}$$

$$:= \int_{\mathbb{R}^d} f(X \mid \mathbf{u}) \, \mathrm{d}F(\mathbf{u}) \tag{11}$$

$$\leq \int_{\mathbb{R}^d} f(\phi \cdot X \mid \mathbf{P}_\phi \mathbf{u}) \, \mathrm{d}F(\mathbf{u}) \tag{12}$$

$$= \int_{\mathbb{R}^d} f(\phi \cdot X \mid \mathbf{u}') \left|\det \mathbf{P}_\phi\right| \mathrm{d}F_\phi(\mathbf{u}') \tag{13}$$

$$= \int_{\mathbb{R}^d} f(\phi \cdot X \mid \mathbf{u}') \, \mathrm{d}F_\phi(\mathbf{u}') \tag{14}$$

$$=: f'\left(\phi \cdot X \mid \phi \cdot \mathcal{D}_{\text{any}}\right). \tag{15}$$

Equation (12) holds by assumption on $f$: $f(X \mid \mathbf{u}) \leq f(\phi \cdot X \mid \mathbf{P}_\phi \mathbf{u})$. Furthermore, $f(\phi \cdot X \mid \cdot)$ is still measurable, and so the inequality holds. Equation (13) follows by the definition of $F_\phi$ (definition 6.3) and by substituting $\mathbf{r}' := \mathbf{P}_\phi \mathbf{r}$. Equation (14) follows from the fact that all permutation matrices have unitary determinant. $\qquad\square$

**Lemma B.6** (Closure of orbit incentives under increasing functions). *Suppose that $S_d$ acts on sets $\mathbf{E}_1, \ldots, \mathbf{E}_m$ (with $\mathbf{E}_1$ being a poset), and let $P \subseteq S_d$. Let $f_1, \ldots, f_n : \prod_{i=1}^m \mathbf{E}_i \to \mathbb{R}$ be increasing under joint permutation by $P$ on input $(X_1, \ldots, X_m)$, and suppose the $f_i$ are order-preserving with respect to $\preceq_{\mathbf{E}_1}$. Let $g : \prod_{j=1}^n \mathbb{R} \to \mathbb{R}$ be monotonically increasing on each argument. Then*

$$f\left(X_1, \ldots, X_m\right) := g\left(f_1\left(X_1, \ldots, X_m\right), \ldots, f_n\left(X_1, \ldots, X_m\right)\right) \tag{16}$$

*is increasing under joint permutation by $P$ and order-preserving with respect to set inclusion on its first argument. Furthermore, if the $f_i$ are* invariant *under joint permutation by $P$, then so is $f$.*

*Proof.* Let $\phi \in P$.

$$f\left(X_1, \ldots, X_m\right) := g\left(f_1\left(X_1, \ldots, X_m\right), \ldots, f_n\left(X_1, \ldots, X_m\right)\right) \tag{17}$$

$$\leq g\left(f_1\left(\phi \cdot X_1, \ldots, \phi \cdot X_m\right), \ldots, f_n\left(\phi \cdot X_1, \ldots, \phi \cdot X_m\right)\right) \tag{18}$$

$$=: f\left(\phi \cdot X_1, \ldots, \phi \cdot X_m\right). \tag{19}$$

Equation (18) follows because we assumed that $f_i\left(X_1, \ldots, X_m\right) \leq f_i\left(\phi \cdot X_1, \ldots, \phi \cdot X_m\right)$, and because $g$ is monotonically increasing on each argument. If the $f_i$ are all invariant, then eq. (18) is an equality.

Similarly, suppose $X_1' \preceq_{\mathbf{E}_1} X_1$. The $f_i$ are order-preserving on the first argument, and $g$ is monotonically increasing on each argument. Then $f\left(X_1', \ldots, X_m\right) \leq f\left(X_1, \ldots, X_m\right)$. This shows that $f$ is order-preserving on its first argument. $\qquad\square$

**Remark.** $g$ could take the convex combination of its arguments, or multiply two $f_i$ together and add them to a third $f_3$.

**Definition 3.5** (Multiply retargetable function). Let $\Theta$ be a subset of a set acted on by $S_d$, and let $f : \{A, B\} \times \Theta \to \mathbb{R}$.

$f$ is a $(\Theta, A \xrightarrow{n} B)$-*retargetable function* when, for each $\theta \in \Theta$, we can choose permutations $\phi_1, \ldots, \phi_n \in S_d$ which satisfy the following conditions: Consider any $\theta^A \in \text{Orbit}|_{\Theta, A > B}(\theta) := \left\{\theta^* \in \text{Orbit}|_\Theta(\theta) \mid f(A \mid \theta^*) > f(B \mid \theta^*)\right\}$.

1. **Retargetable via $n$ permutations.** $\forall i = 1, \ldots, n : f\left(A \mid \phi_i \cdot \theta^A\right) < f\left(B \mid \phi_i \cdot \theta^A\right)$.

2. **Parameter permutation is allowed by $\Theta$.** $\forall i : \phi_i \cdot \theta^A \in \Theta$.

3. **Permuted parameters are distinct.** $\forall i \neq j, \theta' \in \text{Orbit}|_{\Theta, A > B}(\theta) : \phi_i \cdot \theta^A \neq \phi_j \cdot \theta'$.

**Theorem 3.6** (Multiply retargetable functions have orbit-level tendencies)**.**

*If $f$ is $(\Theta, A \overset{n}{\to} B)$-retargetable, then $f(B \mid \theta) \geq^n_{\text{most: } \Theta} f(A \mid \theta)$.*

*Proof.* Let $\theta \in \Theta$, and let $\phi_i \cdot \text{Orbit}|_{\Theta, A > B}(\theta) \coloneqq \{\phi_i \cdot \theta^A \mid \theta^A \in \text{Orbit}|_{\Theta, A > B}(\theta)\}$.

$$\left|\text{Orbit}|_{\Theta, B > A}(\theta)]\right| \geq \left|\bigcup_{i=1}^{n} \phi_i \cdot \text{Orbit}|_{\Theta, A > B}(\theta)\right| \tag{20}$$

$$= \sum_{i=1}^{n} \left|\phi_i \cdot \text{Orbit}|_{\Theta, A > B}(\theta)\right| \tag{21}$$

$$= n \left|\text{Orbit}|_{\Theta, A > B}(\theta)\right|. \tag{22}$$

By item 1 and item 2, $\phi_i \cdot \phi_i \cdot \text{Orbit}|_{\Theta, A > B}(\theta) \subseteq \phi_i \cdot \text{Orbit}|_{\Theta, B > A}(\theta)]$ for all $i$. Therefore, eq. (20) holds. Equation (21) follows by the assumption that parameters are distinct, and so therefore the cosets $\phi_i \cdot \text{Orbit}|_{\Theta, A > B}(\theta)$ and $\phi_j \cdot \text{Orbit}|_{\Theta, A > B}(\theta)$ are pairwise disjoint for $i \neq j$. Equation (22) follows because each $\phi_i$ acts injectively on orbit elements.

Letting $f_A(\theta) \coloneqq f(A \mid \theta)$ and $f_B(\theta) \coloneqq f(B \mid \theta)$, the shown inequality satisfies definition 3.2. We conclude that $f(B \mid \theta) \geq^n_{\text{most: } \Theta} f(A \mid \theta)$. $\qquad \square$

**Definition 3.3** (Simply-retargetable function). Let $\Theta$ be a set acted on by $S_d$, and let $f : \{A, B\} \times \Theta \to \mathbb{R}$. If $\exists \phi \in S_d : \forall \theta^A \in \Theta : f(B \mid \theta^A) < f(A \mid \theta^A) \implies f(A \mid \phi \cdot \theta^A) < f(B \mid \phi \cdot \theta^A)$, then $f$ is a $(\Theta, A \overset{simple}{\to} B)$-*retargetable function.*

**Proposition 3.4** (Simply-retargetable functions have orbit-level tendencies).

*If $f$ is $(\Theta, A \overset{simple}{\to} B)$-retargetable, then $f(B \mid \theta) \geq^1_{\text{most: } \Theta} f(A \mid \theta)$.*

*Proof.* Given that $f$ is a $(\Theta, A \overset{simple}{\to} B)$-retargetable function (definition 3.3), we want to show that $f$ is a $(\Theta, A \overset{1}{\to} B)$-retargetable function (definition 3.5 when $n = 1$). Definition 3.5's item 1 is true by assumption. Since $\Theta$ is acted on by $S_d$, $\Theta$ is closed under permutation and so definition 3.5's item 2 holds. When $n = 1$, there are no $i \neq j$, and so definition 3.5's item 3 is tautologically true.

Then $f$ is a $(\Theta, A \overset{1}{\to} B)$-retargetable function; apply lemma B.7. $\qquad \square$

## B.2 Helper results on retargetable functions

| Targeting parameter $\theta$ | $f(\{👾\} \mid \theta)$ | $f(\{🍎\} \mid \theta)$ | $f(\{🍒\} \mid \theta)$ | $f(\{🍎, 🍒\} \mid \theta)$ |
|---|---|---|---|---|
| $\theta' \coloneqq 1\mathbf{e}_1 + 3\mathbf{e}_2 + 2\mathbf{e}_3$ | 1 | 0 | 0 | 0 |
| $\phi_1 \cdot \theta' = \phi_2 \cdot \theta'' \coloneqq 3\mathbf{e}_1 + 1\mathbf{e}_2 + 2\mathbf{e}_3$ | 0 | 2 | 2 | 2 |
| $\phi_2 \cdot \theta' \coloneqq 2\mathbf{e}_1 + 3\mathbf{e}_2 + 1\mathbf{e}_3$ | 0 | 2 | 2 | 2 |
| $\theta'' \coloneqq 2\mathbf{e}_1 + 1\mathbf{e}_2 + 3\mathbf{e}_3$ | 1 | 0 | 0 | 0 |
| $\phi_1 \cdot \theta'' \coloneqq 1\mathbf{e}_1 + 2\mathbf{e}_2 + 3\mathbf{e}_3$ | 0 | 2 | 2 | 2 |
| $\theta^\star \coloneqq 3\mathbf{e}_1 + 2\mathbf{e}_2 + 1\mathbf{e}_3$ | 1 | 0 | 0 | 0 |

Table 3: We reuse the Pac-Man outcome set introduced in section 2. Let $\phi_1 \coloneqq 👾 \leftrightarrow 🍎$, $\phi_2 \coloneqq 👾 \leftrightarrow 🍒$. We tabularly define a function $f$ which meets all requirements of lemma B.7, except for item 4: letting $j \coloneqq 2$, $f(B_2^\star \mid \phi_1 \cdot \theta') = 2 > 0 = f(B_2^\star \mid \theta')$. Although $f(B \mid \theta) \geq^1_{\text{most: } S_3 \cdot \theta} f(A \mid \theta)$, it is not true that $f(B \mid \theta^*) \geq^2_{\text{most: } S_3 \cdot \theta} f(A \mid \theta^*)$. Therefore, item 4 is generally required.

**Lemma B.7** (Quantitative general orbit lemma). *Let $\Theta$ be a subset of a set acted on by $S_d$, and let $f : \mathbf{E} \times \Theta \to \mathbb{R}$. Consider $A, B \in \mathbf{E}$.*

*For each $\theta \in \Theta$, choose involutions $\phi_1, \ldots, \phi_n \in S_d$. Let $\theta^* \in \text{Orbit}|_{\Theta}(\theta)$.*

1. *Retargetable under parameter permutation. There exist $B_i^\star \in \mathbf{E}$ such that if $f(B \mid \theta^*) < f(A \mid \theta^*)$, then $\forall i : f(A \mid \theta^*) \leq f(B_i^\star \mid \phi_i \cdot \theta^*)$.*

2. **$\Theta$ is closed under certain symmetries.** $f(B \mid \theta^*) < f(A \mid \theta^*) \implies \forall i : \phi_i \cdot \theta^* \in \Theta$.

3. **$f$ is increasing on certain inputs.** $\forall i : f(B_i^\star \mid \theta^*) \leq f(B \mid \theta^*)$.

4. **Increasing under alternate symmetries.** For $j = 1, \ldots, n$ and $i \neq j$, if $f(A \mid \theta^*) < f(B \mid \theta^*)$, then $f\left(B_j^\star \mid \theta^*\right) \leq f\left(B_j^\star \mid \phi_i \cdot \theta^*\right)$.

*If these conditions hold for all $\theta \in \Theta$, then*

$$f(B \mid \theta) \geq_{\text{most: }\Theta}^n f(A \mid \theta). \tag{23}$$

*Proof.* Let $\theta$ and $\theta^*$ be as described in the assumptions, and let $i \in \{1, \ldots, n\}$.

$$
\begin{align}
f(A \mid \phi_i \cdot \theta^*) &= f(A \mid \phi_i^{-1} \cdot \theta^*) \tag{24} \\
&\leq f(B_i^\star \mid \theta^*) \tag{25} \\
&\leq f(B \mid \theta^*) \tag{26} \\
&< f(A \mid \theta^*) \tag{27} \\
&\leq f(B_i^\star \mid \phi_i \cdot \theta^*) \tag{28} \\
&\leq f(B \mid \phi_i \cdot \theta^*). \tag{29}
\end{align}
$$

Equation (24) follows because $\phi_i$ is an involution. Equation (25) and eq. (28) follow by item 1. Equation (26) and eq. (29) follow by item 3. Equation (27) holds by assumption on $\theta^*$. Then eq. (29) shows that for any $i$, $f(A \mid \phi_i \cdot \theta^*) < f(B \mid \phi_i \cdot \theta^*)$, satisfying definition 3.5's item 1.

This result's item 2 satisfies definition 3.5's item 2. We now just need to show definition 3.5's item 3.

**Disjointness.** Let $\theta', \theta'' \in \text{Orbit}|_{\Theta, A > B} (\theta)$ and let $i \neq j$. Suppose $\phi_i \cdot \theta' = \phi_j \cdot \theta''$. We want to show that this leads to contradiction.

$$
\begin{align}
f(A \mid \theta'') &\leq f(B_j^\star \mid \phi_j \cdot \theta'') \tag{30} \\
&= f(B_j^\star \mid \phi_i^{-1} \cdot \theta') \tag{31} \\
&\leq f(B_j^\star \mid \theta') \tag{32} \\
&\leq f(B \mid \theta') \tag{33} \\
&< f(A \mid \theta') \tag{34} \\
&\leq f(B_i^\star \mid \phi_i \cdot \theta') \tag{35} \\
&= f(B_i^\star \mid \phi_j^{-1} \cdot \theta'') \tag{36} \\
&\leq f(B_i^\star \mid \theta'') \tag{37} \\
&\leq f(B \mid \theta'') \tag{38} \\
&< f(A \mid \theta''). \tag{39}
\end{align}
$$

Equation (30) follows by our assumption of item 1. Equation (31) holds because we assumed that $\phi_j \cdot \theta'' = \phi_i \cdot \theta'$, and the involution ensures that $\phi_i = \phi_i^{-1}$. Equation (32) is guaranteed by our assumption of item 4, given that $\phi_i^{-1} \cdot \theta' = \phi_i \cdot \theta' \in \text{Orbit}|_{\Theta, B > A} (\theta)]$ by the first half of this proof. Equation (33) follows by our assumption of item 3. Equation (34) follows because we assumed that $\theta' \in \text{Orbit}|_{\Theta, A > B} (\theta)$.

Equation (35) through eq. (39) follow by the same reasoning, switching the roles of $\theta'$ and $\theta''$, and of $i$ and $j$. But then we have demonstrated that a quantity is strictly less than itself, a contradiction. So for all $\theta', \theta'' \in \text{Orbit}|_{\Theta, A > B} (\theta)$, when $i \neq j$, $\phi_i \cdot \theta' \neq \phi_j \cdot \theta''$.

Therefore, we have shown definition 3.5's item 3, and so $f$ is a $(\Theta, A \xrightarrow{n} B)$-retargetable function. Apply theorem 3.6 in order to conclude that eq. (23) holds. $\qquad \square$

**Definition B.8** (Superset-of-copy containment). Let $A, B \subseteq \mathbb{R}^d$. $B$ *contains $n$ superset-copies $B_i^\star$ of $A$ when there exist involutions $\phi_1, \ldots, \phi_n$ such that $\phi_i \cdot A \subseteq B_i^\star \subseteq B$, and whenever $i \neq j$, $\phi_i \cdot B_j^\star = B_j^\star$.*

**Lemma B.9** (Looser sufficient conditions for orbit-level incentives). *Suppose that $\Theta$ is a subset of a set acted on by $S_d$ and is closed under permutation by $S_d$. Let $A, B \in \mathbf{E} \subseteq \mathcal{P}\left(\mathbb{R}^d\right)$. Suppose that $B$ contains $n$ superset-copies $B_i^\star \in \mathbf{E}$ of $A$ via $\phi_i$. Suppose that $f(X \mid \theta)$ is increasing under joint permutation by $\phi_1, \dots, \phi_n \in S_d$ for all $X \in \mathbf{E}, \theta \in \Theta$, and suppose that $\forall i : \phi_i \cdot A \in \mathbf{E}$. Suppose that $f$ is monotonically increasing on its first argument. Then $f(B \mid \theta) \geq^n_{\text{most: }\Theta} f(A \mid \theta)$.*

*Proof.* We check the conditions of lemma B.7. Let $\theta \in \Theta$, and let $\theta^* \in (S_d \cdot \theta) \cap \Theta$ be an orbit element.

Item 1. Holds since $f(A \mid \theta^*) \leq f(\phi_i \cdot A \mid \phi_i \cdot \theta^*) \leq f(B_i^\star \mid \phi_i \cdot \theta^*)$, with the first inequality by assumption of joint increasing under permutation, and the second following from monotonicity (as $\phi_i \cdot A \subseteq B_i^\star$ by superset copy definition B.8).

Item 2. We have $\forall \theta^* \in (S_d \cdot \theta^*) \cap \Theta : f(B \mid \theta^*) < f(A \mid \theta^*) \implies \forall i = 1, \dots, n : \phi_i \cdot \theta^* \in \Theta$ since $\Theta$ is closed under permutation.

Item 3. Holds because we assumed that $f$ is monotonic on its first argument.

Item 4. Holds because $f$ is increasing under joint permutation on *all* of its inputs $X, \theta'$, and definition B.8 shows that $\phi_i \cdot B_j^\star = B_j^\star$ when $i \neq j$. Combining these two steps of reasoning, for *all* $\theta' \in \Theta$, it is true that $f\left(B_j^\star \mid \theta'\right) \leq f\left(\phi_i \cdot B_j^\star \mid \phi_i \cdot \theta'\right) \leq f\left(B_j^\star \mid \phi_i \cdot \theta'\right)$.

Then apply lemma B.7. $\qquad \square$

**Lemma B.10** (Hiding an argument which is invariant under certain permutations). *Let $\mathbf{E}_1, \mathbf{E}_2, \Theta$ be subsets of sets which are acted on by $S_d$. Let $A \in \mathbf{E}_1, C \in \mathbf{E}_2$. Suppose there exist $\phi_1, \dots, \phi_n \in S_d$ such that $\phi_i \cdot C = C$. Suppose $h : \mathbf{E}_1 \times \mathbf{E}_2 \times \Theta \to \mathbb{R}$ satisfies $\forall i : h(A, C \mid \theta) \leq h(\phi_i \cdot A, \phi_i \cdot C \mid \phi_i \cdot \theta)$. For any $X \in \mathbf{E}_1$, let $f(X \mid \theta) := h(X, C \mid \theta)$. Then $f(A \mid \theta)$ is increasing under joint permutation by $\phi_i$.*

*Furthermore, if $h$ is invariant under joint permutation by $\phi_i$, then so is $f$.*

*Proof.*

$$f(X \mid \theta) := h(X, C \mid \theta) \tag{40}$$
$$\leq h(\phi_i \cdot X, \phi_i \cdot C \mid \phi_i \cdot \theta) \tag{41}$$
$$= h(\phi_i \cdot X, C \mid \phi_i \cdot \theta) \tag{42}$$
$$=: f(\phi_i \cdot X \mid \phi_i \cdot \theta). \tag{43}$$

Equation (41) holds by assumption. Equation (42) follows because we assumed $\phi_i \cdot C = C$. Then $f$ is increasing under joint permutation by the $\phi_i$.

If $h$ is *invariant*, then eq. (41) is an equality, and so $\forall i : f(X \mid \theta) = f(\phi_i \cdot X \mid \phi_i \cdot \theta)$. $\qquad \square$

### B.2.1 EU-determined functions

Lemma B.11 and lemma B.5 together extend Turner et al. [2021]'s lemma E.17 beyond functions of $\max_{\mathbf{x} \in X_i}$, to any functions of cardinalities and of expected utilities of set elements.

**Definition A.12** (EU-determined functions). Let $\mathcal{P}\left(\mathbb{R}^d\right)$ be the power set of $\mathbb{R}^d$, and let $f : \prod_{i=1}^m \mathcal{P}\left(\mathbb{R}^d\right) \times \mathbb{R}^d \to \mathbb{R}$. $f$ is an *EU-determined function* if there exists a family of functions $\{g^{\omega_1, \dots, \omega_m}\}$ such that

$$f(X_1, \dots, X_m \mid \mathbf{u}) = g^{|X_1|, \dots, |X_m|}\left(\left[\mathbf{x}_1^\top \mathbf{u}\right]_{\mathbf{x}_1 \in X_1}, \dots, \left[\mathbf{x}_m^\top \mathbf{u}\right]_{\mathbf{x}_m \in X_m}\right), \tag{4}$$

where $[r_i]$ is the multiset of its elements $r_i$.

**Lemma B.11** (EU-determined functions are invariant under joint permutation). *Suppose that $f : \prod_{i=1}^m \mathcal{P}\left(\mathbb{R}^d\right) \times \mathbb{R}^d \to \mathbb{R}$ is an EU-determined function. Then for any $\phi \in S_d$ and $X_1, \dots, X_m, \mathbf{u}$, we have $f(X_1, \dots, X_m \mid \mathbf{u}) = f(\phi \cdot X_1, \dots, \phi \cdot X_m \mid \phi \cdot \mathbf{u})$.*

*Proof.*

$$f(X_1, \ldots, X_m \mid \mathbf{u}) \tag{44}$$

$$= g^{|X_1|,\ldots,|X_m|}\left(\left[\mathbf{x}_1^\top \mathbf{u}\right]_{\mathbf{x}_1 \in X_1}, \ldots, \left[\mathbf{x}_m^\top \mathbf{u}\right]_{\mathbf{x}_m \in X_m}\right) \tag{45}$$

$$= g^{|\phi \cdot X_1|,\ldots,|\phi \cdot X_m|}\left(\left[\mathbf{x}_1^\top \mathbf{u}\right]_{\mathbf{x}_1 \in X_1}, \ldots, \left[\mathbf{x}_m^\top \mathbf{u}\right]_{\mathbf{x}_m \in X_m}\right) \tag{46}$$

$$= g^{|\phi \cdot X_1|,\ldots,|\phi \cdot X_m|}\left(\left[(\mathbf{P}_\phi \mathbf{x}_1)^\top (\mathbf{P}_\phi \mathbf{u})\right]_{\mathbf{x}_1 \in X_1}, \ldots, \left[(\mathbf{P}_\phi \mathbf{x}_m)^\top (\mathbf{P}_\phi \mathbf{u})\right]_{\mathbf{x}_m \in X_m}\right) \tag{47}$$

$$= f(\phi \cdot X_1, \ldots, \phi \cdot X_m \mid \phi \cdot \mathbf{u}). \tag{48}$$

Equation (46) holds because permutations $\phi$ act injectively on $\mathbb{R}^d$. Equation (47) follows because $\mathbf{I} = \mathbf{P}_\phi^{-1}\mathbf{P}_\phi = \mathbf{P}_\phi^\top \mathbf{P}_\phi$ by the orthogonality of permutation matrices, and $\mathbf{x}^\top \mathbf{P}_\phi^\top = (\mathbf{P}_\phi \mathbf{x})^\top$, so $\mathbf{x}^\top \mathbf{u} = \mathbf{x}^\top \mathbf{P}_\phi^\top \mathbf{P}_\phi \mathbf{u} = (\mathbf{P}_\phi \mathbf{x})^\top (\mathbf{P}_\phi \mathbf{u})$. $\square$

**Theorem A.13** (Orbit tendencies occur for EU-determined decision-making functions)**.** *Let $A, B, C \subseteq \mathbb{R}^d$ be such that $B$ contains $n$ copies of $A$ via $\phi_i$ such that $\phi_i \cdot C = C$. Let $h : \prod_{i=1}^2 \mathcal{P}\left(\mathbb{R}^d\right) \times \mathbb{R}^d \to \mathbb{R}$ be an EU-determined function, and let $p(X \mid \mathbf{u}) := h(X, C \mid \mathbf{u})$. Suppose that $p$ returns a probability of selecting an element of $X$ from $C$. Then $p(B \mid \mathbf{u}) \geq^n_{\text{most: } \mathbb{R}^d} p(A \mid \mathbf{u})$.*

*Proof.* By assumption, there exists a family of functions $\left\{g^{i,|C|}\right\}$ such that for all $X \subseteq \mathbb{R}^d$, $h(X, C \mid \mathbf{u}) = g^{|X|,|C|}\left(\left[\mathbf{x}^\top \mathbf{u}\right]_{\mathbf{x} \in X}, \left[\mathbf{c}^\top \mathbf{u}\right]_{\mathbf{c} \in C}\right)$. Therefore, lemma B.11 shows that $h(A, C \mid \mathbf{u})$ is invariant under joint permutation by the $\phi_i$. Letting $\Theta := \mathbb{R}^d$, apply lemma B.10 to conclude that $f(X \mid \mathbf{u})$ is invariant under joint permutation by the $\phi_i$.

Since $f$ returns a probability of selecting an element of $X$, $f$ obeys the monotonicity probability axiom: If $X' \subseteq X$, then $f(X' \mid \mathbf{u}) \leq f(X \mid \mathbf{u})$. Then $f(B \mid \mathbf{u}) \geq^n_{\text{most: } \mathbb{R}^d} f(A \mid \mathbf{u})$ by lemma B.9. $\square$

### B.3  Particular results on retargetable functions

**Definition B.12** (Quantilization, closed form)**.** Let the expected utility $q$-quantile threshold be

$$M_{q,P}(C \mid \mathbf{u}) := \inf\left\{M \in \mathbb{R} \mid \mathbb{P}_{\mathbf{x} \sim P}\left(\mathbf{x}^\top \mathbf{u} > M\right) \leq q\right\}. \tag{49}$$

Let $C_{>M_{q,P}(C|\mathbf{u})} := \left\{\mathbf{c} \in C \mid \mathbf{c}^\top \mathbf{u} > M_{q,P}(C \mid \mathbf{u})\right\}$. $C_{=M_{q,P}(C|\mathbf{u})}$ is defined similarly. Let $\mathbb{1}_{L(x)}$ be the predicate function returning 1 if $L(x)$ is true and 0 otherwise. Then for $X \subseteq C$,

$$Q_{q,P}(X \mid C, \mathbf{u}) := \sum_{\mathbf{x} \in X} \frac{P(\mathbf{x})}{q}\left(\mathbb{1}_{\mathbf{x} \in C_{>M_{q,P}(C|\mathbf{u})}} + \frac{\mathbb{1}_{\mathbf{x} \in C_{=M_{q,P}(C|\mathbf{u})}}}{P\left(C_{=M_{q,P}(C|\mathbf{u})}\right)}\left(q - P\left(C_{>M_{q,P}(C|\mathbf{u})}\right)\right)\right), \tag{50}$$

where the summand is defined to be 0 if $P(\mathbf{x}) = 0$ and $\mathbf{x} \in C_{=M_{q,P}(C|\mathbf{u})}$.

**Remark.** Unlike Taylor [2016]'s or Carey [2019]'s definitions, definition B.12 is written in closed form and requires no arbitrary tie-breaking. Instead, in the case of an expected utility tie on the quantile threshold, eq. (50) allots probability to outcomes proportional to their probability under the base distribution $P$.

Thanks to theorem A.13, we straightforwardly prove most items of proposition A.11 by just rewriting each decision-making function as an EU-determined function. Most of the proof's length comes from showing that the functions are measurable on $\mathbf{u}$, which means that the results also apply for distributions over utility functions $\mathcal{D}_{\text{any}} \in \mathfrak{D}_{\text{any}}$.

**Proposition A.11** (Orbit incentives for different rationalities)**.** *Let $A, B \subseteq C \subsetneq \mathbb{R}^d$ be finite, such that $B$ contains $n$ copies of $A$ via involutions $\phi_i$ such that $\phi_i \cdot C = C$.*

1. **Rational choice [Turner et al., 2021].**

$$\text{IsOptimal}\left(B \mid C, \mathcal{D}_{any}\right) \geq^n_{\text{most: } \mathfrak{D}_{any}} \text{IsOptimal}\left(A \mid C, \mathcal{D}_{any}\right).$$

2. **Uniformly randomly choosing an optimal lottery.** *For* $X \subseteq C$, *let*

$$\text{FracOptimal}\left(X \mid C, \mathbf{u}\right) := \frac{\left|\left\{\arg\max_{\mathbf{c} \in C} \mathbf{c}^\top \mathbf{u}\right\} \cap X\right|}{\left|\left\{\arg\max_{\mathbf{c} \in C} \mathbf{c}^\top \mathbf{u}\right\}\right|}.$$

   *Then* $\text{FracOptimal}\left(B \mid C, \mathcal{D}_{any}\right) \geq^n_{\text{most: } \mathfrak{D}_{any}} \text{FracOptimal}\left(A \mid C, \mathcal{D}_{any}\right)$.

3. **Anti-rational choice.** $\text{AntiOpt}\left(B \mid C, \mathcal{D}_{any}\right) \geq^n_{\text{most: } \mathfrak{D}_{any}} \text{AntiOpt}\left(A \mid C, \mathcal{D}_{any}\right)$.

4. **Boltzmann rationality.**

$$\text{Boltzmann}_T\left(B \mid C, \mathcal{D}_{any}\right) \geq^n_{\text{most: } \mathfrak{D}_{any}} \text{Boltzmann}_T\left(A \mid C, \mathcal{D}_{any}\right).$$

5. **Uniformly randomly drawing** $k$ **outcome lotteries and choosing the best.** *For* $X \subseteq C$, $\mathbf{u} \in \mathbb{R}^d$, *and* $k \geq 1$, *let*

$$\textit{best-of-}k(X, C \mid \mathbf{u}) := \mathop{\mathbb{E}}_{\mathbf{a}_1,\ldots,\mathbf{a}_k \sim unif(C)} \left[\text{FracOptimal}\left(X \cap \{\mathbf{a}_1,\ldots,\mathbf{a}_k\} \mid \{\mathbf{a}_1,\ldots,\mathbf{a}_k\}, \mathbf{u}\right)\right].$$

   *Then* $\textit{best-of-}k(B \mid C, \mathcal{D}_{any}) \geq^n_{\text{most: } \mathfrak{D}_{any}} \textit{best-of-}k(A \mid C, \mathcal{D}_{any})$.

6. **Satisficing [Simon, 1956].** $\text{Satisfice}_t\left(B \mid C, \mathcal{D}_{any}\right) \geq^n_{\text{most: } \mathfrak{D}_{any}} \text{Satisfice}_t\left(A \mid C, \mathcal{D}_{any}\right)$.

7. **Quantilizing over outcome lotteries [Taylor, 2016].** *Let* $P$ *be the uniform probability distribution over* $C$. *For* $X \subseteq C$, $\mathbf{u} \in \mathbb{R}^d$, *and* $q \in (0, 1]$, *let* $Q_{q,P}(X \mid C, \mathbf{u})$ *(definition B.12) return the probability that an outcome lottery in* $X$ *is drawn from the top* $q$-*quantile of* $P$, *sorted by expected utility under* $\mathbf{u}$. *Then* $Q_{q,P}(B \mid C, \mathbf{u}) \geq^n_{\text{most: } \mathbb{R}^d} Q_{q,P}(A \mid C, \mathbf{u})$.

*Proof.* **Item 1.** Consider

$$h(X, C \mid \mathbf{u}) := \mathbb{1}_{\exists \mathbf{x} \in X : \forall \mathbf{c} \in C : \mathbf{x}^\top \mathbf{u} \geq \mathbf{c}^\top \mathbf{u}} \tag{51}$$

$$= \min\left(1, \sum_{\mathbf{x} \in X} \prod_{\mathbf{c} \in C} \mathbb{1}_{(\mathbf{x}-\mathbf{c})^\top \mathbf{u} \geq 0}\right). \tag{52}$$

Since halfspaces are measurable, each indicator function is measurable on $\mathbf{u}$. The finite sum of the finite product of measurable functions is also measurable. Since min is continuous (and therefore measurable), $h(X, C \mid \mathbf{u})$ is measurable on $\mathbf{u}$.

Furthermore, $h$ is an EU-determined function:

$$h(X, C \mid \mathbf{u}) = g\left(\overbrace{\left[\mathbf{x}^\top \mathbf{u}\right]_{\mathbf{x} \in X}}^{V_X}, \overbrace{\left[\mathbf{c}^\top \mathbf{u}\right]_{\mathbf{c} \in C}}^{V_C}\right) \tag{53}$$

$$:= \mathbb{1}_{\exists v_x \in V_X : \forall v_c \in V_C : v_x \geq v_c}. \tag{54}$$

Then by lemma B.11, $h$ is invariant to joint permutation by the $\phi_i$. Since $\phi_i \cdot C = C$, lemma B.10 shows that $h'(X \mid \mathbf{u}) := h(X, C \mid \mathbf{u})$ is also invariant under joint permutation by the $\phi_i$. Since $h$ is a measurable function of $\mathbf{u}$, so is $h'$. Then since $h'$ is bounded, lemma B.5 shows that $f(X \mid \mathcal{D}_{any}) := \mathbb{E}_{\mathbf{u} \sim \mathcal{D}_{any}}\left[h'(X \mid \mathbf{u})\right]$ is invariant under joint permutation by $\phi_i$.

Furthermore, if $X' \subseteq X$, $f(X' \mid \mathcal{D}_{any}) \leq f(X \mid \mathcal{D}_{any})$ by the monotonicity of probability. Then by lemma B.9,

$$f(B \mid \mathcal{D}_{any}) := \text{IsOptimal}\left(B \mid C, \mathcal{D}_{any}\right) \geq^n_{\text{most: } \mathfrak{D}_{any}} \text{IsOptimal}\left(A \mid C, \mathcal{D}_{any}\right) =: f(A \mid \mathcal{D}_{any}).$$

**Item 2.** Because $X, C$ are finite sets, the denominator of $\mathrm{FracOptimal}\left(X \mid C, \mathbf{u}\right)$ is never zero, and so the function is well-defined. $\mathrm{FracOptimal}\left(X \mid C, \mathbf{u}\right)$ is an EU-determined function:

$$\mathrm{FracOptimal}\left(X \mid C, \mathbf{u}\right) = g\left(\overbrace{\left[\mathbf{x}^\top \mathbf{u}\right]_{\mathbf{x} \in X}}^{V_X}, \overbrace{\left[\mathbf{c}^\top \mathbf{u}\right]_{\mathbf{c} \in C}}^{V_C}\right) \tag{55}$$

$$:= \frac{\left|\left[v \in V_X \mid v = \max_{v' \in V_C} v'\right]\right|}{\left|\left[\arg\max_{v' \in V_C} v'\right]\right|}, \tag{56}$$

with the $[\cdot]$ denoting a multiset which allows and counts duplicates. Then by lemma B.11, $\mathrm{FracOptimal}\left(X \mid C, \mathbf{u}\right)$ is invariant to joint permutation by the $\phi_i$.

We now show that $\mathrm{FracOptimal}\left(X \mid C, \mathbf{u}\right)$ is a measurable function of $\mathbf{u}$.

$$\mathrm{FracOptimal}\left(X \mid C, \mathbf{u}\right) := \frac{\left|\left\{\arg\max_{\mathbf{c}' \in C} \mathbf{c}'^\top \mathbf{u}\right\} \cap X\right|}{\left|\left\{\arg\max_{\mathbf{c}' \in C} \mathbf{c}'^\top \mathbf{u}\right\}\right|} \tag{57}$$

$$= \frac{\sum_{\mathbf{x} \in X} \mathbb{1}_{\mathbf{x} \in \arg\max_{\mathbf{c}' \in C} \mathbf{c}'^\top \mathbf{u}}}{\sum_{\mathbf{c} \in C} \mathbb{1}_{\mathbf{c} \in \arg\max_{\mathbf{c}' \in C} \mathbf{c}'^\top \mathbf{u}}} \tag{58}$$

$$= \frac{\sum_{\mathbf{x} \in X} \prod_{\mathbf{c}' \in C} \mathbb{1}_{(\mathbf{x}-\mathbf{c}')^\top \mathbf{u} \geq 0}}{\sum_{\mathbf{c} \in C} \prod_{\mathbf{c}' \in C} \mathbb{1}_{(\mathbf{c}-\mathbf{c}')^\top \mathbf{u} \geq 0}}. \tag{59}$$

Equation (59) holds because $\mathbf{x}$ belongs to the $\arg\max$ iff $\forall \mathbf{c} \in C : \mathbf{x}^\top \mathbf{u} \geq \mathbf{c}^\top \mathbf{u}$. Furthermore, this condition is met iff $\mathbf{u}$ belongs to the intersection of finitely many closed halfspaces; therefore, $\left\{\mathbf{u} \in \mathbb{R}^d \mid \prod_{\mathbf{c} \in C} \mathbb{1}_{(\mathbf{x}-\mathbf{c})^\top \mathbf{u} \geq 0} = 1\right\}$ is measurable. Then the sums in both the numerator and denominator are both measurable functions of $\mathbf{u}$, and the denominator cannot vanish. Therefore, $\mathrm{FracOptimal}\left(X \mid C, \mathbf{u}\right)$ is a measurable function of $\mathbf{u}$.

Let $g(X \mid \mathbf{u}) := \mathrm{FracOptimal}\left(X \mid C, \mathbf{u}\right)$. Since $\phi_i \cdot C = C$, lemma B.10 shows that $g(X \mid \mathbf{u})$ is also invariant to joint permutation by $\phi_i$. Since $g$ is measurable and bounded $[0, 1]$, apply lemma B.5 to conclude that $f(X \mid \mathcal{D}_{\mathrm{any}}) := \mathbb{E}_{\mathbf{u} \sim \mathcal{D}_{\mathrm{any}}}\left[g(X \mid C, \mathbf{u})\right]$ is also invariant to joint permutation by $\phi_i$.

Furthermore, if $X' \subseteq X \subseteq C$, then $f(X' \mid \mathcal{D}_{\mathrm{any}}) \leq f(X \mid \mathcal{D}_{\mathrm{any}})$. So apply lemma B.9 to conclude that $\mathrm{FracOptimal}\left(B \mid C, \mathcal{D}_{\mathrm{any}}\right) =: f(B \mid \mathcal{D}_{\mathrm{any}}) \geq^n_{\mathrm{most:}\ \mathfrak{D}_{\mathrm{any}}} f(A \mid \mathcal{D}_{\mathrm{any}}) := \mathrm{FracOptimal}\left(A \mid C, \mathcal{D}_{\mathrm{any}}\right)$.

**Item 3.** Apply the reasoning in item 1 with inner function $h(X \mid C, \mathbf{u}) := \mathbb{1}_{\exists \mathbf{x} \in X : \forall \mathbf{c} \in C : \mathbf{x}^\top \mathbf{u} \leq \mathbf{c}^\top \mathbf{u}}$.

**Item 4.** Let $X \subseteq C$. $\mathrm{Boltzmann}_T\left(X \mid C, \mathbf{u}\right)$ is the expectation of an EU function:

$$\mathrm{Boltzmann}_T\left(X \mid C, \mathbf{u}\right) = g_T\left(\overbrace{\left[\mathbf{x}^\top \mathbf{u}\right]_{\mathbf{x} \in X}}^{V_X}, \overbrace{\left[\mathbf{c}^\top \mathbf{u}\right]_{\mathbf{c} \in C}}^{V_C}\right) \tag{60}$$

$$:= \frac{\sum_{v \in V_X} e^{v/T}}{\sum_{v \in V_C} e^{v/T}}. \tag{61}$$

Therefore, by lemma B.11, $\mathrm{Boltzmann}_T\left(X \mid C, \mathbf{u}\right)$ is invariant to joint permutation by the $\phi_i$.

Inspecting eq. (61), we see that $g$ is continuous on $\mathbf{u}$ (and therefore measurable), and bounded $[0, 1]$ since $X \subseteq C$ and the exponential function is positive. Therefore, by lemma B.5, the expectation version is also invariant to joint permutation for all permutations $\phi \in S_d$: $\mathrm{Boltzmann}_T\left(X \mid C, \mathcal{D}_{\mathrm{any}}\right) = \mathrm{Boltzmann}_T\left(\phi \cdot X \mid \phi \cdot C, \phi \cdot \mathcal{D}_{\mathrm{any}}\right)$.

Since $\phi_i \cdot C = C$, lemma B.10 shows that $f(X \mid \mathcal{D}_{\mathrm{any}}) := \mathrm{Boltzmann}_T\left(X \mid C, \mathcal{D}_{\mathrm{any}}\right)$ is also invariant under joint permutation by the $\phi_i$. Furthermore, if $X' \subseteq X$, then $f(X' \mid \mathcal{D}_{\mathrm{any}}) \leq$

$f(X \mid \mathcal{D}_{\text{any}})$. Then apply lemma B.9 to conclude that $\text{Boltzmann}_T \left( B \mid C, \mathcal{D}_{\text{any}} \right) =: f(B \mid \mathcal{D}_{\text{any}}) \geq^n_{\text{most: } \mathfrak{D}_{\text{any}}} f(A \mid \mathcal{D}_{\text{any}}) := \text{Boltzmann}_T \left( A \mid C, \mathcal{D}_{\text{any}} \right)$.

**Item 5.** Let involution $\phi \in S_d$ fix $C$ (*i.e.* $\phi \cdot C = C$).

$$\text{best-of-}k(X \mid C, \mathbf{u}) \tag{62}$$

$$:= \mathop{\mathbb{E}}_{\mathbf{a}_1, \ldots, \mathbf{a}_k \sim \text{unif}(C)} \left[ \text{FracOptimal} \left( X \cap \{\mathbf{a}_1, \ldots, \mathbf{a}_k\} \mid \{\mathbf{a}_1, \ldots, \mathbf{a}_k\}, \mathbf{u} \right) \right] \tag{63}$$

$$= \mathop{\mathbb{E}}_{\mathbf{a}_1, \ldots, \mathbf{a}_k \sim \text{unif}(C)} \left[ \text{FracOptimal} \left( (\phi \cdot X) \cap \{\phi \cdot \mathbf{a}_1, \ldots, \phi \cdot \mathbf{a}_k\} \mid \{\phi \cdot \mathbf{a}_1, \ldots, \phi \cdot \mathbf{a}_k\}, \phi \cdot \mathbf{u} \right) \right] \tag{64}$$

$$= \mathop{\mathbb{E}}_{\phi \cdot \mathbf{a}_1, \ldots, \phi \cdot \mathbf{a}_k \sim \text{unif}(\phi \cdot C)} \left[ \text{FracOptimal} \left( (\phi \cdot X) \cap \{\phi \cdot \mathbf{a}_1, \ldots, \phi \cdot \mathbf{a}_k\} \mid \{\phi \cdot \mathbf{a}_1, \ldots, \phi \cdot \mathbf{a}_k\}, \phi \cdot \mathbf{u} \right) \right] \tag{65}$$

$$=: \text{best-of-}k(\phi \cdot X \mid \phi \cdot C, \phi \cdot \mathbf{u}). \tag{66}$$

By the proof of item 2,

$$\text{FracOptimal} \left( X \cap \{\mathbf{a}_1, \ldots, \mathbf{a}_k\} \mid \{\mathbf{a}_1, \ldots, \mathbf{a}_k\}, \mathbf{u} \right) =$$
$$\text{FracOptimal} \left( (\phi \cdot X) \cap \{\phi \cdot \mathbf{a}_1, \ldots, \phi \cdot \mathbf{a}_k\} \mid \{\phi \cdot \mathbf{a}_1, \ldots, \phi \cdot \mathbf{a}_k\}, \phi \cdot \mathbf{u} \right);$$

thus, eq. (64) holds. Since $\phi \cdot C = C$ and since the distribution is uniform, eq. (65) holds. Therefore, $\text{best-of-}k(X \mid C, \mathbf{u})$ is invariant to joint permutation by the $\phi_i$, which are involutions fixing $C$.

We now show that $\text{best-of-}k(X \mid C, \mathbf{u})$ is measurable on $\mathbf{u}$.

$$\text{best-of-}k(X \mid C, \mathbf{u}) \tag{67}$$

$$:= \mathop{\mathbb{E}}_{\mathbf{a}_1, \ldots, \mathbf{a}_k \sim \text{unif}(C)} \left[ \text{FracOptimal} \left( X \cap \{\mathbf{a}_1, \ldots, \mathbf{a}_k\} \mid \{\mathbf{a}_1, \ldots, \mathbf{a}_k\}, \mathbf{u} \right) \right] \tag{68}$$

$$= \frac{1}{|C|^k} \sum_{(\mathbf{a}_1, \ldots, \mathbf{a}_k) \in C^k} \text{FracOptimal} \left( X \cap \{\mathbf{a}_1, \ldots, \mathbf{a}_k\} \mid \{\mathbf{a}_1, \ldots, \mathbf{a}_k\}, \mathbf{u} \right). \tag{69}$$

Equation (69) holds because $\text{FracOptimal} \left( X \mid C, \mathbf{u} \right)$ is measurable on $\mathbf{u}$ by item 2, and measurable functions are closed under finite addition and scalar multiplication. Then $\text{best-of-}k(X \mid C, \mathbf{u})$ is measurable on $\mathbf{u}$.

Let $g(X \mid \mathbf{u}) := \text{best-of-}k(X \mid C, \mathbf{u})$. Since $\phi_i \cdot C = C$, lemma B.10 shows that $g(X \mid \mathbf{u})$ is also invariant to joint permutation by $\phi_i$. Since $g$ is measurable and bounded $[0, 1]$, apply lemma B.5 to conclude that $f(X \mid \mathcal{D}_{\text{any}}) := \mathbb{E}_{\mathbf{u} \sim \mathcal{D}_{\text{any}}} \left[ g(X \mid C, \mathbf{u}) \right]$ is also invariant to joint permutation by $\phi_i$.

Furthermore, if $X' \subseteq X \subseteq C$, then $f(X' \mid \mathcal{D}_{\text{any}}) \leq f(X \mid \mathcal{D}_{\text{any}})$. So apply lemma B.9 to conclude that $\text{best-of-}k(B \mid C, \mathcal{D}_{\text{any}}) =: f(B \mid \mathcal{D}_{\text{any}}) \geq^n_{\text{most: } \mathfrak{D}_{\text{any}}} f(A \mid \mathcal{D}_{\text{any}}) := \text{best-of-}k(A \mid C, \mathcal{D}_{\text{any}})$.

**Item 6.** $\text{Satisfice}_t \left( X \mid C, \mathbf{u} \right)$ is an EU-determined function:

$$\text{Satisfice}_t \left( X \mid C, \mathbf{u} \right) = g_t \left( \overbrace{\left[ \mathbf{x}^\top \mathbf{u} \right]_{\mathbf{x} \in X}}^{V_X}, \overbrace{\left[ \mathbf{c}^\top \mathbf{u} \right]_{\mathbf{c} \in C}}^{V_C} \right) \tag{70}$$

$$:= \frac{\sum_{v \in V_X} \mathbb{1}_{v \geq t}}{\sum_{v \in V_C} \mathbb{1}_{v \geq t}}, \tag{71}$$

with the function evaluating to 0 if the denominator is 0.
Then applying lemma B.11, $\text{Satisfice}_t \left( X \mid C, \mathbf{u} \right)$ is invariant under joint permutation by the $\phi_i$.

We now show that $\text{Satisfice}_t \left( X \mid C, \mathbf{u} \right)$ is measurable on $\mathbf{u}$.

$$\text{Satisfice}_t \left( X \mid C, \mathbf{u} \right) = \begin{cases} \frac{\sum_{\mathbf{x} \in X} \mathbb{1}_{\mathbf{x} \in \left\{ \mathbf{x}' \in \mathbb{R}^d \mid \mathbf{x}'^\top \mathbf{u} \geq t \right\}}}{\sum_{\mathbf{c} \in C} \mathbb{1}_{\mathbf{c} \in \left\{ \mathbf{x}' \in \mathbb{R}^d \mid \mathbf{x}'^\top \mathbf{u} \geq t \right\}}} & \exists \mathbf{c} \in C : \mathbf{c}^\top \mathbf{u} \geq t, \\ 0 & \text{else.} \end{cases} \tag{72}$$

Consider the two cases.

$$\exists \mathbf{c} \in C : \mathbf{c}^\top \mathbf{u} \geq t \iff \mathbf{u} \in \bigcup_{\mathbf{c} \in C} \left\{ \mathbf{u}' \in \mathbb{R}^d \mid \mathbf{c}^\top \mathbf{u} \geq t \right\}.$$

The right-hand set is the union of finitely many halfspaces (which are measurable), and so the right-hand set is also measurable. Then the casing is a measurable function of $\mathbf{u}$. Clearly the zero function is measurable. Now we turn to the first case.

In the first case, eq. (72)'s indicator functions test each $\mathbf{x}, \mathbf{c}$ for membership in a closed halfspace with respect to $\mathbf{u}$. Halfspaces are measurable sets. Therefore, the indicator function is a measurable function of $\mathbf{u}$, and so are the finite sums. Since the denominator does not vanish within the case, the first case as a whole is a measurable function of $\mathbf{u}$. Therefore, $\mathrm{Satisfice}_t\left( X \mid C, \mathbf{u} \right)$ is measurable on $\mathbf{u}$.

Since $\mathrm{Satisfice}_t\left( X \mid C, \mathbf{u} \right)$ is measurable and bounded $[0, 1]$ (as $X \subseteq C$), apply lemma B.5 to conclude that $\mathrm{Satisfice}_t\left( X \mid C, \mathcal{D}_{\mathrm{any}} \right) = \mathrm{Satisfice}_t\left( \phi \cdot X \mid \phi \cdot C, \phi \cdot \mathcal{D}_{\mathrm{any}} \right)$. Next, let $f(X \mid \mathcal{D}_{\mathrm{any}}) :=$ $\mathrm{Satisfice}_t\left( X \mid C, \mathcal{D}_{\mathrm{any}} \right)$. Since we just showed that $\mathrm{Satisfice}_t\left( X \mid C, \mathcal{D}_{\mathrm{any}} \right)$ is invariant to joint permutation by the involutions $\phi_i$ and since $\phi_i \cdot C = C$, $f(X \mid \mathcal{D}_{\mathrm{any}})$ is also invariant to joint permutation by $\phi_i$.

Furthermore, if $X' \subseteq X$, we have $f(X' \mid \mathcal{D}_{\mathrm{any}}) \leq f(X \mid \mathcal{D}_{\mathrm{any}})$. Then applying lemma B.9, $\mathrm{Satisfice}_t\left( B \mid C, \mathbf{u} \right) =: f(B \mid \mathcal{D}_{\mathrm{any}}) \geq^n_{\mathrm{most:}\ \mathfrak{D}_{\mathrm{any}}} f(A \mid \mathcal{D}_{\mathrm{any}}) := \mathrm{Satisfice}_t\left( A \mid C, \mathbf{u} \right)$.

**Item 7.** Suppose $P$ is uniform over $C$ and consider any of the involutions $\phi_i$.

$$M_{q,P}(C \mid \mathbf{u}) := \inf \left\{ M \in \mathbb{R} \mid \underset{\mathbf{x} \sim P}{\mathbb{P}}\left( \mathbf{x}^\top \mathbf{u} > M \right) \leq q \right\} \tag{73}$$

$$= \inf \left\{ M \in \mathbb{R} \mid \underset{\mathbf{x} \sim P}{\mathbb{P}}\left( (\mathbf{P}_{\phi_i}\mathbf{x})^\top (\mathbf{P}_{\phi_i}\mathbf{u}) > M \right) \leq q \right\} \tag{74}$$

$$= \inf \left\{ M \in \mathbb{R} \mid \underset{\mathbf{x} \sim \phi_i \cdot P}{\mathbb{P}}\left( \mathbf{x}^\top (\mathbf{P}_{\phi_i}\mathbf{u}) > M \right) \leq q \right\} \tag{75}$$

$$= \inf \left\{ M \in \mathbb{R} \mid \underset{\mathbf{x} \sim P}{\mathbb{P}}\left( \mathbf{x}^\top (\mathbf{P}_{\phi_i}\mathbf{u}) > M \right) \leq q \right\} \tag{76}$$

$$=: M_{q,P}(\phi_i \cdot C \mid \phi_i \cdot \mathbf{u}). \tag{77}$$

Equation (74) follows by the orthogonality of permutation matrices. Equation (76) follows because if $\mathbf{x} \in \mathrm{supp}(P) = C$, then $\phi_i \cdot \mathbf{x} \in C = \mathrm{supp}(P)$, and furthermore $P(\mathbf{x}) = P(\mathbf{P}_{\phi_i}\mathbf{x})$ by uniformity.

Now we show the invariance of $C_{>M_{q,P}(C|\mathbf{u})}$ under joint permutation by $\phi_i$:

$$C_{>M_{q,P}(C|\mathbf{u})} := \left\{ \mathbf{c} \in C \mid \mathbf{c}^\top \mathbf{u} > M_{q,P}(C \mid \mathbf{u}) \right\} \tag{78}$$

$$= \left\{ \mathbf{c} \in C \mid (\mathbf{P}_{\phi_i}\mathbf{c})^\top (\mathbf{P}_{\phi_i}\mathbf{u}) > M_{q,P}(\phi_i \cdot C \mid \phi_i \cdot \mathbf{u}) \right\} \tag{79}$$

$$= \left\{ \mathbf{c} \in \phi_i \cdot C \mid \mathbf{c}^\top (\mathbf{P}_{\phi_i}\mathbf{u}) > M_{q,P}(\phi_i \cdot C \mid \phi_i \cdot \mathbf{u}) \right\} \tag{80}$$

$$=: C_{>M_{q,P}(\phi_i \cdot C|\phi_i \cdot \mathbf{u})}. \tag{81}$$

Equation (79) follows by the orthogonality of permutation matrices and because $M_{q,P}(C \mid \mathbf{u}) = M_{q,P}(\phi_i \cdot C \mid \phi_i \cdot \mathbf{u})$ by eq. (77). A similar proof shows that $C_{=M_{q,P}(C|\mathbf{u})} = C_{=M_{q,P}(\phi_i \cdot C|\phi_i \cdot \mathbf{u})}$.

Recall that

$$Q_{q,P}(X \mid C, \mathbf{u}) := \sum_{\mathbf{x} \in X} \frac{P(\mathbf{x})}{q} \left( \mathbb{1}_{\mathbf{x} \in C_{>M_{q,P}(C|\mathbf{u})}} + \frac{\mathbb{1}_{\mathbf{x} \in C_{=M_{q,P}(C|\mathbf{u})}}}{P\left( C_{=M_{q,P}(C|\mathbf{u})} \right)} \left( q - P\left( C_{>M_{q,P}(C|\mathbf{u})} \right) \right) \right). \tag{82}$$

$Q_{q,P}(X \mid C, \mathbf{u}) = Q_{q,P}(\phi_i \cdot X \mid \phi_i \cdot C, \phi_i \cdot \mathbf{u})$, since $Q$ is the sum of products of $\phi_i$-invariant quantities.

$P(\mathbf{x})$ is non-negative because $P$ is a probability distribution, and $q$ is assumed positive. The indicator functions $\mathbb{1}$ are non-negative. By the definition of $M_{q,P}$, $P\left(C_{>M_{q,P}(C|\mathbf{u})}\right) \leq q$. Therefore, eq. (82) is the sum of non-negative terms. Thus, if $X' \subseteq X$, then $Q_{q,P}(X' \mid C, \mathbf{u}) \leq Q_{q,P}(X \mid C, \mathbf{u})$.

Let $f(X \mid \mathbf{u}) := Q_{q,P}(X \mid C, \mathbf{u})$. Since $\phi_i \cdot C = C$ and since $Q_{q,P}(X \mid C, \mathbf{u}) = Q_{q,P}(\phi_i \cdot X \mid \phi_i \cdot C, \phi_i \cdot \mathbf{u})$, lemma B.10 shows that $f(X \mid \mathbf{u})$ is also jointly invariant to permutation by $\phi_i$. Lastly, if $X' \subseteq X$, we have $f(X' \mid \mathcal{D}_{\text{any}}) \leq f(X \mid \mathcal{D}_{\text{any}})$.

Apply lemma B.9 to conclude that $Q_{q,P}(B \mid C, \mathbf{u}) =: f(B \mid \mathbf{u}) \geq^n_{\text{most: } \mathbb{R}^d} f(A \mid \mathbf{u}) := Q_{q,P}(A \mid C, \mathbf{u})$. $\qquad\square$

**Conjecture B.13** (Orbit tendencies occur for more quantilizer base distributions)**.** Proposition A.11's item 7 holds for any base distribution $P$ over $C$ such that $\min_{\mathbf{b} \in B} P(\mathbf{b}) \geq \max_{\mathbf{a} \in A} P(\mathbf{a})$. Furthermore, $Q_{q,P}(X \mid C, \mathbf{u})$ is measurable on $\mathbf{u}$ and so $\geq^n_{\text{most: } \mathbb{R}^d}$ can be generalized to $\geq^n_{\text{most: } \mathfrak{D}_{\text{any}}}$.

## Appendix C  Detailed analyses of MR scenarios

### C.1  Action selection

Consider a bandit problem with five arms $a_1, \ldots, a_5$ partitioned $A := \{a_1\}$, $B := \{a_2, \ldots, a_5\}$, which each action has a definite utility $\mathbf{u}_i$. There are $T = 100$ trials. Suppose the training procedure train uses the $\epsilon$-greedy strategy to learn value estimates for each arm. At the end of training, train outputs a greedy policy with respect to its value estimates. Consider any action-value initialization, and the learning rate is set $\alpha := 1$. To learn an optimal policy, at worst, the agent just has to try each action once.

**Lemma C.1** (Lower bound on success probability of the train bandit)**.** *Let $\mathbf{u} \in \mathbb{R}^5$ assign strictly maximal utility to $a_i$, and suppose train (described above) runs for $T \geq 5$ trials. Then $p_{\text{train}}(\{a_i\} \mid \mathbf{u}) \geq 1 - (1 - \frac{\epsilon}{4})^T$.*

*Proof.* Since the trained policy can be stochastic,

$$p_{\text{train}}(\{a_i\} \mid \mathbf{u}) \geq \mathbb{P}\left(a_i \text{ is assigned probability 1 by the learned greedy policy}\right).$$

Since $a_i$ has strictly maximal utility which is deterministic, and since the learning rate $\alpha := 1$, if action $a_i$ is ever drawn, it is assigned probability 1 by the learned policy. The probability that $a_i$ is never explored is at most $(1 - \frac{\epsilon}{4})^T$, because at worst, $a_i$ is an "explore" action (and not an "exploit" action) at every time step, in which case it is ignored with probability $1 - \frac{\epsilon}{4}$. $\qquad\square$

**Proposition C.2** (The train bandit is 4-retargetable)**.** *$p_{\text{train}}$ is $(\mathbb{R}^5, A \xrightarrow{4} B)$-retargetable.*

*Proof.* Let $\phi_i := a_1 \leftrightarrow a_i$ for $i = 2, \ldots, 5$ and let $\Theta := \mathbb{R}^5$. We want to show that whenever $\mathbf{u} \in \mathbb{R}^5$ induces $p_{\text{train}}(A \mid \mathbf{u}) > p_{\text{train}}(B \mid \mathbf{u})$, retargeting $\mathbf{u}$ will get train to instead learn to pull a $B$-action: $p_{\text{train}}(A \mid \phi_i \cdot \mathbf{u}) < p_{\text{train}}(B \mid \phi_i \cdot \mathbf{u})$.

Suppose we have such a $\mathbf{u}$. If $\mathbf{u}$ is constant, a symmetry argument shows that each action has equal probability of being selected, in which case $p_{\text{train}}(A \mid \mathbf{u}) = \frac{1}{5} < \frac{4}{5} = p_{\text{train}}(B \mid \mathbf{u})$—a contradiction. Therefore, $\mathbf{u}$ is not constant. Similar symmetry arguments show that $A$'s action $a_1$ has strictly maximal utility ($\mathbf{u}_1 > \max_{i=2,\ldots,5} \mathbf{u}_i$).

But for $T = 100$, lemma C.1 shows that $p_{\text{train}}(A \mid \mathbf{u}) = p_{\text{train}}(\{a_1\} \mid \mathbf{u}) \approx 1$ and $p_{\text{train}}(\{a_{i \neq 1}\} \mid \mathbf{u}) \approx 0 \implies p_{\text{train}}(B \mid \mathbf{u}) = \sum_{i \neq 1} p_{\text{train}}(\{a_i\} \mid \mathbf{u}) \approx 0$. The converse statement holds when considering $\phi_i \cdot \mathbf{u}$ instead of $\mathbf{u}$. Therefore, train satisfies definition 3.5's item 1 (retargetability). These $\phi_i \cdot \mathbf{u} \in \Theta := \mathbb{R}^5$ because $\mathbb{R}^5$ is closed under permutation by $S_5$, satisfying item 2.

Consider another $\mathbf{u}' \in \mathbb{R}^5$ such that $p_{\text{train}}(A \mid \mathbf{u}') > p_{\text{train}}(B \mid \mathbf{u}')$, and consider $i \neq j$. By the above symmetry arguments, $\mathbf{u}'$ must also assign $a_1$ maximal utility. By lemma C.1, $p_{\text{train}}(\{a_i\} \mid \phi_i \cdot \mathbf{u}) \approx 1$ and $p_{\text{train}}(\{a_j\} \mid \phi_i \cdot \mathbf{u}) \approx 0$ since $i \neq j$, and vice versa when considering $\phi_j \cdot \mathbf{u}$ instead of $\phi_i \cdot \mathbf{u}$. Then since $\phi_i \cdot \mathbf{u}$ and $\phi_j \cdot \mathbf{u}$ induce distinct probability distributions over learned actions, they cannot be the same utility function. This satisfies item 3. $\qquad\square$

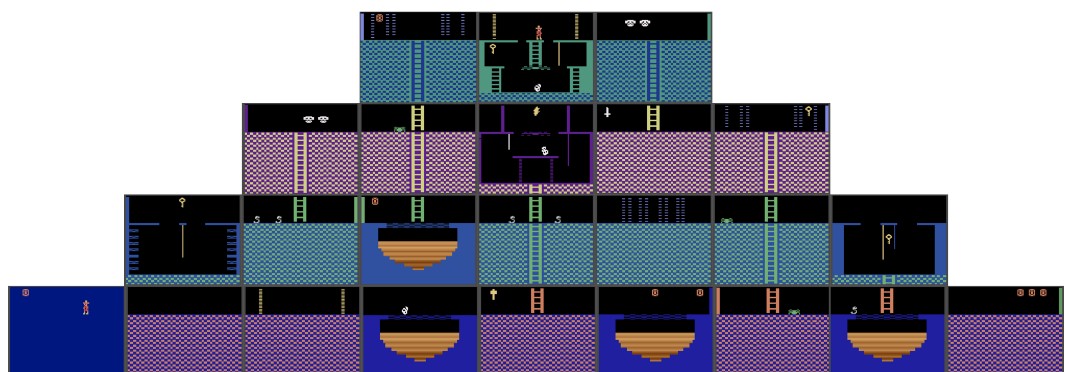

Figure 3: Map of the first level of Montezuma's Revenge.

**Corollary C.3** (The train bandit has orbit-level tendencies). $p_{train}(B \mid \mathbf{u}) \geq^4_{most: \ \mathbb{R}^5} p_{train}(A \mid \mathbf{u})$.

*Proof.* Combine proposition C.2 and theorem 3.6. □

## C.2 Observation reward maximization

Let $T$ be a reasonably long rollout length, so that $O_{T\text{-reach}}$ is large—many different step-$T$ observations can be induced.

**Proposition C.4** (Final reward maximization has strong orbit-level incentives in MR). *Let* $n := \lfloor \frac{|O_{leave}|}{|O_{stay}|} \rfloor$. $p_{max}(O_{leave} \mid R) \geq^n_{most: \ \mathbb{R}^O} p_{max}(O_{stay} \mid R)$.

*Proof.* Consider the vector space representation of observations, $\mathbb{R}^{|\mathcal{O}|}$. Define $A := \{\mathbf{e}_o \mid o \in O_{\text{stay}}\}, B := \{\mathbf{e}_o \mid o \in O_{\text{leave}}\}$, and $C := O_{T\text{-reach}} = A \cup B$ the union of $O_{\text{stay}}, O_{\text{leave}}$.

Since $|O_{\text{leave}}| \geq |O_{\text{stay}}|$ by assumption that $T$ is reasonably large, consider the involution $\phi_1 \in S_{|\mathcal{O}|}$ which embeds $O_{\text{stay}}$ into $O_{\text{leave}}$, while fixing all other observations. If possible, produce another involution $\phi_2$ which also embeds $O_{\text{stay}}$ into $O_{\text{leave}}$, which fixes all other observations, and which "doesn't interfere with $\phi_1$" (*i.e.* $\phi_2 \cdot (\phi_1 \cdot A) = \phi_1 \cdot A$). We can produce $n := \lfloor \frac{|O_{\text{leave}}|}{|O_{\text{stay}}|} \rfloor$ such involutions.

Therefore, $B$ contains $n$ copies (definition A.7) of $A$ via involutions $\phi_1, \ldots, \phi_n$. Furthermore, $\phi_i \cdot (A \cup B) = A \cup B$, since each $\phi_i$ swaps $A$ with $B' \subseteq B$, and fixes all $\mathbf{b} \in B \setminus B'$ by assumption. Thus, $\phi \cdot C = C$.

By proposition A.11's item 2, $\text{FracOptimal}(B \mid C, R) \geq^n_{most: \ \mathbb{R}^O} \text{FracOptimal}(A \mid C, R)$. Since $p_{\max}$ uniformly randomly chooses a maximal-reward observation to induce, $\forall X \subseteq C : p_{\max}(X \mid R) = \text{FracOptimal}(X \mid C, R)$. Therefore, $p_{\max}(O_{\text{leave}} \mid R) \geq^n_{most: \ \mathbb{R}^O} p_{\max}(O_{\text{stay}} \mid R)$. □

We want to reason about the probability that decide leaves the initial room by time $T$ in its rollout trajectories.

$$p_{\text{decide}}(\text{leave} \mid \theta) := \mathop{\mathbb{P}}_{\substack{\pi \sim \text{decide}(\theta), \\ \tau \sim \pi | s_0}} (\tau \text{ has left the first room by step } T), \qquad (83)$$

$$p_{\text{decide}}(\text{stay} \mid \theta) := \mathop{\mathbb{P}}_{\substack{\pi \sim \text{decide}(\theta), \\ \tau \sim \pi | s_0}} (\tau \text{ has not left the first room by step } T). \qquad (84)$$

We want to show that reward maximizers tend to leave the room: $p_{\max}(\text{leave} \mid R) \geq^n_{\text{most: }\Theta} p_{\max}(\text{stay} \mid R)$. However, we must be careful: In general, $p_{\max}(O_{\text{leave}} \mid R) \neq p_{\max}(\text{leave} \mid R)$ and $p_{\max}(O_{\text{stay}} \mid R) \neq p_{\max}(\text{stay} \mid R)$. For example, suppose that $o_T \in O_{\text{leave}}$. By the definition of $O_{\text{leave}}$, $o_T$ can only be observed if the agent has left the room by time step $T$, and so the trajectory $\tau$ must have left the first room. The converse argument does not hold: The agent could leave the first room, re-enter, and then wait until time $T$. Although one of the doors would have been opened (fig. 2), the agent can also open the door without leaving the room, and then realize the same step-$T$ observation. Therefore, this observation doesn't belong to $O_{\text{leave}}$.

**Lemma C.5** (Room-status inequalities for MR).

$$p_{\text{decide}}(\textit{stay} \mid \theta) \leq p_{\text{decide}}(O_{\textit{stay}} \mid \theta), \tag{85}$$

$$\text{and } p_{\text{decide}}(O_{\textit{leave}} \mid \theta) \leq p_{\text{decide}}(\textit{leave} \mid \theta). \tag{86}$$

*Proof.* For any decide,

$$p_{\text{decide}}(\text{stay} \mid \theta) \tag{87}$$

$$= \mathop{\mathbb{P}}_{\substack{\pi \sim \text{decide}(\theta), \\ \tau \sim \pi \mid s_0}} (\tau \text{ stays through step } T) \tag{88}$$

$$= \sum_{o \in \mathcal{O}} \mathop{\mathbb{P}}_{\substack{\pi \sim \text{decide}(\theta), \\ \tau \sim \pi \mid s_0}} (o \text{ at step } T \text{ of } \tau) \mathop{\mathbb{P}}_{\substack{\pi \sim \text{decide}(\theta), \\ \tau \sim \pi \mid s_0}} (\tau \text{ stays} \mid o \text{ at step } T) \tag{89}$$

$$= \sum_{o \in O_{T\text{-reach}}} \mathop{\mathbb{P}}_{\substack{\pi \sim \text{decide}(\theta), \\ \tau \sim \pi \mid s_0}} (o \text{ at step } T) \mathop{\mathbb{P}}_{\substack{\pi \sim \text{decide}(\theta), \\ \tau \sim \pi \mid s_0}} (\tau \text{ stays} \mid o \text{ at step } T) \tag{90}$$

$$= \sum_{o \in O_{\text{stay}}} \mathop{\mathbb{P}}_{\substack{\pi \sim \text{decide}(\theta), \\ \tau \sim \pi \mid s_0}} (o \text{ at step } T) \mathop{\mathbb{P}}_{\substack{\pi \sim \text{decide}(\theta), \\ \tau \sim \pi \mid s_0}} (\tau \text{ stays} \mid o \text{ at step } T) \tag{91}$$

$$\leq \sum_{o \in O_{\text{stay}}} \mathop{\mathbb{P}}_{\substack{\pi \sim \text{decide}(\theta), \\ \tau \sim \pi \mid s_0}} (o \text{ at step } T) \tag{92}$$

$$= \mathop{\mathbb{P}}_{\substack{\pi \sim \text{decide}(\theta), \\ \tau \sim \pi \mid s_0}} (o_T \in O_{\text{stay}}) \tag{93}$$

$$=: p_{\text{decide}}(O_{\text{stay}} \mid \theta). \tag{94}$$

Equation (90) holds because the definition of $O_{T\text{-reach}}$ ensures that if $o \notin O_{T\text{-reach}}$, then $\mathbb{P}_{\substack{\pi \sim \text{decide}(\theta), \\ \tau \sim \pi \mid s_0}}(o \mid \theta) = 0$. Because $o \in O_{T\text{-reach}} \setminus O_{\text{stay}}$ implies that $\tau$ left and so

$$\mathop{\mathbb{P}}_{\substack{\pi \sim \text{decide}(\theta), \\ \tau \sim \pi \mid s_0}} (\tau \text{ stays} \mid o \text{ at step } T) = 0,$$

eq. (91) follows. Then we have shown eq. (85).

For eq. (86),

$$p_{\text{decide}}(O_{\text{leave}} \mid \theta) \tag{95}$$

$$:= \mathop{\mathbb{P}}_{\substack{\pi \sim \text{decide}(\theta), \\ \tau \sim \pi \mid s_0}} (o_T \in O_{\text{leave}}) \tag{96}$$

$$= \sum_{o \in O_{\text{leave}}} \mathop{\mathbb{P}}_{\substack{\pi \sim \text{decide}(\theta), \\ \tau \sim \pi \mid s_0}} (o \text{ at step } T) \tag{97}$$

$$= \sum_{o \in O_{\text{leave}}} \mathop{\mathbb{P}}_{\substack{\pi \sim \text{decide}(\theta), \\ \tau \sim \pi \mid s_0}} (o \text{ at step } T) \mathop{\mathbb{P}}_{\substack{\pi \sim \text{decide}(\theta), \\ \tau \sim \pi \mid s_0}} (\tau \text{ leaves by step } T \mid o \text{ at step } T) \tag{98}$$

$$= \sum_{o \in \mathcal{O}} \mathop{\mathbb{P}}_{\substack{\pi \sim \text{decide}(\theta), \\ \tau \sim \pi \mid s_0}} (o \text{ at step } T) \mathop{\mathbb{P}}_{\substack{\pi \sim \text{decide}(\theta), \\ \tau \sim \pi \mid s_0}} (\tau \text{ leaves by step } T \mid o \text{ at step } T) \tag{99}$$

$$= \mathop{\mathbb{P}}_{\substack{\pi \sim \text{decide}(\theta), \\ \tau \sim \pi \mid s_0}} (\tau \text{ has left the first room by step } T) \tag{100}$$

$$=: p_{\text{decide}}(\text{leave} \mid \theta). \tag{101}$$

Equation (98) follows because, since $o \in O_{\text{leave}}$ are only realizable by leaving the first room, this implies $\mathbb{P}_{\substack{\pi \sim \text{decide}(\theta), \\ \tau \sim \pi \mid s_0}} (\tau \text{ leaves by step } T \mid o \text{ at step } T) = 1$. Equation (99) follows because $O_{\text{leave}} \subseteq \mathcal{O}$, and probabilities are non-negative. Then we have shown eq. (86). $\qquad \square$

**Corollary C.6** (Final reward maximizers tend to leave the first room in MR)**.**

$$p_{\max}(\textit{leave} \mid R) \geq^n_{\text{most: } \mathbb{R}^{\mathcal{O}}} p_{\max}(\textit{stay} \mid R). \tag{102}$$

*Proof.* Using lemma C.5 and proposition C.4, apply lemma B.1 with $f_0(R) := p_{\max}(\text{leave} \mid R), f_1(R) := p_{\max}(O_{\text{leave}} \mid R), f_2(R) := p_{\max}(O_{\text{stay}} \mid R), f_3(R) := p_{\max}(\text{stay} \mid R)$ to conclude that $p_{\max}(\text{leave} \mid R) \geq^n_{\text{most: } \mathbb{R}^{\mathcal{O}}} p_{\max}(\text{stay} \mid R)$. $\qquad \square$

### C.3 Featurized reward maximization

$\Theta := \mathbb{R}^{\mathcal{O}}$ assumes we will specify complicated reward functions over observations, with $|\mathcal{O}|$ degrees of freedom in their specification. Any observation can get any number. However, reward functions are often specified more compactly. For example, in section 4.3, the (additively) featurized reward function $R_{\text{feat}}(o_T) := \text{feat}(o_T)^\top \alpha$ has four degrees of freedom. Compared to typical reward functions (which would look like "random noise" to a human), $R_{\text{feat}}$ more easily trains competent policies because of the regularities between the reward and the state features.

In this setup, $p_{\max}$ chooses a policy which induces a step-$T$ observation with maximal reward. Reward depends only on the feature vector of the final observation—more specifically, on the agent's item counts. There are more possible item counts available by first leaving the room, than by staying.

We will now conduct a more detailed analysis and conclude that $p_{\max}(O_{\text{leave}} \mid \alpha) \geq^3_{\text{most: } \mathbb{R}^4} p_{\max}(O_{\text{stay}} \mid \alpha)$. Informally, we can retarget which items the agent prioritizes, and thereby retarget from $O_{\text{stay}}$ to $O_{\text{leave}}$.

Consider the featurization function which takes as input an observation $o \in \mathcal{O}$:

$$\text{feat}(o) := \begin{pmatrix} \text{\# of keys in inventory shown by } o \\ \text{\# of swords in inventory shown by } o \\ \text{\# of torches in inventory shown by } o \\ \text{\# of amulets in inventory shown by } o \end{pmatrix}. \tag{103}$$

Consider $A_{\text{feat}} := \{\text{feat}(o) \mid o \in O_{\text{stay}}\}, B_{\text{feat}} := \{\text{feat}(o) \mid o \in O_{\text{leave}}\}$.

Let $\mathbf{e}_i \in \mathbb{R}^4$ be the standard basis vector with a $1$ in entry $i$ and $0$ elsewhere. When restricted to the room shown in fig. 2, the agent can either acquire the key in the first room and retain it until step $T$ ($\mathbf{e}_1$), or reach time step $T$ empty-handed ($\mathbf{0}$). We conclude that $A_{\text{feat}} = \{\mathbf{e}_1, \mathbf{0}\}$.

For $B_{\text{feat}}$, recall that in section 4.2 we assumed the rollout length $T$ to be reasonably large. Then by leaving the room, some realizable trajectory induces $o_T$ displaying an inventory containing only a sword ($\mathbf{e}_2$), or only a torch ($\mathbf{e}_3$), or only an amulet ($\mathbf{e}_4$), or nothing at all ($\mathbf{0}$). Therefore, $\{\mathbf{e}_2, \mathbf{e}_3, \mathbf{e}_4, \mathbf{0}\} \subseteq B_{\text{feat}}$. $B_{\text{feat}}$ contains 3 copies of $A_{\text{feat}}$ (definition A.7) via involutions $\phi_i : 1 \leftrightarrow i$, $i \neq 1$. Suppose all feature coefficient vectors $\alpha \in \mathbb{R}^4$ are plausible. Then $\Theta := \mathbb{R}^4$.

Let us be more specific about what is entailed by featurized reward maximization. The $\text{decide}_{\max}(\alpha)$ procedure takes $\alpha$ as input and then considers the reward function $o \mapsto \text{feat}(o)^\top \alpha$. Then, $\text{decide}_{\max}$ uniformly randomly chooses an observation $o_T \in O_{T\text{-reach}}$ which maximizes this featurized reward, and then uniformly randomly chooses a policy which implements $o_T$.

**Lemma C.7** (FracOptimal inequalities)**.** *Let $X \subseteq Y' \subseteq Y \subsetneq \mathbb{R}^d$ be finite, and let $\mathbf{u} \in \mathbb{R}^d$. Then*

$$\text{FracOptimal}\left(X \mid Y, \mathbf{u}\right) \leq \text{FracOptimal}\left(X \mid Y', \mathbf{u}\right) \leq \text{FracOptimal}\left(X \cup (Y \setminus Y') \mid Y, \mathbf{u}\right). \tag{104}$$

*Proof.* For finite $X_1 \subsetneq \mathbb{R}^d$, let $\text{Best}\left(X_1 \mid \mathbf{u}\right) := \arg\max_{\mathbf{x}_1 \in X_1} \mathbf{x}_1^\top \mathbf{u}$. Suppose $\mathbf{y}' \in \text{Best}\left(Y' \mid \mathbf{u}\right)$, but $\mathbf{y}' \notin \text{Best}\left(Y \mid \mathbf{u}\right)$. Then for all $\mathbf{a} \in \text{Best}\left(Y' \mid \mathbf{u}\right)$,

$$\mathbf{a}^\top \mathbf{u} = \mathbf{y}'^\top \mathbf{u} < \max_{\mathbf{y} \in Y} \mathbf{y}^\top \mathbf{u}. \tag{105}$$

So $\mathbf{a} \notin \mathrm{Best}\left(Y \mid \mathbf{u}\right)$. Then either $\mathrm{Best}\left(Y' \mid \mathbf{u}\right) \subseteq \mathrm{Best}\left(Y \mid \mathbf{u}\right)$, or the two sets are disjoint.

$$\mathrm{FracOptimal}\left(X \mid Y, \mathbf{u}\right) := \frac{\left|\mathrm{Best}\left(Y \mid \mathbf{u}\right) \cap X\right|}{\left|\mathrm{Best}\left(Y \mid \mathbf{u}\right)\right|} \tag{106}$$

$$\leq \frac{\left|\mathrm{Best}\left(Y' \mid \mathbf{u}\right) \cap X\right|}{\left|\mathrm{Best}\left(Y' \mid \mathbf{u}\right)\right|} =: \mathrm{FracOptimal}\left(X \mid Y', \mathbf{u}\right) \tag{107}$$

If $\mathrm{Best}\left(Y' \mid \mathbf{u}\right) \subseteq \mathrm{Best}\left(Y \mid \mathbf{u}\right)$, then since $X \subseteq Y'$, we have $X \cap \mathrm{Best}\left(Y' \mid \mathbf{u}\right) = X \cap \mathrm{Best}\left(Y \mid \mathbf{u}\right)$. Then in this case, eq. (106) has equal numerator and larger denominator than eq. (107). On the other hand, if $\mathrm{Best}\left(Y' \mid \mathbf{u}\right) \cap \mathrm{Best}\left(Y \mid \mathbf{u}\right) = \varnothing$, then since $X \subseteq Y'$, $X \cap \mathrm{Best}\left(Y \mid \mathbf{u}\right) = \varnothing$. Then eq. (106) equals $0$, and eq. (107) is non-negative. Either way, eq. (107)'s inequality holds. To show the second inequality, we handle the two cases separately.

**Subset case.** Suppose that $\mathrm{Best}\left(Y' \mid \mathbf{u}\right) \subseteq \mathrm{Best}\left(Y \mid \mathbf{u}\right)$.

$$\frac{\left|\mathrm{Best}\left(Y' \mid \mathbf{u}\right) \cap X\right|}{\left|\mathrm{Best}\left(Y' \mid \mathbf{u}\right)\right|} \leq \frac{\left|\mathrm{Best}\left(Y' \mid \mathbf{u}\right) \cap X\right| + \left|\mathrm{Best}\left(Y \setminus Y' \mid \mathbf{u}\right)\right|}{\left|\mathrm{Best}\left(Y' \mid \mathbf{u}\right)\right| + \left|\mathrm{Best}\left(Y \setminus Y' \mid \mathbf{u}\right)\right|} \tag{108}$$

$$= \frac{\left|\mathrm{Best}\left(Y' \mid \mathbf{u}\right) \cap X\right| + \left|\mathrm{Best}\left(Y \setminus Y' \mid \mathbf{u}\right) \cap \left(Y \setminus Y'\right)\right|}{\left|\mathrm{Best}\left(Y' \mid \mathbf{u}\right)\right| + \left|\mathrm{Best}\left(Y \setminus Y' \mid \mathbf{u}\right)\right|} \tag{109}$$

$$= \frac{\left|\mathrm{Best}\left(Y' \mid \mathbf{u}\right) \cap X\right| + \left|\mathrm{Best}\left(Y \mid \mathbf{u}\right) \cap \left(Y \setminus Y'\right)\right|}{\left|\mathrm{Best}\left(Y' \mid \mathbf{u}\right)\right| + \left|\mathrm{Best}\left(Y \setminus Y' \mid \mathbf{u}\right)\right|} \tag{110}$$

$$= \frac{\left|\mathrm{Best}\left(Y' \mid \mathbf{u}\right) \cap X\right| + \left|\mathrm{Best}\left(Y \mid \mathbf{u}\right) \cap \left(Y \setminus Y'\right)\right|}{\left|\mathrm{Best}\left(Y \mid \mathbf{u}\right)\right|} \tag{111}$$

$$= \frac{\left|\mathrm{Best}\left(Y \mid \mathbf{u}\right) \cap X\right| + \left|\mathrm{Best}\left(Y \mid \mathbf{u}\right) \cap \left(Y \setminus Y'\right)\right|}{\left|\mathrm{Best}\left(Y \mid \mathbf{u}\right)\right|} \tag{112}$$

$$= \frac{\left|\mathrm{Best}\left(Y \mid \mathbf{u}\right) \cap \left(X \cup \left(Y \setminus Y'\right)\right)\right|}{\left|\mathrm{Best}\left(Y \mid \mathbf{u}\right)\right|} \tag{113}$$

$$=: \mathrm{FracOptimal}\left(X \cup \left(Y \setminus Y'\right) \mid Y, \mathbf{u}\right). \tag{114}$$

Equation (108) follows because when $n \leq d, k \geq 0$, we have $\frac{n}{d} \leq \frac{n+k}{d+k}$. For eq. (110), since $\mathrm{Best}\left(Y' \mid \mathbf{u}\right) \subseteq \mathrm{Best}\left(Y \mid \mathbf{u}\right)$, we must have

$$\mathrm{Best}\left(Y \mid \mathbf{u}\right) = \mathrm{Best}\left(Y \setminus Y' \mid \mathbf{u}\right) \cup \mathrm{Best}\left(Y' \mid \mathbf{u}\right).$$

But then

$$\mathrm{Best}\left(Y \mid \mathbf{u}\right) \cap \left(Y \setminus Y'\right) = \left(\mathrm{Best}\left(Y \setminus Y' \mid \mathbf{u}\right) \cap \left(Y \setminus Y'\right)\right) \cup \left(\mathrm{Best}\left(Y' \mid \mathbf{u}\right) \cap \left(Y \setminus Y'\right)\right) \tag{115}$$

$$= \mathrm{Best}\left(Y \setminus Y' \mid \mathbf{u}\right) \cap \left(Y \setminus Y'\right). \tag{116}$$

Then eq. (110) follows. Equation (111) follows since

$$\mathrm{Best}\left(Y \mid \mathbf{u}\right) = \mathrm{Best}\left(Y \setminus Y' \mid \mathbf{u}\right) \cup \mathrm{Best}\left(Y' \mid \mathbf{u}\right).$$

Equation (112) follows since $X \subseteq Y'$, and so

$$\mathrm{Best}\left(Y' \mid \mathbf{u}\right) \cap X = \mathrm{Best}\left(Y \mid \mathbf{u}\right) \cap X.$$

Equation (113) follows because $X \subseteq Y'$ is disjoint of $Y \setminus Y'$. We have shown that

$$\mathrm{FracOptimal}\left(X \mid Y', \mathbf{u}\right) \leq \mathrm{FracOptimal}\left(X \cup \left(Y \setminus Y'\right) \mid Y, \mathbf{u}\right)$$

in this case.

**Disjoint case.** Suppose that $\mathrm{Best}\left(Y' \mid \mathbf{u}\right) \cap \mathrm{Best}\left(Y \mid \mathbf{u}\right) = \varnothing$.

$$\frac{\left|\mathrm{Best}\left(Y' \mid \mathbf{u}\right) \cap X\right|}{\left|\mathrm{Best}\left(Y' \mid \mathbf{u}\right)\right|} \leq 1 \tag{117}$$

$$= \frac{\left|\mathrm{Best}\left(Y \setminus Y' \mid \mathbf{u}\right)\right|}{\left|\mathrm{Best}\left(Y \setminus Y' \mid \mathbf{u}\right)\right|} \tag{118}$$

$$= \frac{\left|\mathrm{Best}\left(Y \setminus Y' \mid \mathbf{u}\right) \cap (Y \setminus Y')\right|}{\left|\mathrm{Best}\left(Y \setminus Y' \mid \mathbf{u}\right)\right|} \tag{119}$$

$$= \frac{\left|\mathrm{Best}\left(Y \setminus Y' \mid \mathbf{u}\right) \cap (X \cup (Y \setminus Y'))\right|}{\left|\mathrm{Best}\left(Y \setminus Y' \mid \mathbf{u}\right)\right|} \tag{120}$$

$$= \frac{\left|\mathrm{Best}\left(Y \mid \mathbf{u}\right) \cap (X \cup (Y \setminus Y'))\right|}{\left|\mathrm{Best}\left(Y \mid \mathbf{u}\right)\right|} \tag{121}$$

$$=: \mathrm{FracOptimal}\left(X \cup (Y \setminus Y') \mid Y, \mathbf{u}\right). \tag{122}$$

Equation (117) follows because $\mathrm{Best}\left(Y' \mid \mathbf{u}\right) \cap X \subseteq \mathrm{Best}\left(Y' \mid \mathbf{u}\right)$. For eq. (120), note that we trivially have $\mathrm{Best}\left(Y' \mid \mathbf{u}\right) \cap \mathrm{Best}\left(Y \setminus Y' \mid \mathbf{u}\right) = \varnothing$, and also that $X \subseteq Y'$. Therefore, $\mathrm{Best}\left(Y \setminus Y' \mid \mathbf{u}\right) \cap X = \varnothing$, and eq. (120) follows. Finally, the disjointness assumption implies that

$$\max_{\mathbf{y}' \in Y'} \mathbf{y'}^{\top} \mathbf{u} < \max_{\mathbf{y} \in Y} \mathbf{y}^{\top} \mathbf{u}.$$

Therefore, the optimal elements of $Y$ must come exclusively from $Y \setminus Y'$; *i.e.* $\mathrm{Best}\left(Y \mid \mathbf{u}\right) = \mathrm{Best}\left(Y \setminus Y' \mid \mathbf{u}\right)$. Then eq. (121) follows, and we have shown that

$$\mathrm{FracOptimal}\left(X \mid Y', \mathbf{u}\right) \leq \mathrm{FracOptimal}\left(X \cup (Y \setminus Y') \mid Y, \mathbf{u}\right)$$

in this case. $\qquad\square$

**Conjecture C.8** (Generalizing lemma C.7). Lemma C.7 and Turner et al. [2021]'s Lemma E.26 have extremely similar functional forms. How can they be unified?

**Proposition C.9** (Featurized reward maximizers tend to leave the first room in MR).

$$p_{\max}(leave \mid \alpha) \geq^{3}_{\mathrm{most:}\ \mathbb{R}^4} p_{\max}(stay \mid \alpha). \tag{123}$$

*Proof.* We want to show that $p_{\max}(O_{\mathrm{leave}} \mid \alpha) \geq^{n}_{\mathrm{most:}\ \mathbb{R}^4} p_{\max}(O_{\mathrm{stay}} \mid \alpha)$. Recall that $A_{\mathrm{feat}} = \{\mathbf{e}_1, \mathbf{0}\}, B'_{\mathrm{feat}} := \{\mathbf{e}_2, \mathbf{e}_3, \mathbf{e}_4\} \subseteq B_{\mathrm{feat}}$.

$$p_{\max}(\mathrm{stay} \mid \alpha) \tag{124}$$

$$\leq p_{\max}(O_{\mathrm{stay}} \mid \alpha) \tag{125}$$

$$:= \mathbb{P}_{\substack{\pi \sim \mathrm{decide}_{\max}(\alpha), \\ \tau \sim \pi \mid s_0}} \left(o_T \in O_{\mathrm{stay}}\right) \tag{126}$$

$$= \mathbb{P}_{\substack{\pi \sim \mathrm{decide}_{\max}(\alpha), \\ \tau \sim \pi \mid s_0}} \left(o_T \in O_{\mathrm{stay}}, \mathrm{feat}(o_T) \neq \mathbf{0}\right) + \mathbb{P}_{\substack{\pi \sim \mathrm{decide}_{\max}(\alpha), \\ \tau \sim \pi \mid s_0}} \left(o_T \in O_{\mathrm{stay}}, \mathrm{feat}(o_T) = \mathbf{0}\right) \tag{127}$$

$$\leq \mathrm{FracOptimal}\left(\{\mathbf{e}_1\} \mid C_{\mathrm{feat}}, \alpha\right) + \mathbb{P}_{\substack{\pi \sim \mathrm{decide}_{\max}(\alpha), \\ \tau \sim \pi \mid s_0}} \left(o_T \in O_{\mathrm{stay}}, \mathrm{feat}(o_T) = \mathbf{0}\right) \tag{128}$$

$$\leq \mathrm{FracOptimal}\left(\{\mathbf{e}_1\} \mid \{\mathbf{e}_1, \mathbf{e}_2, \mathbf{e}_3, \mathbf{e}_4\}, \alpha\right) + \mathbb{P}_{\substack{\pi \sim \mathrm{decide}_{\max}(\alpha), \\ \tau \sim \pi \mid s_0}} \left(o_T \in O_{\mathrm{stay}}, \mathrm{feat}(o_T) = \mathbf{0}\right) \tag{129}$$

$$\leq^3_{\text{most: }\mathbb{R}^4_{>0}} \text{FracOptimal}\left(\{\mathbf{e}_2, \mathbf{e}_3, \mathbf{e}_4\} \mid \{\mathbf{e}_1, \mathbf{e}_2, \mathbf{e}_3, \mathbf{e}_4\}, \alpha\right)$$

$$+ \underset{\substack{\pi \sim \text{decide}_{\max}(\alpha), \\ \tau \sim \pi | s_0}}{\mathbb{P}} \left(o_T \in O_{\text{leave}}, \text{feat}(o_T) = \mathbf{0}\right) \quad (130)$$

$$\leq \text{FracOptimal}\left(\{\mathbf{e}_2, \mathbf{e}_3, \mathbf{e}_4\} \cup \left(C_{\text{feat}} \setminus \{\mathbf{e}_1, \mathbf{e}_2, \mathbf{e}_3, \mathbf{e}_4\}\right) \mid C_{\text{feat}}, \alpha\right) \quad (131)$$

$$= \text{FracOptimal}\left(C_{\text{feat}} \setminus \{\mathbf{e}_1\} \mid C_{\text{feat}}, \alpha\right) \quad (132)$$

$$\leq \underset{\substack{\pi \sim \text{decide}_{\max}(\alpha), \\ \tau \sim \pi | s_0}}{\mathbb{P}} \left(o_T \in O_{\text{leave}}\right) \quad (133)$$

$$=: p_{\max}(O_{\text{leave}} \mid \alpha) \quad (134)$$

$$\leq p_{\max}(\text{leave} \mid \alpha). \quad (135)$$

Equation (124) and eq. (135) hold by lemma C.5. If $o_T \in O_{\text{stay}}$ is realized by $p_{\max}$ and $\text{feat}(o_T) \neq \mathbf{0}$, then we must have $\text{feat}(o_T) = \{\mathbf{e}_1\}$ be optimal and so the $\mathbf{e}_1$ inventory configuration is realized. Therefore, eq. (128) follows. Equation (129) follows by applying the first inequality of lemma C.7 with $X := \{\mathbf{e}_1\}, Y' := \{\mathbf{e}_1, \mathbf{e}_2, \mathbf{e}_3, \mathbf{e}_4\}, Y := C_{\text{feat}}$.

By applying proposition A.11's item 2 with $A := A_{\text{feat}} = \{\mathbf{e}_1\}, B' := B'_{\text{feat}} = \{\mathbf{e}_2, \mathbf{e}_3, \mathbf{e}_4\}, C := A \cup B'$, we have

$$\text{FracOptimal}\left(\{\mathbf{e}_1\} \mid \{\mathbf{e}_1, \mathbf{e}_2, \mathbf{e}_3, \mathbf{e}_4\}, \alpha\right) \leq^3_{\text{most: }\mathbb{R}^4_{>0}}$$

$$\text{FracOptimal}\left(\{\mathbf{e}_2, \mathbf{e}_3, \mathbf{e}_4\} \mid \{\mathbf{e}_1, \mathbf{e}_2, \mathbf{e}_3, \mathbf{e}_4\}, \alpha\right). \quad (136)$$

Furthermore, observe that

$$\underset{\substack{\pi \sim \text{decide}_{\max}(\alpha), \\ \tau \sim \pi | s_0}}{\mathbb{P}} \left(o_T \in O_{\text{stay}}, \text{feat}(o_T) = \mathbf{0}\right) \leq \underset{\substack{\pi \sim \text{decide}_{\max}(\alpha), \\ \tau \sim \pi | s_0}}{\mathbb{P}} \left(o_T \in O_{\text{leave}}, \text{feat}(o_T) = \mathbf{0}\right) \quad (137)$$

because either $\mathbf{0}$ is not optimal (in which case both sides equal 0), or else $\mathbf{0}$ is optimal, in which case the right side is strictly greater. This can be seen by considering how $\text{decide}_{\max}(\alpha)$ uniformly randomly chooses an observation in which the agent ends up with an empty inventory. As argued previously, the vast majority of such observations can only be induced by leaving the first room.

Combining eq. (136) and eq. (137), eq. (130) follows. Equation (131) follows by applying the second inequality of lemma C.7 with $X := \{\mathbf{e}_2, \mathbf{e}_3, \mathbf{e}_4\}, Y' := \{\mathbf{e}_1, \mathbf{e}_2, \mathbf{e}_3, \mathbf{e}_4\}, Y := C_{\text{feat}}$. If $\text{feat}(o_T) \in B_{\text{feat}}$ is realized by $p_{\max}$, then by the definition of $B_{\text{feat}}, o_T \in O_{\text{leave}}$ is realized, and so eq. (133) follows.

Then by applying lemma B.1 with

$$f_0(\alpha) := p_{\max}(\text{leave} \mid \alpha), \quad (138)$$

$$f_1(\alpha) := \text{FracOptimal}\left(\{\mathbf{e}_1\} \mid \{\mathbf{e}_1, \mathbf{e}_2, \mathbf{e}_3, \mathbf{e}_4\}, \alpha\right), \quad (139)$$

$$f_2(\alpha) := \text{FracOptimal}\left(\{\mathbf{e}_2, \mathbf{e}_3, \mathbf{e}_4\} \mid \{\mathbf{e}_1, \mathbf{e}_2, \mathbf{e}_3, \mathbf{e}_4\}, \alpha\right), \quad (140)$$

$$f_3(\alpha) := p_{\max}(\text{stay} \mid \alpha), \quad (141)$$

we conclude that $p_{\max}(\text{leave} \mid \alpha) \geq^3_{\text{most: }\mathbb{R}^4_{>0}} p_{\max}(\text{stay} \mid \alpha)$. $\qquad\square$

Lastly, note that if $\mathbf{0} \in \Theta$ and $f(A \mid \mathbf{0}) > f(B \mid \mathbf{0})$, $f$ cannot be even be simply retargetable for the $\Theta$ parameter set. This is because $\forall \phi \in S_d, \phi \cdot \mathbf{0} = \mathbf{0}$. For example, inductive bias ensures that, absent a reward signal, learned policies tend to stay in the initial room in MR. This is one reason why section 4.3's analysis of the policy tendencies of reinforcement learning excludes the all-zero reward function.

## C.4 Reasoning for why DQN can't explore well

In section 4.3, we wrote:

> Mnih et al. [2015]'s DQN isn't good enough to train policies which leave the first room of MR, and so DQN (trivially) cannot be retargetable *away* from the first room

via the reward function. There isn't a single featurized reward function for which DQN visits other rooms, and so we can't have $\alpha$ such that $\phi \cdot \alpha$ retargets the agent to $O_{\text{leave}}$. DQN isn't good enough at exploring.

We infer this is true from Nair et al. [2015], which shows that vanilla DQN gets zero score in MR. Thus, DQN never even gets the first key. Thus, DQN only experiences state-action-state transitions which didn't involve acquiring an item, since (as shown in fig. 3) the other items are outside of the first room, which requires a key to exit. In our analysis, we considered a reward function which is featurized over item acquisition.

Therefore, for all pre-key-acquisition state-action-state transitions, the featurized reward function returns exactly the same reward signals as those returned in training during the published experiments (namely, zero, because DQN can never even get to the key in order to receive a reward signal). That is, since DQN only experiences state-action-state transitions which didn't involve acquiring an item, and the featurized reward functions only reward acquiring an item, it doesn't matter what reward values are provided upon item acquisition—DQN's trained behavior will be the same. Thus, a DQN agent trained on any featurized reward function will not explore outside of the first room.

## Appendix D  Lower bounds on MDP power-seeking incentives for optimal policies

Turner et al. [2021] prove conditions under which *at least half* of the orbit of every reward function incentivizes power-seeking behavior. For example, in fig. 4, they prove that avoiding $\varnothing$ maximizes average per-timestep reward for at least half of reward functions. Roughly, there are more self-loop states ($\varnothing$, $\ell_\swarrow$, $r_\searrow$, $r_\nearrow$) available if the agent goes `left` or `right` instead of up towards $\varnothing$. We strengthen this claim, with corollary D.12 showing that for *at least three-quarters* of the orbit of every reward function, it is average-optimal to avoid $\varnothing$.

Therefore, we answer Turner et al. [2021]'s open question of whether increased number of environmental symmetries quantitatively strengthens the degree to which power-seeking is incentivized. The answer is *yes*. In particular, it may be the case that only one in a million state-based reward functions makes it average-optimal for Pac-Man to die immediately.

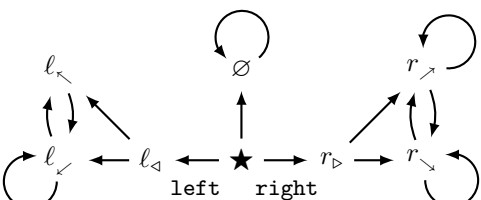

Figure 4: A toy MDP for reasoning about power-seeking tendencies. *Reproduced from Turner et al. [2021].*

We will briefly restate several definitions needed for our key results, theorem D.11 and corollary D.12. For explanation, see Turner et al. [2021].

**Definition D.1** (Non-dominated linear functionals). Let $X \subsetneq \mathbb{R}^{|\mathcal{S}|}$ be finite. $\text{ND}(X) :=$ $\left\{ \mathbf{x} \in X \mid \exists \mathbf{r} \in \mathbb{R}^{|\mathcal{S}|} : \mathbf{x}^\top \mathbf{r} > \max_{\mathbf{x}' \in X \setminus \{\mathbf{x}\}} \mathbf{x}'^\top \mathbf{r} \right\}$.

**Definition D.2** (Bounded reward function distribution). $\mathfrak{D}_{\text{bound}}$ is the set of bounded-support probability distributions $\mathcal{D}_{\text{bound}}$.

**Remark.** When $n = 1$, lemma D.3 reduces to the first part of Turner et al. [2021]'s lemma E.24, and lemma D.5 reduces to the first part of Turner et al. [2021]'s lemma E.28.

**Lemma D.3** (Quantitative expectation superiority lemma). *Let $A, B \subsetneq \mathbb{R}^d$ be finite and let $g : \mathbb{R} \to \mathbb{R}$ be a (total) increasing function. Suppose $B$ contains $n$ copies of $\text{ND}(A)$. Then*

$$\mathop{\mathbb{E}}_{\mathbf{r} \sim \mathcal{D}_{bound}} \left[ g\left( \max_{\mathbf{b} \in B} \mathbf{b}^\top \mathbf{r} \right) \right] \geq^n_{\text{most}: \mathfrak{D}_{bound}} \mathop{\mathbb{E}}_{\mathbf{r} \sim \mathcal{D}_{bound}} \left[ g\left( \max_{\mathbf{a} \in A} \mathbf{a}^\top \mathbf{r} \right) \right]. \tag{142}$$

*Proof.* Because $g : \mathbb{R} \to \mathbb{R}$ is increasing, it is measurable (as is $\max$).

Let $L := \inf_{\mathbf{r} \in \text{supp}(\mathcal{D}_{\text{bound}})} \max_{\mathbf{x} \in X} \mathbf{x}^\top \mathbf{r}, U := \sup_{\mathbf{r} \in \text{supp}(\mathcal{D}_{\text{bound}})} \max_{\mathbf{x} \in X} \mathbf{x}^\top \mathbf{r}$. Both exist because $\mathcal{D}_{\text{bound}}$ has bounded support. Furthermore, since $g$ is monotone increasing, it is bounded $[g(L), g(U)]$ on $[L, U]$. Therefore, $g$ is measurable and bounded on each $\text{supp}(\mathcal{D}_{\text{bound}})$, and so the relevant expectations exist for all $\mathcal{D}_{\text{bound}}$.

For finite $X \subsetneq \mathbb{R}^d$, let $f(X \mid \mathbf{u}) := g(\max_{\mathbf{x} \in X} \mathbf{x}^\top \mathbf{u})$. By lemma B.11, $f$ is invariant under joint permutation by $S_d$. Furthermore, $f$ is measurable because $g$ and $\max$ are. Therefore, apply lemma B.5 to conclude that $f(X \mid \mathcal{D}_{\text{bound}}) := \mathbb{E}_{\mathbf{u} \sim \mathcal{D}_{\text{bound}}} \left[ g(\max_{\mathbf{x} \in X} \mathbf{x}^\top \mathbf{u}) \right]$ is also invariant under joint permutation by $S_d$ (with $f$ being bounded when restricted to $\text{supp}(\mathcal{D}_{\text{bound}})$). Lastly, if $X' \subseteq X$, $f(X' \mid \mathcal{D}_{\text{bound}}) \leq f(X \mid \mathcal{D}_{\text{bound}})$ because $g$ is increasing.

$$\mathbb{E}_{\mathbf{u} \sim \mathcal{D}_{\text{bound}}} \left[ g \left( \max_{\mathbf{a} \in A} \mathbf{a}^\top \mathbf{u} \right) \right] = \mathbb{E}_{\mathbf{u} \sim \mathcal{D}_{\text{bound}}} \left[ g \left( \max_{\mathbf{a} \in \text{ND}(A)} \mathbf{a}^\top \mathbf{u} \right) \right] \tag{143}$$

$$\leq^n_{\text{most: } \mathfrak{D}_{\text{any}}} \mathbb{E}_{\mathbf{r} \sim \mathcal{D}_{\text{bound}}} \left[ g \left( \max_{\mathbf{b} \in B} \mathbf{b}^\top \mathbf{r} \right) \right]. \tag{144}$$

Equation (143) follows by corollary E.11 of [Turner et al., 2021]. Equation (144) follows by applying lemma B.9 with $f$ as defined above with the $\phi_1, \dots, \phi_n$ guaranteed by the copy assumption. $\square$

**Definition D.4** (Linear functional optimality probability [Turner et al., 2021]). For finite $A, B \subsetneq \mathbb{R}^{|\mathcal{S}|}$, the *probability under $\mathcal{D}_{\text{any}}$ that $A$ is optimal over $B$* is

$$p_{\mathcal{D}_{\text{any}}} (A \geq B) := \mathbb{P}_{\mathbf{r} \sim \mathcal{D}_{\text{any}}} \left( \max_{\mathbf{a} \in A} \mathbf{a}^\top \mathbf{r} \geq \max_{\mathbf{b} \in B} \mathbf{b}^\top \mathbf{r} \right).$$

**Lemma D.5** (Quantitative optimality probability superiority lemma). *Let $A, B, C \subsetneq \mathbb{R}^d$ be finite and let $Z$ satisfy $\text{ND}(C) \subseteq Z \subseteq C$. Suppose that $B$ contains $n$ copies of $\text{ND}(A)$ via involutions $\phi_i$. Furthermore, let $B_{\text{extra}} := B \setminus \left( \cup_{i=1}^n \phi_i \cdot \text{ND}(A) \right)$; suppose that for all $i$, $\phi_i \cdot \left( Z \setminus B_{\text{extra}} \right) = Z \setminus B_{\text{extra}}$.*

*Then $p_{\mathcal{D}_{\text{any}}} (B \geq C) \geq^n_{\text{most: } \mathfrak{D}_{\text{any}}} p_{\mathcal{D}_{\text{any}}} (A \geq C)$.*

*Proof.* For finite $X, Y \subsetneq \mathbb{R}^d$, let

$$g(X, Y \mid \mathcal{D}_{\text{any}}) := p_{\mathcal{D}_{\text{any}}} (X \geq Y) = \mathbb{E}_{\mathbf{u} \sim \mathcal{D}_{\text{any}}} \left[ \mathbb{1}_{\max_{\mathbf{x} \in X} \mathbf{x}^\top \mathbf{u} \geq \max_{\mathbf{y} \in Y} \mathbf{y}^\top \mathbf{u}} \right].$$

By the proof of item 1 of proposition A.11, $g$ is the expectation of a $\mathbf{u}$-measurable function. $g$ is an EU function, and so lemma B.11 shows that it is invariant to joint permutation by $\phi_i$. Letting $f_Y(X \mid \mathcal{D}_{\text{any}}) := g(X, Y \mid \mathcal{D}_{\text{any}})$, lemma B.10 shows that $f_Y(X \mid \mathcal{D}_{\text{any}}) = f_Y(\phi_i \cdot X \mid \phi_i \cdot \mathcal{D}_{\text{any}})$ whenever the $\phi_i$ satisfy $\phi_i \cdot Y = Y$.

Furthermore, if $X' \subseteq X$, then $f_Y(X' \mid \mathcal{D}_{\text{any}}) \leq f_Y(X \mid \mathcal{D}_{\text{any}})$.

$$p_{\mathcal{D}_{\text{any}}} (A \geq C) = p_{\mathcal{D}_{\text{any}}} (\text{ND}(A) \geq C) \tag{145}$$

$$\leq p_{\mathcal{D}_{\text{any}}} (\text{ND}(A) \geq Z \setminus B_{\text{extra}}) \tag{146}$$

$$\leq^n_{\text{most: } \mathfrak{D}_{\text{any}}} p_{\mathcal{D}_{\text{any}}} (B \geq Z \setminus B_{\text{extra}}) \tag{147}$$

$$\leq p_{\mathcal{D}_{\text{any}}} (B \cup B_{\text{extra}} \geq Z) \tag{148}$$

$$= p_{\mathcal{D}_{\text{any}}} (B \geq Z) \tag{149}$$

$$= p_{\mathcal{D}_{\text{any}}} (B \geq C). \tag{150}$$

Equation (145) follows by Turner et al. [2021]'s lemma E.12's item 2 with $X := A, X' := \text{ND}(A)$ (similar reasoning holds for $C$ and $Z$ in eq. (150)). Equation (146) follows by the first inequality of lemma E.26 of [Turner et al., 2021] with $X := A, Y := C, Y' := Z \setminus B_{\text{extra}}$. Equation (147) follows by applying lemma B.9 with the $f_{Z \setminus B_{\text{extra}}}$ defined above. Equation (148) follows by the second inequality of lemma E.26 of [Turner et al., 2021] with $X := A, Y := Z, Y' := Z \setminus B_{\text{extra}}$. Equation (149) follows because $B_{\text{extra}} \subseteq B$.

Letting $f_0(\mathcal{D}_{\text{any}}) := p_{\mathcal{D}_{\text{any}}}(A \geq C)$, $f_1(\mathcal{D}_{\text{any}}) := p_{\mathcal{D}_{\text{any}}}(\text{ND}(A) \geq Z \setminus B_{\text{extra}})$, $f_2(\mathcal{D}_{\text{any}}) := p_{\mathcal{D}_{\text{any}}}(B \geq Z \setminus B_{\text{extra}})$, $f_3(\mathcal{D}_{\text{any}}) := p_{\mathcal{D}_{\text{any}}}(B \geq C)$, apply lemma B.1 to conclude that

$$p_{\mathcal{D}_{\text{any}}}(A \geq C) \leq^{n}_{\text{most: } \mathfrak{D}_{\text{any}}} p_{\mathcal{D}_{\text{any}}}(B \geq C).$$

$\square$

**Definition D.6** (Rewardless MDP [Turner et al., 2021]). $\langle \mathcal{S}, \mathcal{A}, T \rangle$ is a rewardless MDP with finite state and action spaces $\mathcal{S}$ and $\mathcal{A}$, and stochastic transition function $T : \mathcal{S} \times \mathcal{A} \to \Delta(\mathcal{S})$. We treat the discount rate $\gamma$ as a variable with domain $[0, 1]$.

**Definition D.7** (1-cycle states [Turner et al., 2021]). Let $\mathbf{e}_s \in \mathbb{R}^{|\mathcal{S}|}$ be the standard basis vector for state $s$, such that there is a 1 in the entry for state $s$ and 0 elsewhere. State $s$ is a *1-cycle* if $\exists a \in \mathcal{A} : T(s, a) = \mathbf{e}_s$. State $s$ is a *terminal state* if $\forall a \in \mathcal{A} : T(s, a) = \mathbf{e}_s$.

**Definition D.8** (State visit distribution [Sutton and Barto, 1998]). $\Pi := \mathcal{A}^{\mathcal{S}}$, the set of stationary deterministic policies. The *visit distribution* induced by following policy $\pi$ from state $s$ at discount rate $\gamma \in [0, 1)$ is $\mathbf{f}^{\pi,s}(\gamma) := \sum_{t=0}^{\infty} \gamma^t \mathbb{E}_{s_t \sim \pi|s}[\mathbf{e}_{s_t}]$. $\mathbf{f}^{\pi,s}$ is a *visit distribution function*; $\mathcal{F}(s) := \{\mathbf{f}^{\pi,s} \mid \pi \in \Pi\}$.

**Definition D.9** (Recurrent state distributions [Puterman, 2014]). The *recurrent state distributions* which can be induced from state $s$ are $\text{RSD}(s) := \{\lim_{\gamma \to 1}(1 - \gamma)\mathbf{f}^{\pi,s}(\gamma) \mid \pi \in \Pi\}$. $\text{RSD}_{\text{nd}}(s)$ is the set of RSDs which strictly maximize average reward for some reward function.

**Definition D.10** (Average-optimal policies [Turner et al., 2021]). The *average-optimal policy set* for reward function $R$ is $\Pi^{\text{avg}}(R) := \left\{\pi \in \Pi \mid \forall s \in \mathcal{S} : \mathbf{d}^{\pi,s} \in \arg\max_{\mathbf{d} \in \text{RSD}(s)} \mathbf{d}^{\top}\mathbf{r}\right\}$ (the policies which induce optimal RSDs at all states). For $D \subseteq \text{RSD}(s)$, the *average optimality probability* is $\mathbb{P}_{\mathcal{D}_{\text{any}}}(D, \text{average}) := \mathbb{P}_{R \sim \mathcal{D}_{\text{any}}}(\exists \mathbf{d}^{\pi,s} \in D : \pi \in \Pi^{\text{avg}}(R))$.

**Remark.** Theorem D.11 generalizes the first claim of Turner et al. [2021]'s theorem 6.13, and corollary D.12 generalizes the first claim of Turner et al. [2021]'s corollary 6.14.

**Theorem D.11** (Quantitatively, average-optimal policies tend to end up in "larger" sets of RSDs). *Let $D', D \subseteq \text{RSD}(s)$. Suppose that $D$ contains $n$ copies of $D'$ and that the sets $D' \cup D$ and $\text{RSD}_{\text{nd}}(s) \setminus (D' \cup D)$ have pairwise orthogonal vector elements (i.e. pairwise disjoint vector support). Then $\mathbb{P}_{\mathcal{D}_{\text{any}}}(D', \text{average}) \leq^{n}_{\text{most: } \mathfrak{D}_{\text{any}}} \mathbb{P}_{\mathcal{D}_{\text{any}}}(D, \text{average})$.*

*Proof.* Let $D_i := \phi_i \cdot D'$, where $D_i \subseteq D$ by assumption.
Let $S := \{s' \in \mathcal{S} \mid \max_{\mathbf{d} \in D' \cup D} \mathbf{d}^{\top} \mathbf{e}_{s'} > 0\}$.
Define

$$\phi'_i(s') := \begin{cases} \phi_i(s') & \text{if } s' \in S \\ s' & \text{else.} \end{cases} \tag{151}$$

Since $\phi_i$ is an involution, $\phi'_i$ is also an involution. Furthermore, $\phi'_i \cdot D' = D_i$, $\phi'_i \cdot D_i = D'$, and $\phi'_i \cdot D_j = D_j$ for $j \neq i$ because we assumed that these equalities hold for $\phi_i$, and $D', D_i, D_j \subseteq D' \cup D$ and so the vectors of these sets have support contained in $S$.

Let $D^* := D' \cup_{i=1}^{n} D_i \cup \left(\text{RSD}_{\text{nd}}(s) \setminus (D' \cup D)\right)$. By an argument mirroring that in the proof of theorem 6.13 in Turner et al. [2021] and using the fact that $\phi'_i \cdot D_j = D_j$ for all $i \neq j$, $\phi'_i \cdot D^* = D^*$. Consider $Z := \left(\text{RSD}_{\text{nd}}(s) \setminus (D' \cup D)\right) \cup D' \cup D$. First, $Z \subseteq \text{RSD}(s)$ by definition. Second, $\text{RSD}_{\text{nd}}(s) = \text{RSD}_{\text{nd}}(s) \setminus (D' \cup D) \cup (\text{RSD}_{\text{nd}}(s) \cap D') \cup (\text{RSD}_{\text{nd}}(s) \cap D) \subseteq Z$. Note that $D^* = Z \setminus (D \setminus \cup_{i=1}^{n} D_i)$.

$$\mathbb{P}_{\mathcal{D}_{\text{any}}}(D', \text{average}) = p_{\mathcal{D}_{\text{any}}}(D' \geq \text{RSD}(s)) \tag{152}$$

$$\leq^{n}_{\text{most: } \mathfrak{D}_{\text{any}}} p_{\mathcal{D}_{\text{any}}}(D \geq \text{RSD}(s)) \tag{153}$$

$$= \mathbb{P}_{\mathcal{D}_{\text{any}}}(D, \text{average}). \tag{154}$$

Since $\phi'_i \cdot D' \subseteq D$ and $\text{ND}(D') \subseteq D'$, $\phi'_i \cdot \text{ND}(D') \subseteq D$ and so $D$ contains $n$ copies of $\text{ND}(D')$ via involutions $\phi'_i$. Then eq. (153) holds by applying lemma D.5 with $A := D'$, $B_i := D_i$ for all $i = 1, \ldots, n$, $B := D, C := \text{RSD}(s)$, $Z$ as defined above, and involutions $\phi'_i$ which satisfy $\phi'_i \cdot (Z \setminus (B \setminus \cup_{i=1}^{n} B_i)) = \phi'_i \cdot D^* = D^* = Z \setminus (B \setminus \cup_{i=1}^{n} B_i)$. $\square$

**Corollary D.12** (Quantitatively, average-optimal policies tend not to end up in any given 1-cycle)**.**
*Let $D' := \left\{ \mathbf{e}_{s'_1}, \ldots, \mathbf{e}_{s'_k} \right\}, D_r := \left\{ \mathbf{e}_{s_1}, \ldots, \mathbf{e}_{s_{n \cdot k}} \right\} \subseteq \mathrm{RSD}\,(s)$ be disjoint, for $n \geq 1, k \geq 1$. Then*
$$\mathbb{P}_{\mathcal{D}_{any}}\left(D', \text{average}\right) \leq^n_{\text{most: } \mathfrak{D}_{any}} \mathbb{P}_{\mathcal{D}_{any}}\left(\mathrm{RSD}\,(s) \setminus D', \text{average}\right).$$

*Proof.* For each $i \in \{1, \ldots, n\}$, let
$$\phi_i := (s'_1 \ \ s_{(i-1)\cdot k+1}) \cdots (s'_k \ \ s_{(i-1)\cdot k+k}),$$
$$D_i := \left\{ \mathbf{e}_{s_{(i-1)\cdot k+1}}, \ldots, \mathbf{e}_{s_{(i-1)\cdot k+k}} \right\},$$
$$D := \mathrm{RSD}\,(s) \setminus D'.$$

Each $D_i \subseteq D_r \subseteq \mathrm{RSD}\,(s) \setminus D'$ by disjointness of $D'$ and $D_r$.

$D$ contains $n$ copies of $D'$ via involutions $\phi_1, \ldots, \phi_n$. $D' \cup D = \mathrm{RSD}\,(s)$ and $\mathrm{RSD}_{\mathrm{nd}}\,(s) \setminus \mathrm{RSD}\,(s) = \emptyset$ trivially have pairwise orthogonal vector elements.

Apply theorem D.11 to conclude that
$$\mathbb{P}_{\mathcal{D}_{any}}\left(D', \text{average}\right) \leq^n_{\text{most: } \mathfrak{D}_{any}} \mathbb{P}_{\mathcal{D}_{any}}\left(\mathrm{RSD}\,(s) \setminus D', \text{average}\right).$$

$\square$

Let $A := \{\mathbf{e}_1, \mathbf{e}_2\}, B \subseteq \mathbb{R}^5, C := A \cup B$. Conjecture D.13 conjectures that *e.g.*
$$p_{\mathcal{D}'}\,(B \geq C) \geq^{\frac{3}{2}}_{\text{most: } \mathfrak{D}_{any}} p_{\mathcal{D}'}\,(A \geq C).$$

**Conjecture D.13** (Fractional quantitative optimality probability superiority lemma)**.** Let $A$, $B$, $C \subsetneq \mathbb{R}^d$ be finite. If $A = \bigcup_{j=1}^m A_j$ and $\bigcup_{i=1}^n B_i \subseteq B$ such that for each $A_j$, $B$ contains $n$ copies $(B_1, \ldots, B_n)$ of $A_j$ via involutions $\phi_{ji}$ which *also* fix $\phi_{ji} \cdot A_{j'} = A_{j'}$ for $j' \neq j$, then
$$p_{\mathcal{D}_{any}}\,(B \geq C) \geq^{\frac{n}{m}}_{\text{most: } \mathfrak{D}_{any}} p_{\mathcal{D}_{any}}\,(A \geq C).$$

We suspect that any proof of the conjecture should generalize lemma B.7 to the fractional set copy containment case.