# OpenReview forum: "Parametrically Retargetable Decision-Makers Tend To Seek Power"
_NeurIPS.cc/2022/Conference — NeurIPS 2022 Accept_

### Official Review · Reviewer_Dyfy · 2022-07-07

**Rating:** 7
**Confidence:** 4
**Soundness:** 3 good
**Presentation:** 2 fair
**Contribution:** 4 excellent

**Summary:**

This paper defines retargetable algorithms and formally shows that they have power-seeking tendencies (i.e. make choices that leave more options open), building on existing work which showed that optimal policies tend to seek power. Since any algorithms that make decisions based on utility of outcomes are retargetable, this is a significant generalization of the previous work. The authors then apply the retargetability criterion to different algorithms in Montezuma's revenge and show that more generally capable algorithms are more retargetable and therefore are more likely to produce power-seeking policies.

**Questions:**

It would be helpful to add a notation paragraph defining key terms and variables:
* It's not entirely clear what a "decision-maker" is - I assume that a decision-maker p is a goal-conditioned policy, and instantiating it for a specific parameter theta gives a policy $f=p(\theta)$? (The term "algorithm" in section 4 seems to be used interchangeably with "decision-maker".) Is retargetability is property of the algorithm, and power-seeking a property of the policy?
* Section 3 is intended to give general definitions, but still refers to A and B as "boxes" as in the card example in Section 2. I assume A and B are supposed to be action sets?
* What does d represent in section 3? Is it the size of the action space?
* It's unclear what "highly retargetable decision-makers" means in Section 3 - are these multiply-retargetable functions with high n in def 3.5?
* Clarify the assumptions made about the environment (e.g. finite action space?)

Adding a related work section would really help clarify the relationship with existing work, e.g.:
* State explicitly how this paper generalizes results from Turner et al (2021).
* Summarize how this work relates to the papers on quantilization that are cited in the reference section but not mentioned in the main text.
* Since this paper doesn't focus on making the case for risks from power-seeking incentives, it would be good to refer to works that motivate the importance of this problem, e.g. Basic AI Drives (Omohundro 2008) and Existential Risk from Power-Seeking AI (Carlsmith 2021).

I would argue in favour of accepting this paper conditional on the above issues being addressed, which I expect can be done without major changes to the content of the paper. It would be helpful if the authors can include a notation paragraph and a related work paragraph in the author response if space allows.

**Limitations:**

The paper addresses some limitations in the second paragraph of section 6.1. It would be helpful to label this section as "Future work and limitations" to clarify this, since this paragraph is not about future work.

**Strengths And Weaknesses:**

This paper makes progress on understanding and formalizing the important problem of power-seeking incentives, which is considered the main mechanism for large-scale harm from advanced AI. It generalizes previous work by Turner et al (2021) that established power-seeking incentives for optimal policies. This paper demonstrates power-seeking tendencies for a broad class of non-optimal algorithms (in principle, any algorithm that chooses between outcomes based on utility), which implies that these incentives are likely to arise in practice for modern ML algorithms. While the previous work assumes a finite MDP setting, the results in this paper apply to more complex environments, e.g. as illustrated on the Montezuma's revenge game in Section 4. These results are novel and significant.

The main weaknesses of this paper are clarity of presentation and insufficient discussion of related work. I expect it to be difficult for readers who haven't read Turner et al (2021) to follow this paper, and I think it should be more self-contained (e.g. include the definition of power from the previous work and explain why definition 3.2 implies power-seeking).

The fictional dialogue in section 2 is intended to convey intuition, but I don't think it serves that purpose well, so I would suggest moving it to the appendix for clarity and conciseness. Something that would help build intuition for the concept of retargetability would be to provide other examples of non-retargetable decision makers (besides the trivially non-retargetable f_stubborn).

The application of the retargetability results to Montezuma's Revenge in Section 4 would more illuminating if it was more specific. For example, it would be great to clarify what are d, A and B in sections 4.2-4.4. Section 3 defines retargetability over action sets, while sections 4.2-4.4 make claims about retargetability over observations without defining what that means. In particular, it's unclear how retargetability over a continuous observation space works since the permutation $\phi$ is over a finite set of elements.

---

> ### Author Response · Authors · 2022-08-01
> **Response to Dyfy**
>
> Thank you for your comments.
>
> **Notation.** In the RL setting, $d$ is the size of the state space, but this is not required in general. Retargetability is a property of the policy training process, and power-seeking is a property of the trajectories chosen by a trained policy. More precisely, the policy training process takes as input a parameterization 𝜃 and outputs a probability distribution over policies. For each trained policy drawn from this distribution, the environment, starting state, and the drawn policy jointly specify a probability distribution over trajectories. Therefore, the training process associates each parameterization 𝜃 with the mixture distribution $P$ over trajectories (with the mixture taken over the distribution of trained policies).
>
> A policy training process can be _simply retargeted from one trajectory set $A$ to another trajectory set $B$_ when there exists a permutation $\phi\in S_d$ such that, for every 𝜃 for which $P(A\mid 𝜃)>P(B\mid 𝜃)$, we have $P(A\mid \phi\cdot 𝜃)<P(B\mid \phi \cdot 𝜃)$. For this work, we echo Turner et al. (2021) in saying that a trained policy $\pi$ _seeks power_ when $\pi$ actions which navigate to states with high optimal value for a wide range of reward functions. Generally, high-power states are able to reach a wide range of other states, and so allow bigger option sets $B$ (compared to the options $A$ available without seeking power).
>
> For the “box” issue in section 3, see footnote 1 on page 4; we will rewrite that portion to be more immediately clear as to what A and B are. “Highly-retargetable decision-makers” indeed means multiply-retargetable functions with high $n$.
>
> We do not assume a finite environment per se, although we currently don’t see how to apply our results to non-trivial infinite environments. Theorem 3.6 is intentionally abstract and noncommittal in terms of defining even the structure of A and B, so as to enable e.g. future work proving optimal policy tendencies in finite POMDPs, to our analysis of bandit situations in Appendix C.1.
>
> **Related work.** We will add the following: “In this work, we do not motivate the risks from AI power-seeking. We refer the reader to (Omohundro 2008) and (Carlsmith 2021). Turner et al. (2021) show that, given certain environmental symmetries in an MDP, the optimal-policy-producing algorithm $f_\gamma$(state visitation distribution set, state-based reward function) is 1-retargetable via the reward function, from smaller to larger sets of environmental options. Appendix A shows that optimality is not required, and instead a wide-range of decision-making procedures satisfy the retargetability criterion. Furthermore, we generalize from 1-retargetability to $n$-fold-retargetability whenever option set $B$ contains “$n$ copies” of set $A$ (definition A.6 in the appendix).”
>
> Quantilization is referenced in Appendix A, but we do not consider it important enough to mention in the main text. If you have a strong opinion here, we are open to further discussion.

---

### Official Review · Reviewer_a6si · 2022-07-08

**Rating:** 7
**Confidence:** 4
**Soundness:** 3 good
**Presentation:** 1 poor
**Contribution:** 4 excellent

**Summary:**

Previous work has shown that power-seeking emerges for many reward functions in optimal policies. This is because power-seeking refers to maximizing the number of options the agent has, and having more options allows the agent to maximize more possible reward functions.

This paper mathematically shows that a similar result holds for various algorithms that produce _suboptimal_ policies, specifically to algorithms that are "retargetable": A suboptimal algorithm may not be able to reach _every_ achievable option needed to maximize its reward function, but as long as we can always retarget the algorithm (for example by changing the reward function) to reach _some_ 'powerful' state which yields many options, it tends to reach such a state under many possible reward functions. Formally, an algorithm is defined as retargetable from target A to target B if, assuming it reaches A, we can change the algorithm's parameters (e.g. its reward function) so that it reaches B instead. Retargetable algorithms include random, greedy, and optimal decision-making among others.

The paper illustrates its conclusions through a toy example and formal analysis in Montezuma's Revenge. It also argues that RL algorithms become increasingly retargetable as they become more capable, so that we will increasingly observe power-seeking.


**Questions:**

How can retargetability imply power-seeking if the random policy is retargetable but intuitively doesn't seek power?

The paper shows that if an algorithm can be retargeted from some arbitrary target A to some target B, then most parameter choices (i.e. reward functions) will lead the agent to reach B. The paper concludes that the algorithm then 'seeks power' (in some undefined sense). The latter claim lacks clear justification. The paper does give some examples of such powerful events B (reaching the next room and picking a box with many cards) but a general argument or intuition seems to be missing.
    - For example, the event A could correspond to 'power' in which case the agent 'avoids power'. To say something about power-seeking, the theorems would have to specify which target (A or B) is more powerful.

An algorithm can be retargetable from a specific target A to target B but the paper sometimes talks about retargetable algorithms without refererring to specific targets. What does retargetable mean here?


**Limitations:**

Yes

**Strengths And Weaknesses:**

__Strengths__

Power-seeking is a property of intelligent agents that has long been hypothesized and has important consequences for safety. It is good to see that recently some formal progress is made to understand power-seeking (in the sense of preferring actions that lead to more open options).

But previous work assumes perfectly optimal agents acting in fully-observable environments--a strict assumption that poses the question if power-seeking is merely an artifact of optimal policies. The present paper answers that question in the negative.

It also introduces a sufficient condition for decision algorithms to seek power: retargetability. Having a sufficient condition should improve our understanding of when algorithms will seek power or not. Establishing this condition is highly mathematically non-trivial and provides tools for future analyses.

__Weaknesses__

Although the abstract and introduction are excellently written, the mathematical presentation is often confusing and needs more clarity and reduced ambiguity (see detailed comments). Although the ambiguities mostly resolve after very careful reading, they make the paper hard to read. This is likely to be a major problem for average readers. The authors could get more external feedback to remedy this.

I recommend accepting this paper, although I think it would have much more impact with a clearer presentation, because I think our community should reward authors for tackling important but difficult problems with hard-to-explain solutions in young subfields where readers will lack expertise. Additionally, as long as the conclusions are sound, most readers do not need to read the entire mathematical content of this paper.


__-------------------- Detailed comments ------------------------__

__Introduction and abstract__

These sections were a pleasure to read.

"A wide range of decision-makers share these power-seeking tendencies—they are not unique to reward maximizers." This is a key claim that lacks reference. I think you mean "We show that a wide range....".

In L33, the meaning of parameterizations is unclear. (The meaning of 'similar decisions' is also unclear but probably refers to seeking power?) Parameterizations usually refer to different choices for a change of variable. It is also unclear which parameters you refer to (algorithm parameters, not policy parameters).

The paper's conclusion but not its argument is given in the introduction (and the argument doesn't become very clear later). I've tried to spell out the intuitive arguments in my summary above, in case that is helpful.

__Section 2__

(Comments ordered by priority)

L 49 talks about "reparameterizing" the decision rule. This threw me off quite a bit since reparameterization normally refers to a change of variables but you seem to simply mean a change of the parameter theta without changing the parameter space Theta. The confusion is worsened because the text confuses lower case theta and upper case Theta multiple times.

Section 2 often refers to definitions and propositions in later sections. It is unclear if one should skip ahead and read those or not. Most readers will not do so, including myself, and this caused me to not understand the meaning of words like 'retargetable'.

Dialogue: Overall, the dialogue added lots of confusion for me without adding insight. If I wasn't reviewing the paper, I would have stopped reading here. I'd recommend to delete it or rewrite it from scratch. I think the rest of section 2 gets the point across on its own. Problems with the dialogue include but are not limited to: 1) The meaning of retargetable is unclear at this point. 2) Bob suggests that the agent could make decisions 'on a whim' and 'ignore the reward signal', but this seems nonsensical at first sight since that is not how RL works. So it was initially unclear why this suggestion is included in the dialogue. Perhaps a clearer formulation would be: "Alice: ...most parameters nudge the agent to pick a card from B ... Bob: But the parameter need not nudge the agent towards anything. For example, the decision-making rule could be a function that picks actions uniform randomly without using the parameter.

Table 1
    - The table doesn't clearly communicate its point without needing reference to the accompanying text. The point appears to be that "most utility function parameters incentivize the agent to draw a card from box B _because box B contains more options than A_. The point of permuting the utility function is also not immediately clear; it appears to be that most permutations induce picking from box B.
    - Consider bolding the best card

Consider renaming 'decision rule' to 'decision algorithm' since that is what the reader should usually have in mind.
Overall, the card example in section 2 is very simple (this is a good thing) so it should be possible and valuable to make the explanation short.

Picking box B is not a very intuitive conceptualization of power-seeking. Another example may be better.

__Section 3__

The point of the opening sentence was unclear. "requires that the parameters θ ∈ Θ be "modifiable"...": what does it mean to 'require' this? You mean you require that the orbit of θ is also in Θ?

Definition 3.3:
        - you mean "for all theta^A in Theta"?
        - A double "if" is followed by a single "then". This seems grammatically incorrect which obscures the definition's meaning.

"a set acted on by Sd, the symmetric group on d elements": Unclear. How is a set acted on by Sd different from a set not acted on by Sd?

Consider renaming symmetric group to permutation group for clarity. Even easier, rename it to 'set of permutations'.

Why do you consider specifically permutations and nothing else? Are these in some sense highly general and cover all interesting reward functions / parameters?

You could explain why B is not retargetable to A. (And why this implies the agent seeks power.)

What can the permutations refer to in practice other than the specific case of shuffling the rewards of a set of state-action pairs? In general, it is unclear why you need an abstract formalism that goes beyond this specific case.

"A parameter Θ’s orbit": did you mean lower-case theta? This problem appears multiple times.


__Section 4__

You could direct the reader to the most important results here, which are currently buried deep in the section.

The subsection on initial action selection is not specific to MR so could be placed in its own section. Readers who want to read specifically about MR do not need to read this subsection. In fact, your conclusions here can be broad whereas the paper's structure suggests that they only apply to MR.

Section 4.2: the paper is about power-seeking in suboptimal decision-makers, but the example f_max is optimal at reaching the target observation so this example seems less important.

Section 4.4: alpha_key and alpha_sword are not defined. At first it seemed that these were reward functions, not dimensions one reward function.

__Section 5__

This section reads a bit speculative since the concrete results (for example the formal results) are only in the appendix and the reader can't verify how useful those results are without reading the appendix.

Typos
    - Table and figure names are usually capitalized (also for clarity you could write "Table 1 (first row)" in L55).

Can you comment on how strong a requirement retargetability is? It seems fairly strong because it requires that something holds for every possible theta.

---

> ### Author Response · Authors · 2022-08-01
> **Response to a6si**
>
> Thank you for your detailed feedback. We agree that the presentation needs work, and think that your comments will substantially improve the paper.
>
> > How can retargetability imply power-seeking if the random policy is retargetable but intuitively doesn't seek power?
>
> The random-action policy is generally not retargetable in a sequential decision-making setting. Consider a situation where most possible outcomes require the agent to survive the first timestep, and suppose that only 1 in 10 available actions will allow the agent to survive. If an agent uniformly randomly selects an outcome and then implements a plan which realizes that outcome, then with extremely high probability (for most possible outcomes), the agent will stay alive at the first timestep. The randomness is over outcomes, not over actions. In contrast, the random policy only chooses the survival action with probability 1/10. (Lines 201–206 were meant to communicate this point. Apparently, that paragraph needs to be rewritten.)
>
> > The paper concludes that the algorithm then 'seeks power' (in some undefined sense).
>
> We made a pedagogical mistake here. Our submission implicitly assumes background provided by [Turner et al. 2021]. Insofar as their results show that optimal policies tend to seek power (in certain kinds of Markov decision processes), our results show the same for retargetable decision-makers (because we relax their optimality requirement in appendix A). We will justify and explain the power-seeking claims in the camera-ready.
>
> > An algorithm can be retargetable from a specific target A to target B but the paper sometimes talks about retargetable algorithms without refererring to specific targets. What does retargetable mean here?
>
> Consider the claim “GO-EXPLORE is a relatively retargetable algorithm.” We speculated that GO-EXPLORE is relatively retargetable within Montezuma’s Revenge, in the sense of most item-featurized reward functions training policies which leave the first room. We expect that even in a range of environments beyond Montezuma’s Revenge, GO-EXPLORE is better at exploring, and therefore more able to explore distant high-reward states, and therefore its trained policies account for distant high-reward states. GO-EXPLORE will therefore tend to have a relatively high retargetability towards a range of states, relative to eg DQN.

---

### Official Review · Reviewer_MMUx · 2022-07-11

**Rating:** 4
**Confidence:** 4
**Soundness:** 3 good
**Presentation:** 3 good
**Contribution:** 2 fair

**Summary:**

In this paper, the authors introduce the concept of "retargetable policies" and the application of this to models of decision making by artificial agents. Beginning with a motivating example of an agent choosing between two boxes containing playing cards with associated utilities, the paper then formalizes the concept of "retargetability". The paper then presents a case study of applying the framework that is presented to the Atari 2600 game Montezuma's revenge, and then discuss how this concept leads to power-seeking tendencies by the agents.

**Questions:**

Suggestions:
- The abstract (and paper's title) mention "retargetable policies" but these are not defined until later in the paper. At least a brief definition in the abstract would be helpful.
- The imagined dialogue within Section 2 is too lengthy and somewhat confusing. It's not clear why there is a leap from choosing the boxes with cards to training an RL agent to play Pac-Man.
- The comment on lines 116-117 of placing these results in the context of the Turner et al. (2021) paper only appears relevant if looking at the two papers side-by-side. Connecting these two papers, if this is truly important for the exposition here, could be done in a dedicated subsection for related work.
- The connection to RL in general needs to be clearer. For example, why do the agents need to be "trained via reinforcement learning" (line 301)?
- Looking at different RL algorithms in the context of this framework (rather than a cursory discussion of DQN and GO-EXPLORE) would be useful to be convincing about the generality of this framework.

**Limitations:**

The main limitations of this paper are how this relates to other decision-making approaches. This paper makes a logical leap from introducing a framework for evaluating the effect of permuting parameters of decision-making policies to the implied conclusion that this may result in AIs that seek power over humans. While this framework may have the potential for rich applications, drawing this kind of conclusion does not seem to be warranted given the discussion in the paper.

**Strengths And Weaknesses:**

Strengths:
- The paper presents a compelling way of looking at classes of reinforcement learning algorithms, in terms of ways of permuting rewards.
- The paper is generally written very clearly and the presentation is of high quality.

Weaknesses:
- There is too much reliance on the Turner et al. (2021) paper throughout. In particular, there is no separate discussion of "power" as defined in that paper. Presumably, the authors are using the same definition, but this needs to be clarified for this to be a stand-alone paper.
- Theoretical results that seem to be core to the results (specifically, the power-seeking results alluded to in Section 5.1) are not stated or presented until the Supplemental Material.
- The point in lines 278-280 about policies becoming more retargetable over "impressive outcomes" is vague and speculative. Since this seems to be important to the connection between this framework and the notion of power, it would be important to be more precise here.
- The different classes of RL algorithms are insufficiently explored here. For example, the point about DQN in lines 254-258 not being "good enough at exploring" is interesting but speculative. In particular, the statement on lines 257-258 that "[t]here isn't a single featurized reward function for which DQN visits other rooms" needs to be more rigorously defended.

---

> ### Author Response · Authors · 2022-08-01
> **Response to MMUx**
>
> Thank you for your feedback.
>
> > The imagined dialogue within Section 2 is too lengthy and somewhat confusing. It's not clear why there is a leap from choosing the boxes with cards to training an RL agent to play Pac-Man.
>
> Several other reviewers agreed. We will cut the dialogue.
>
> > The comment on lines 116-117 of placing these results in the context of the Turner et al. (2021) paper only appears relevant if looking at the two papers side-by-side. Connecting these two papers, if this is truly important for the exposition here, could be done in a dedicated subsection for related work.
>
> We will add a subsection discussing the notion of power in Turner et al. (2021) and its relationship to the current paper. We agree that it is needed to make the paper self-contained.
>
>  > The connection to RL in general needs to be clearer. For example, why do the agents need to be "trained via reinforcement learning" (line 301)? Looking at different RL algorithms in the context of this framework ...
>
> We mentioned reinforcement learning because it is currently the most widely used approach to train AI agents. As you correctly implied, our framework applies equally well for other retargetable functions such as MDP planning agents and agents that learn by imitation. We will include a wider variety of planning agents to make the discussion more general.
>
> > Theoretical results that seem to be core to the results (specifically, the power-seeking results alluded to in Section 5.1) are not stated or presented until the Supplemental Material.
>
> We extended Turner et al.’s power-seeking results. While these results are significant, the main paper does not allow space for explaining and stating those results. We chose to present the key retargetability result via Theorem 3.6, as most of our other important results follow as corollaries. As mentioned to reviewer a6si, we will contextualize and explain how retargetability relates to power-seeking:
>
> “Our submission implicitly assumes background provided by [Turner et al. 2021]. Insofar as their results show that optimal policies tend to seek power (in certain kinds of Markov decision processes), our results show the same for retargetable decision-makers (because we relax their optimality requirement in appendix A). We will justify and explain the power-seeking claims in the camera-ready.”
>
> > The different classes of RL algorithms are insufficiently explored here. For example, the point about DQN in lines 254-258 not being "good enough at exploring" is interesting but speculative. In particular, the statement on lines 257-258 that "[t]here isn't a single featurized reward function for which DQN visits other rooms" needs to be more rigorously defended.
>
> We can infer this is true from _Massively parallel methods for deep reinforcement learning_ (2015), which points out how vanilla DQN gets zero score in Montezuma's Revenge. Thus, DQN never even gets the first key. Thus, DQN only experiences state-action-state transitions which didn't involve acquiring an item. In our analysis, we considered a reward function which is featurized over item acquisition. Therefore, for all pre-key-acquisition state-action-state transitions, the featurized reward function returns exactly the same reward signals as those returned in training during the published experiments (namely, zero, because DQN can never even get to the key in order to receive a reward signal). That is, since DQN only experiences state-action-state transitions which didn't involve acquiring an item, and the featurized reward functions only reward acquiring an item, it doesn't matter what reward values are provided during item acquisition—DQN's trained behavior will be the same.
>
> Thus, a DQN agent trained on any featurized reward function will not explore outside of the first room. We will add this point to the paper to clarify for readers.

---

### Official Review · Reviewer_CTGR · 2022-07-18

**Rating:** 6
**Confidence:** 4
**Soundness:** 3 good
**Presentation:** 2 fair
**Contribution:** 4 excellent

**Summary:**

This paper extends “optimal policies tend to seek power” [Turner et al. 2021], arriving at similar results using only a novel notion of “retargetability”.  This suggests that a wide range of “parametrically retargetable decision-makers” will seek power, and more competent learners are more likely to.  The “Parametrically retargetable” here does *not* refer to the parameters of the model, but rather things like hyperparameters, or parameters of a reward function, which influence the outcome of learning.

The phrase “tends to” refers to Definition 3.2; per this definition, a decision-maker “tends to” do X if, for *any* setting of these parameters, most permutations of them result in it doing X.  This requires assuming the parameters are acted on by a permutation group; in the examples provided, the parameters are elements of $\mathbb{R}^n$ and this action works by permuting dimensions.  This can be viewed as a way to talk about what fraction of a space of infinite volume parameter space results in the decision-maker doing X, but I’m not sure how seriously to take that analogy (which is also suggested by line 128, but not discussed in any detail).  The central result is that “retargetability” implies such permutation-orbit-level tendencies.  A decision-maker can be retargeted to do X if there is a single fixed permutation $\phi$ such that: for any $\theta$ for which it didn’t do X, it *does* do X for $\phi \theta$.  These definitions are phrased in terms of tending to do X *over Y* and retargetting *from Y* to X, but I’m not sure how important that is; I think what I’ve described is a simpler special case.

The latter sections of the paper discuss how these results apply to various forms of decision-making, ending with a discussion of reinforcement learning which argues that more advanced RL algorithms will be more retargetable and hence more power-seeking.


**Questions:**

Questions:
* Why does Definition 3.2 require 2 functions?  Is it appropriate to talk about this in terms of the “simpler special case” as in my summary?
* Are lines 259-262 just restating the definition of retargetability?  What purpose does that serve?
* Do you agree with the first weakness I listed?  Or do the results do more than formalize that argument, i.e. providing a stronger argument for power-seeking?  How does this affect the practical significance of the results?
* Why the change from $f_B(\theta)$ to $f(B,\theta)$?  Is this related to footnote 1?
* I understand RL and similar algorithms/settings to be the motivation for the work.  But how specific are the results to RL?  Are there useful take-aways for supervised learning?  Can you be more explicit about the scope?
* Can you elaborate on the counting argument (143)?  Given the centrality of this result, I think it would be worth walking through it in more detail.  One thing in particular that might help build intuition is to show how it could (hypothetically) fail to hold, e.g. a concrete example where cosets are not pairwise disjoint.

Suggestions:
* I think the dialogue and example in Section 2 are useful, but they aren’t a replacement for directly explaining the intuition of the work and the concepts involved in text.  In particular, I think retargetability should be defined informally before the dialogue.
* Overall, I would suggest a significant rewrite to focus much more on clearly explaining the core concepts, with less space spent on Section 4 (e.g. you could probably cut an entire subsection).
* The central concepts and definitions should be clearly emphasized; orbit-level tendencies and retargetability might each warrant a subsection including a paragraph of motivation, a paragraph with an informal definition, and a formal definition.
* Overall, the writing is a bit light on “scaffolding”, i.e. statements that guide the reader through the work, and help them keep track of what role each section/paragraph/etc. is playing in the bigger picture narrative/development.
* I think “parameters” and “$\theta$” are potentially confusing here, since they do not refer to parameters of the model, but rather the learning algorithm/process.  More generally, you should discuss more what these “parameters” might represent, e.g. beyond reward functions.  Should we consider hyperparameters like learning rate part of $\theta$?  These are part of how desired behavior is specified in practice…
* The terminology “decision-makers” is also a bit off, I think, since it is more like learners that might also be doing decision-making.
* I think $\sigma$ is more commonly used to denote a permutation, and would be preferable here; this would free up $\phi$ as a potential replacement for $\theta$.
* The argument of lines 300-302 should be emphasized and spelled out.  How exactly does the theory yield this prediction?  Which results contribute to that prediction and how?
* I disagree with lines 304-305.  We know that reward functions can be misspecified, and they are arguably better viewed as a method of providing a useful training signal for a policy rather than a definition of what optimal behavior would look like.
* 264: $\Theta^{++}$ is undefined.
* 268: clarify that fig 2 is in the Appendix
* 262-274 is hard to understand without knowing details of Montezuma’s revenge.
* Cite the earlier version of GO-EXPLORE as well, to preempt misunderstanding that Montezuma’s revenge was only solved in 2021.
* I think 194 should say “4-retargetable”
* It should be made explicit that the symmetric group acts via permuting dimensions (and which dimensions…) in the examples given.
* 17: replace “;” with “and”
* 21-29 are a bit unclear, probably because they are too terse (e.g. it is not yet clear what is meant by “parameter inputs”).
* Are orbit tendencies well-characterized as “the fraction of $\theta$ for which…” (128)?  If so, elaborate/explain/defend this way of speaking.


**Strengths And Weaknesses:**


Strengths:
* Power-seeking is one of the most important concepts in AI existential safety.
* This paper is one of a very few to make a substantive technical contribution to this topic, and I believe it represents meaningful progress on understanding power-seeking.
* The insight that retargetability is sufficient for power-seeking is significant and highly original.

Weaknesses:
* This approach to understanding power-seeking doesn’t really address the main mechanisms by which we might expect to avoid building power-seeking agents: inductive biases and targeted feedback.  I view this paper as formalizing the argument that “alignment is hard because there are so many outcomes which involve human extinction, and we need to somehow direct the impact of AI systems towards the small fraction of those which do not”.  However, this argument doesn’t address the difficulty of alignment *relative to* the tools we have for alignment.
* The central points and ideas of the paper were not very clear.  I believe these are novel and subtle, and thus difficult to communicate effectively, but I think there is significant room for improvement.
* The phrase “tends to” is not explicitly defined, but should be, given its central role.  It should also be explained and justified why the formal definition matches common use (to the extent it does).
* I believe $f_{blah}$ is overloaded in a confusing way, referring to the probability of an outcome, something a bit more generic (footnote 1), or a decision-making process.
* The connection between the formally established orbit-level tendencies and power-seeking wasn’t clear enough, and relies a bit too much on Turner et al. 2021, I think.  I recommend front-loading something like some of the discussion in Section 5, e.g. into the introduction or at the end of Section 3.

Overall, I think there are many opportunities for improving the clarity and exposition of this work.  I believe it would benefit from a substantial rewrite (more than would be appropriate for a revision), which might significantly increase its impact.  Still, I think the contributions are sound and important enough to (weakly) recommend acceptance.

---

> ### Author Response · Authors · 2022-08-01
> **Response to CTGR**
>
> We are excited that you consider the retargetability insight to be original, and thank you for your insightful advice and remarks.
>
> > Why does Definition 3.2 require 2 functions? Is it appropriate to talk about this in terms of the “simpler special case” as in my summary?
>
> We originally intended backwards compatibility with the notation of Turner et al. (2021). However, we could also just create a single function $f$ by "uncurrying": $f(A\mid \theta) := f_A(\theta), f(B\mid \theta) := f_B(\theta)$. We will change the definition to this for clarity.
>
> The simpler special case is worth considering, but neglects probabilistic outcomes (e.g. a 30% chance of X given $\theta$, and a 70% chance of X given $\phi \cdot \theta$).
>
> > Do you agree with the first weakness I listed?
>
> Yes. However, we view your point as a limitation rather than a weakness of the paper.
>
> > Why the change from $f_B(\theta)$ to $f(B,\theta)$?
>
> Just a formatting inconsistency. Thanks for pointing it out.
>
> > But how specific are the results to RL? Are there useful take-aways for supervised learning? Can you be more explicit about the scope?
>
> We are uncertain about e.g. implications for supervised learning. We do not presently see how to show results beyond the planning and RL contexts. We will note this prospect in the paper.
>
> > One thing in particular that might help build intuition is to show how it could (hypothetically) fail to hold, e.g. a concrete example where cosets are not pairwise disjoint.
>
> There’s a trivial example where $A=B$, $n=2$, and $\phi_1=\phi_2$, such that the cosets are identical (and thus not pairwise disjoint), in which case 2-retargetability doesn’t hold. More helpfully, we do have a nontrivial example. It’s Table 3 on page 18 in Appendix C. However, the example is quite arcane and probably not worth the trouble to explain, and we do not see an obvious fix to that problem. We will keep considering whether there are better ways to give more intuition for the counting argument.

---

### Comment · Reviewer_a6si · 2022-08-05
**Summary of reviews on clarity; request**

Although all reviewers seem to think the work is original, non-trivial, and significant, multiple reviewers have said that the paper would have more impact with better presentation / clarity.

To recommend this paper without hesitation, I would expect that during the discussion period, the authors lay out a fairly detailed (and clear) plan for how they'll improve clarity. (For example, by listing how they'll address each feedback point.)

Some feedback on clarity from other reviewers and myself that I'd highlight (non-exhaustive):

- "Overall, the writing is a bit light on “scaffolding”" (see chap. 3-6 [here](https://sites.duke.edu/niou/files/2014/07/WilliamsJosephM1990StyleTowardClarityandGrace.pdf) for guidance)
- Clarifying the connection between orbit-level tendencies and power-seeking
- Explaining retargetability earlier. Perhaps with a better example that more clearly connects retargtability to power-seeking than the cards example does.
- Clarify meaning of 'parameters' and 'decision maker' earlier
- Illustrating core concepts more; possibly while cutting parts of section 4 and the dialogue for space.
- Concretize / justify speculative parts in section 4.4 and 5

I'd also recommend to get another round of feedback from colleagues after making these changes, as the content is evidently complex to communicate.

---

> ### Author Response · Authors · 2022-08-08
> **Plan for clarity**
>
> Thank you for gathering some of the feedback on clarity. Here are some details concerning our current plan:
> > "Overall, the writing is a bit light on “scaffolding”"
>
> In the beginning of each section, we will add signposting and scaffolding. For example, at the beginning of section 3, we will write:
>
> "Section 2 informally illustrated parametric retargetability in the context of swapping which utilities are assigned to which outcomes in the Pac-Man video game. Swapping the utility assignments also swapped the agent's final decisions. For example, if death is anti-rational, and then death's utility is swapped with the cherry utility, then now the cherry is anti-rational. In this section, we formalize the notion of parametric retargetability and of ``most'' parameter inputs producing a given result. In section 4, we will use these formal notions to reason about the behavior of RL-trained policies in the Montezuma's Revenge video game."
>
> > Explaining retargetability earlier. Perhaps with a better example that more clearly connects retargtability to power-seeking than the cards example does.
>
> Instead of the cards example, we will build off of Turner et al.'s Pac-Man example: The agent can choose between dying immediately, or finishing in a terminal state where it has collected a cherry, and finishing in a terminal state where it has collected an apple. We will explain the example as follows:
>
> "Turner et al. consider the Pac-Man video game, in which an agent consumes pellets, navigates a maze, and avoids deadly ghosts. Instead of the usual score function, Turner et al. consider optimal action across a range of state-based reward functions. They show that most reward functions have an (average-)optimal policy which avoids immediate death in order to navigate to a future terminal state.
>
> Our results show that optimality is not required. Instead, if the agent's decision-making is \emph{parametrically retargetable} from death to other outcomes, Pac-Man avoids the ghost under most decision-making parameter inputs. To build intuition about these notions, consider three outcomes: Immediate death to a nearby ghost, consuming a cherry, and consuming an apple.
>
> We begin with the optimality argument, following Turner et al. Suppose that the agent has a utility function $\mathbf{u}$ assigning a real number to each outcome. For example, death could have utility 10, the cherry 5, and the apple 0. Then an agent which maximizes $\mathbf{u}$ would die to the ghost. However, most ``variants" of $\mathbf{u}$..."
>
> We will continue this example appropriately.
>
> > Clarifying the connection between orbit-level tendencies and power-seeking
>
> At the end of section 2, we will add the following paragraph:
>
> "The larger set of outcomes {cherry, apple} can only be induced if Pac-Man stays alive. Intuitively, navigating to this larger set is power-seeking, because the agent retains more optionality (i.e. the agent can't do anything when dead). Furthermore, for most parameter settings, retargetable decision-makers induce an element of the larger set of outcomes. Therefore, we say that \emph{parametrically retargetable agents tend to seek power}."
>
> > Clarify meaning of 'parameters' and 'decision maker' earlier
>
> In our reply to Reviewer Dyfy, we included an explanation of these concepts. We will include this explanation early in the paper.
>
> > Illustrating core concepts more; possibly while cutting parts of section 4 and the dialogue for space.
>
> In addition to the above modifications, we will cut the dialogue and move section 4.3 to Appendix C.3.
>
> > Concretize / justify speculative parts in section 4.4 and 5
>
> Concerning DQN being unable to explore given any item-featurized reward signal, we will include the which we explanation provided to MMUx.
>
> In our reply to Reviewer Dyfy, we also provided a "related work" paragraph which we will include in camera-ready.

---

### Meta-Review · Area_Chair_qp6i · 2022-08-29

**Recommendation:** Accept
**Confidence:** Less certain

**Metareview:**

The paper studies an alignment problem - that of agent seeking powers, and extends previous work (Turner, 2021 - which showed that optimal policies seek power, to demonstrate more generally that parametrically retargetable policies (policies whose 'target' can be changed by simple change of hyperparameters of the agent) also tend to seek power. The problem is interesting and under-studied, and all reviewers agreed that the work was 'original, non-trivial and significant'. Most concerns were regarding presentation, which could be at times vague and imprecise (in the mathematical parts) or unintuitive (in the informal parts). The authors presented a plan to significantly address clarity of the paper, which alleviated many of the reviewers concern. Please do ensure that the final version include these improvements.

**Award:**

No

---

### Decision · Program_Chairs · 2022-09-14

Accept